# Sparse, Efficient and Explainable Data Attribution with DualXDA

**Galip Ümit Yolcu***
*Fraunhofer Heinrich Hertz Institute*

**Moritz Weckbecker***
*Fraunhofer Heinrich Hertz Institute*

**Thomas Wiegand**
*Fraunhofer Heinrich Hertz Institute*
*Technische Universität Berlin*
*BIFOLD – Berlin Institute for the Foundations of Learning and Data*

**Wojciech Samek**[†]                                    *wojciech.samek@hhi.fraunhofer.de*
*Fraunhofer Heinrich Hertz Institute*
*Technische Universität Berlin*
*BIFOLD – Berlin Institute for the Foundations of Learning and Data*

**Sebastian Lapuschkin**[†]                             *sebastian.lapuschkin@hhi.fraunhofer.de*
*Fraunhofer Heinrich Hertz Institute*
*Technological University Dublin*

**Reviewed on OpenReview:** *https://openreview.net/forum?id=qfx81N884A*

## Abstract

Data Attribution (DA) is an emerging approach in the field of eXplainable Artificial Intelligence (XAI), aiming to identify influential training datapoints which determine model outputs. It seeks to provide transparency about the model and individual predictions, e.g. for model debugging, identifying data-related causes of suboptimal performance. However, existing DA approaches suffer from prohibitively high computational costs and memory demands when applied to even medium-scale datasets and models, forcing practitioners to resort to approximations that may fail to capture the true inference process of the underlying model. Additionally, current attribution methods exhibit low sparsity, resulting in non-negligible attribution scores across a high number of training examples, hindering the discovery of decisive patterns in the data. In this work, we introduce DualXDA, a framework for sparse, efficient and explainable DA, comprised of two interlinked approaches, Dual Data Attribution (DualDA) and eXplainable Data Attribution (XDA): With DualDA, we propose a novel approach for efficient and effective DA, leveraging Support Vector Machine theory to provide fast and naturally sparse data attributions for AI predictions. In extensive quantitative analyses, we demonstrate that DualDA achieves high attribution quality, excels at solving a series of evaluated downstream tasks, while at the same time improving explanation time by a factor of up to 4,100,000× compared to the original Influence Functions method, and up to 11,000× compared to the method's most efficient approximation from literature to date. We further introduce XDA, a method for enhancing Data Attribution with capabilities from feature attribution methods to explain why training samples are relevant for the prediction of a test sample in terms of impactful features, which we showcase and verify qualitatively in detail. Taken together, our contributions in DualXDA ultimately point towards a future of eXplainable AI applied at unprecedented scale, enabling transparent, efficient and novel analysis of even the largest neural architectures – such as Large Language Models – and

---

*Equal contribution by authors.
[†]Corresponding authors.

fostering a new generation of interpretable and accountable AI systems. The implementation of our methods, as well as the full experimental protocol, is available on `github`[1].

## 1 Introduction

Despite the remarkable achievements of deep learning approaches, these methods remain black-boxes due to the opacity of their information processing procedures. Explainable AI (XAI) (Samek et al., 2019) has emerged to address the need to elucidate the inference processes of these models. Specifically, *global* XAI aims to provide insights into overall model characteristics, while *local* XAI explains outputs for individual test samples. Early approaches have been centered on feature attribution (Bach et al., 2015; Ribeiro et al., 2016), assigning an attribution score to each input feature, which indicates *which part* of the input is more or most influential in determining the model output. Later approaches provide different explanation paradigms, such as concept-based (Kim et al., 2018; Achtibat et al., 2023) and counterfactual explanations (Guidotti, 2024).

Feature-centered approaches put a strong emphasis on model inputs, parameters and representations but ignore the training data that the model has observed during optimization. A relatively recent lens of interpretability, Data Attribution (DA) attributes the model outputs to samples of the training dataset, producing attribution scores for each (training) datapoint. This is a complementary approach to feature attribution, allowing to rank the training data with respect to their effect in determining the model parameters (global DA) or specific outputs (local DA).

Data quality is integral in order to obtain robust model performance: Biased sampling (Lapuschkin et al., 2019), poor data quality management (Lin et al., 2022) or adversarial attacks (Shafahi et al., 2018; Zhu et al., 2019), can drive the model to suboptimal solutions. Therefore, in addition to understanding and debugging the decision processes of the model, DA methods have found applications in identifying and solving data-related problems, such as detecting mislabeled samples (Koh & Liang, 2017; Yeh et al., 2018; Pruthi et al., 2020), designing (Fang et al., 2020) and counteracting backdoor attacks (Hammoudeh & Lowd, 2022). Furthermore, these attribution methods have demonstrated effectiveness in dataset distillation (Liu et al., 2021; Naujoks et al., 2024), i.e. finding small subsets of training data which can be used to optimize on without having to sacrifice model performance. This is an important step towards the creation of efficient training loops for deep learning models, which usually require large amounts of data and energy for large scale training in order to be effective predictors (Thompson et al., 2020).

In practice, the utility of most DA methods is limited by two key factors: First, state-of-the-art methods depend on the estimation of inverse Hessian vector products, which is computationally expensive. While recent work has improved the efficiency of several approaches to DA with the intent to make them feasible for applications on larger language models such as BART or BERT (Guo et al., 2021; Ladhak et al., 2022), these methods remain challenging to scale and are often impractical under strict computational constraints, limiting their accessibility in real-world applications. Some methods can be adjusted to reduce computational load by resorting to lower quality estimations of attributions. Still, the application of these methods in time-critical scenarios, where attributions are required in the order of seconds or less following inference, is exceedingly difficult due to the tightly imposed temporal constraints contrasting their computational complexity.

Secondly, many existing DA methods produce dense attribution scores, i.e., they assign non-negligible importance to a large portion of the training dataset. In contrast, sparse attributions, where only a small subset of candidate components receive substantial credit, can lead to explanations of lower complexity, which have been shown to enhance human interpretability (Colin et al., 2022). The connection between sparsity and interpretability has been well-established in feature attribution literature, where sparse explanations mitigate cognitive load and improve the users' ability to understand model behavior (Chalasani et al., 2020; Warnecke et al., 2020; Bhatt et al., 2021). This desideratum receives empirical support from Ilyas et al. (Ilyas et al., 2022), who demonstrate that ground-truth data attributions are inherently sparse, with only a small subset of training samples exerting influence on the model output predicted on individual test points. To enforce the effect of attribution sparsity, an additional (artificial) sparsification step is sometimes applied to non-sparse methods (Park et al., 2023). This process discards a large portion of the attributions, which in turn,

---

[1] https://github.com/gumityolcu/DualXDA

may potentially result in attributions that do not reflect the actual prediction patterns of the model being explained. Regarding explanations of black-box models, the property of closely representing the inference of a model it aims to explain is referred to as "faithfulness" in XAI literature (Hedström et al., 2023). Therefore, sparse attributions that are also faithful to the model are highly preferred in DA.

To address both problems, we turn to established machine learning algorithms, where the influence of individual training data points has already been studied in detail. Concretely, we build upon the concept of Support Vector Machines (SVMs) (Cortes & Vapnik, 1995). SVMs perform linear classification by optimizing weight vectors in terms of a linear combination of a small subset of the training data. This allows the derivation of an alternative formulation of the prediction function as a sum of independent contributions relating to each training sample, by solving the associated dual problem (Crammer & Singer, 2001). Kernel SVMs, in turn, learn linear classifiers on nonlinear representations of training samples, which are then used to perform inference on test data. Therefore, given a neural network predictor and assuming the penultimate layer thereof as a mapping into a reproducing kernel Hilbert space, we build a SVM-based surrogate model replacing the final layer of the network, yielding an interpretable and faithful surrogate for the whole model.

Finally, we address another limitation of DA methods: the attribution values themselves are opaque, providing no insight into why certain training datapoints are deemed important for a given prediction. To this end, we combine data attribution with feature attribution to identify relevant features that explain the attribution values in both training and test input domains simultaneously.

In our paper, we make the **following contributions**:

- We introduce DualDA: a novel DA method which employs multiclass kernel SVMs (Crammer & Singer, 2001) as surrogates for deep learning models. Based on their dual problem, we derive sparse and high-quality explanations, both locally and globally, in a time- and memory-efficient manner.

- We evaluate DualDA in terms of quality and sparsity of explanations, and runtime as well as memory efficiency. In total, we compare nine different methods on seven quantitative metrics across three different datasets and models, in the most extensive quantitative evaluation of prominent DA methods so far. We find that DualDA performs at state-of-the-art while providing an up to $4{,}100{,}000\times$ speed-up compared to the least efficient competing method and up to $11{,}000\times$ compared with the SoTA implementations of Influence Functions. These efficiency gains meaningfully expand the practical usability of data attribution, including scenarios where existing methods are too computationally demanding.

- We prove that DualDA attribution scores agree within the multiclass kernelized SVM setting, on almost all datapoints, with attributions produced by both the gradient-based Influence Functions framework (Koh & Liang, 2017) (with minor modification) and the kernel surrogate-based Representer Points framework (Yeh et al., 2018).

- We also introduce XDA: a method that leverages the popular Layer-wise Relevance Propagation (LRP) (Bach et al., 2015; Montavon et al., 2019; Samek et al., 2021; Achtibat et al., 2024) – a state-of-the-art feature attribution (FA) method – and eXplains attribution values assigned to samples by mapping them back through the model to the (input) feature space. This yields insights as to why a certain training datapoint is important by identifying relevant features – such as pixels or tokens – both in the training and test input domains simultaneously. To our knowledge, XDA is the first method to enable a seamless integration of feature attribution and data attribution, offering joint explanations that connect model behaviour across feature and data provenance dimensions.

- We demonstrate how our DualXDA framework can be used to explain model decisions with both data attribution values and feature attribution explanations in case studies across three different datasets, substantiating the abilities of DualDA and XDA to explain counterfactuals via an ablation study.

In summary, we propose DualDA as an efficient, sparse and effective alternative to existing Data Attribution methods. Building on this foundation, we present XDA as a bridge to feature attribution, enabling – for the first time – joint, interpretable explanations that connect influential training samples to meaningful input

features. Together, these components form the DualXDA framework, providing novel opportunities for deeper, more actionable analyses into the inner workings of modern neural networks.

## 2 Related Work

Let us assume a supervised learning regime with given training data $\mathcal{D} = \{z_i = (x_i, y_i) \mid i \in [N]\}$, where $x_i \in \mathbb{R}^d$ are samples from the input domain and $y_i \in [K]$ are the corresponding class labels. A neural network $\Phi(\cdot; \theta) : \mathbb{R}^d \to \mathbb{R}^K$ with trained parameter $\theta$ takes a sample from the input domain and emits logits for all classes, which can be turned into class probabilities or a class prediction. In the following, we consider networks with a final fully-connected layer: In this case, the network parameters consist of a weight matrix $W \in \mathbb{R}^{K \times d_f}$ and remaining parameters $\vartheta$ so that the network function can be split into a feature extractor $f(\cdot; \vartheta) : \mathbb{R}^d \to \mathbb{R}^{d_f}$ consisting of the neural network from the input up to the final layer and a final affine layer $g(\cdot; W) : \mathbb{R}^{d_f} \to \mathbb{R}^K$:

$$\Phi(\cdot; \theta) = (g \circ f)(\cdot; \vartheta, W)$$

To incorporate a bias term into this formulation, the extracted feature vectors can be extended by appending a constant value of 1.

### 2.1 What is Data Attribution?

The goal of Data Attribution (DA) is to identify training data that have been influential in determining the prediction of a model, or the model itself.

**Definition 2.1.** *Global* DA methods assign each pair $(z_i, c)$ of training sample $z_i \in \mathcal{D}$ and class $c \in [K]$ a real-valued attribution which indicates the relevance or influence of the training datapoint on the fit of the model $\Phi(\cdot; \theta)_c$. *Local* DA methods produce a function $\tau_c : \mathbb{R}^d \times [N] \to \mathbb{R}$ for each class $c$. The inputs to this function are a test sample and the index for a training point. The value $\tau_c(x, i)$ indicates the relevance or influence of the training datapoint $z_i$ on the model output $\Phi(x; \theta)_c$.

DA has historically been framed as the problem of approximating the ground-truth effects of model retraining under altered training data compositions (Koh & Liang, 2017; Park et al., 2023). Recent work (Wei et al., 2024) advocates for a different standpoint, aiming to estimate the changes induced by fine-tuning an already trained model with an altered training set instead. Other authors do not claim for their attribution values to approximate a ground truth, instead they are interested in the utility of their attribution values in fulfilling certain downstream tasks. In all of these formulations, the resulting attributions are interpreted such that positive attribution means the training sample contains evidence for the model decision while negative attribution is indicative of evidence against the decision being explained.

While there are many different settings that DA has been studied in, we consider a *post-hoc attribution* setup, where we are given the training dataset, knowledge about training hyperparameters, and model parameters collected at different epochs of a single training run.

### 2.2 Data Attribution Methods

Existing DA methods can be broadly categorized into four different groups, in terms of how they produce attribution scores, as listed below. Detailed explanations and formulas for all methods can be found in Appendix D.1.

**Similarity-based approaches** compute the similarity between the training and test samples, building on the intuition that the model inferences should be informed of training samples that produce similar latent representations (Pezeshkpour et al., 2021), or through other measures of similarity (Singla et al., 2023). However, these attributions are *class-unspecific* as the measured similarity is independent of the predicted class.

**Gradient-based approaches** estimate the effect of retraining on a new training set where a sample has been removed from the training corpus. In this framework, a positive attribution indicates a decrease in loss, which indicates more confidence for the associated class. To this end, these methods employ Taylor

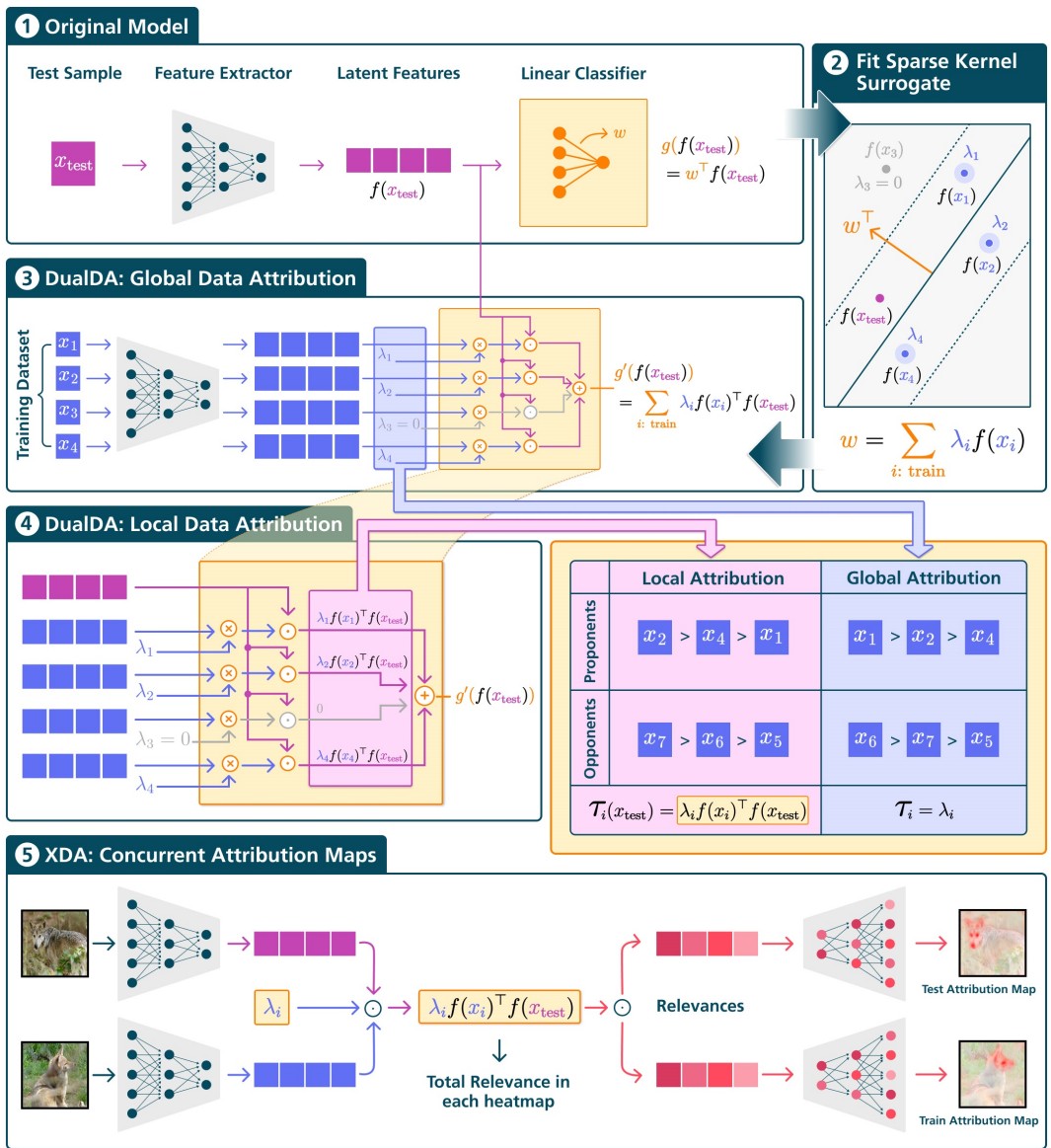

Figure 1: DualDA efficiently identifies training samples which are influential for both the overall model fit (**global attribution**) as well as for the prediction for specific test samples (**local attribution**). ① Our method assumes models with a nonlinear **feature extractor** $f$ followed by a fully-connected layer as the **classification head** $g$. ② DualDA substitutes the final layer of the original model with a **linear SVM**. The resulting weight vector $w$ can then be expressed as a linear combination of the final layer latent embeddings of training samples. Note that a binary classification case is visualized for the sake of simplicity and legibility, whereas DualDA employs a multiclass SVM. ③ The **global attribution** of each training datapoint is quantified by its corresponding scalar coefficient $\lambda_i$ in the linear decomposition of $w$. ④ Moreover, since $w$ is represented as a combination of training feature embeddings, we can decompose the output of the surrogate model for a given test point into a sum of contributions from each training sample. This **local attribution** (i.e. the contribution of a training point to the prediction for a specific test point) is given by the inner product of the feature embeddings of the training and test samples, scaled by the global influence coefficient of the training sample. ⑤ To trace these influences back to the input space, our XDA approach employs Layer-wise Relevance Propagation (LRP) on DualDA attributions. The method propagates the attributions from the surrogate model's output, through the feature extractor, down to the input pixels for both training and test samples. The result is a pair of attribution heatmaps – one for each training–test pair – highlighting input regions that contributed positively or negatively to the model's inference.

approximations, which involve gradients of the training and test sample. The earliest method, Influence Functions (Koh & Liang, 2017), calculates a first-order approximation using the parameters of the final trained model. As this involves calculations of the inverse Hessian, which are infeasible to compute in limited time for even modestly-sized models, approximations using the LiSSA algorithm (Agarwal et al., 2017), the Arnoldi method (Schioppa et al., 2022) or the (Eigenvalue-corrected) Kronecker-Factored Approximate Curvature method (EK-FAC) (Grosse et al., 2023) are commonly used. Furthermore, the method TRAK (Park et al., 2023) applied to a single model can be seen as a version of Influence Functions with a general Gauss-Newton approximation of the Hessian. To include the behaviour of the model over the entire training process, the method TracIn (Pruthi et al., 2020) collects the dot product between the gradients at different epochs and accumulates them in a weighted manner in relation to the corresponding step sizes. GradDot and GradCos (Charpiat et al., 2019) are simplifications of TracIn which only consider the (normalised) dot product at the final epoch.

**Surrogate-based approaches** propose exchanging parts of the network for similarly behaving surrogates which are inherently interpretable in terms of the importance of training data points for predictions. The Representer Points (Yeh et al., 2018) method retrains only the final fully-connected layer of the model using a modified loss with an added $\ell_2$-penalty. This enables the application of the Representer Theorem to decompose the networks output into the sum of kernel evaluations involving the test point and the individual training points.

**Retraining-based approaches** sample counterfactual models by retraining the model on various altered training sets and then use these samples to estimate the model output for an arbitrary training subset. Consequently, these methods are generally regarded as baselines and are very demanding in terms of computational resources (Park et al., 2023).

## 3  Dual eXplainable Data Attribution

In this section, we introduce our two novel methods: **DualDA**, which efficiently produces sparse and faithful data attribution, and **XDA**, which explains the attribution values from DualDA by highlighting relevant features in the input space responsible for the attribution between pairs of training samples and the test input. Combined, DualDA and XDA constitute the **DualXDA** framework.

DualDA is a surrogate-based DA method, which replaces the final linear layer of the model with a linear multiclass SVM. We selected SVMs as surrogates for four reasons: (1) they closely approximate the behavior of the original model, (2) are inherently interpretable in a DA sense in their dual form, (3) yield sparse attributions, (4) and enable fast training and inference, as further outlined below.

(1) Prior research shows that gradient descent in various models has an implicit bias towards margin maximizing behaviour (Gunasekar et al., 2018; Soudry et al., 2018; Gidel et al., 2019; Ji & Telgarsky, 2019; Tarzanagh et al., 2023) which is the defining feature of SVMs. In particular, recent studies demonstrate convergence of neural networks to kernelized SVMs under certain conditions in binary and multiclass classification scenarios (Lyu & Li, 2019; Ravi et al., 2024). This provides the theoretical motivation to assume that an SVM can serve as a faithful surrogate for the original final layer of a neural network predictor. We validate this assumption empirically in Section 4.5.

(2) The learned weights of an SVM in feature space can be exactly expressed as a weighted sum of representations of the training datapoints by solving the corresponding dual problem. This provides a natural approach to global DA, and in extension to attributing the neural network outputs for individual test points.

(3) When working with large datasets containing millions of samples, attribution methods that yield similar or indistinguishable attributions across a large number of data points become difficult to interpret effectively (Iwata & Yoshikawa, 2021). SVMs exhibit an implicit bias towards sparsity in their attribution values as, under constrained conditions, only a limited number of support vectors contribute in determining the model's decision boundary. While this sparsity can enhance interpretability by focusing on the most influential samples, excessive sparsity may hinder the expressiveness of the resulting explanations. To mitigate this, the degree of sparsity can be controlled through the choice of the regularization hyperparameter of the underlying

SVM, allowing a balance between conciseness and representational richness in the attributions, as shown in detail in Appendix F.3.

(4) The training of a linear SVM has a worst-case time complexity between $\mathcal{O}(dN)$ and $\mathcal{O}(dN^2)$ (Hush et al., 2006), where $N$ is the size of the attribution (e.g. training) dataset and $d$ is the size of the input space of the SVM, i.e. the size of the penultimate layer. This allows efficient applicability on large datasets and models. Additionally, computing attributions for a single pair of train and test samples requires only an inner product. The computational complexity per local attribution score is therefore only $\mathcal{O}(d)$. This enables near-instantaneous attribution calculation for new test samples after caching hidden layer activations of the attribution dataset and SVM weights.

To explain the attribution values of DualDA, XDA maps calculate attribution values back through the model to input space, identifying relevant input features (e.g., pixels in images or tokens in text) for the model inferences. XDA simultaneously creates relevance mappings for both training and test samples, producing corresponding relevance heatmaps specific to each pair of samples that discover the relationship between them. This approach renders our DualXDA framework inherently explainable by elucidating the interaction between training and test samples. Moreover, it enhances feature attribution analysis, i.e., the process of tracing model behaviour to input features, by decomposing feature explanations into the constituent effects of individual training samples.

A further extension of our approach including explanations using human understandable concepts to fully unify explanations on the data-level, feature-level and concept-level ( inspired by (Achtibat et al., 2023) ) is given in Appendix H.

### 3.1 DualDA: Using Support Vector Machines for Sparse and Efficient Data Attribution

This subsection presents a step-by-step derivation of DualDA attributions, with a graphical overview provided in Figure 1. DualDA uses the soft-margin multiclass SVM formulation introduced by Crammer and Singer (Crammer & Singer, 2001) as a surrogate model. The contribution of each training sample to the output of this surrogate model can be determined in a straightforward way, which we examine in detail below.

We start by fitting a weight matrix $W$, consisting of one weight vector $w_c^\top$ per row for every class $c$ predicted by the original model, where each weight vector may optionally incorporate a scalar bias parameter (see Steps ① to ② in Figure 1). We learn these parameters to classify data points in the kernel space induced by the feature extractor $f(\cdot; \vartheta)$ (cf. Step ① in Figure 1). This is achieved by solving the (primal) optimization problem

$$
\begin{aligned}
\min_W \quad & \tfrac{1}{2}||W||_2^2 + C \sum_{i=1}^N \xi_i \\
\text{s.t.} \quad & \forall i \in [N],\ \forall c \in [K] \\
& w_{y_i}^\top f(x_i; \vartheta) - w_c^\top f(x_i; \vartheta) + \delta_{cy_i} \geq 1 - \xi_i\ .
\end{aligned}
\tag{1}
$$

Here, the hyperparameter $C > 0$ regulates the sparsity of the solution: A high value for $C$ will produce a solution where $W$ depends only on few training points, whereas a low value for $C$ creates a solution to which many datapoints contribute. The surrogate model output is then given by $Wf(x_{\text{test}}; \vartheta) \approx \Phi(x_{\text{test}}; \theta)$. This programme aims to ensure that for a vector $f_i = f(x_i; \vartheta)$ the class score $w_{y_i}^\top f_i$ corresponding to the true class $y_i$ is at least greater by 1 than the score for all other classes $c$. If this difference is smaller than one, it will contribute to the loss. Therefore, this problem is equivalent to $\ell_2$-regularized empirical risk minimization with a multiclass Hinge loss[2]. Due to the formulation of SVMs through margin maximization, the same problem can be seen as a margin maximization problem for a multiclass classifier. In this formulation, the hyperparameter $C$ determines how hard the margins are, i.e. how strongly each datapoint contributes to the final loss: A higher value increases the penalty for classification errors, encouraging the model to reduce training mistakes at the risk of overfitting.

As per standard SVM theory, we can formulate the dual problem. The full mathematical formulation of the dual problem is excessive in length and not easily interpretable, we therefore refer the interested reader to

---

[2]This formulation differs from training multiple SVMs as it is common for multiclass classification problems, since here, information from different classification heads is shared through the constraints.

Appendix A, where the detailed derivation is given. However, through solving the dual problem, we obtain dual variables $\alpha_{ic}$ for each training vector $f_i$ and each class $c$. These relate to the optimal weight vectors $w_c$ through the formula

$$w_c = \sum_{i=1}^{N} \lambda_{ic} f_i, \tag{2}$$

where the coefficients $\lambda_{ic}$ defined as

$$\lambda_{ic} = \begin{cases} (C - \alpha_{iy_i}) & \text{if } y_i = c \\ -\alpha_{ic} & \text{else} \end{cases} \tag{3}$$

express how much each training feature vector $f_i$ contributes to the surrogate fit and can therefore be interpreted as global attribution scores (see Step ③ in Figure 1). Through this formula, we can further identify the individual contributions of each $f_i$ to the predicted class score of a test vector $f_{\text{test}}$ for some class $c$:

$$w_c^\top f_{\text{test}} = \sum_{i=1}^{N} \lambda_{ic} f_i^\top f_{\text{test}} = \sum_{i:y_i=c} (C - \alpha_{iy_i}) f_i^\top f_{\text{test}} - \sum_{i:y_i \neq c} \alpha_{ic} f_i^\top f_{\text{test}} \tag{4}$$

Given this motivation, DualDA defines the attribution value of a training sample $x_i$, given the prediction of a test sample $x_{\text{test}}$ and class $c$, as

$$\tau_c^{\text{DD}}(x_{\text{test}}, i) = \lambda_{ic} f_i^\top f_{\text{test}}, \tag{5}$$

where DD constitutes the abbreviation of DualDA. This local attribution is depicted in Step ④ in Figure 1. We refer the reader to Appendix A for the full derivation. By this definition, DualDA fulfills an approximate **conservation property**, meaning that the attributions over all training samples sum up to the output of the model, i.e.

$$\Phi(x_{\text{test}})_c \approx w_c^\top f_{\text{test}} = \sum_i \tau_c^{\text{DD}}(x_{\text{test}}, i). \tag{6}$$

**Relating DualDA to other approaches** While we introduce DualDA with a focus on SVM theory, it can be interpreted within the framework of the Representer Points method as an $\ell_2$-regularized linear surrogate fitted on top of the extracted features in the penultimate layer. The methods differ in the loss used to train the surrogate. Whereas Representer Points keeps the original loss and simply adds an additional $\ell_2$-regularization, DualDA attributions can be derived by using a multiclass Hinge loss in the formula for Representer Points in Equation (D.6). This derivation of DualDA is detailed in Appendix B.

Furthermore, we can align our surrogate-based approach with the framework of gradient-based approaches. Applying a generalized version of Influence Functions outlined in Corollary 1 of (Naujoks et al., 2024), we can investigate how the optimally derived $W$ in Equation (1) changes, under an infinitesimal downweighting of the contribution of datapoint $f_i$ and how this in turn affects the class score $w_j^\top f_{\text{test}}$. This influence value aligns with the attribution calculated by DualDA, for datapoints that do not lie exactly on the margin. We report this result in Theorem 3.1 and refer the reader to Appendix C for the proof.

**Theorem 3.1.** *Let $W$ denote the solution to the SVM optimization problem in Equation (1). Denote by $W^i$ the parameters of an SVM trained on the same dataset and a optimization criterion modified w.r.t. to the $i$-th training sample:*

$$\begin{aligned} \min_W \quad & \tfrac{1}{2}\|W\|_2^2 + C\sum_{j=1}^N \xi_j - \varepsilon C \xi_i \\ s.t. \quad & \forall j \in [N], \ \forall c \in [K] \\ & w_{y_j}^\top f(x_j; \vartheta) - w_c^\top f(x_j; \vartheta) + \delta_{cy_j} \geq 1 - \xi_j, \end{aligned} \tag{7}$$

*The contribution of the $i$-th training sample is down-weighted by a factor of $\varepsilon$. Then, for any test sample $x$ the infinitesimal change in $Wf(x; \vartheta)$ is equal to the DualDA attributions, if the $i$-th training sample does not lie exactly on the margin of the SVM, i.e. $\arg\max_c\{1 - [\delta_{cy_i} + w_{y_i}^\top f_i - w_c^\top f_i]\}$ is unique. In this case, the following equality holds:*

$$\tau_c^{\text{DD}}(x, i) = \left[ \frac{\partial (W - W^i)^\top x}{\partial \varepsilon} \bigg|_{\varepsilon=0} \right]_c \tag{8}$$

### 3.2 XDA: Mapping Data Attribution to the Feature Space

While DualDA constitutes scalable and practically applicable solution for DA, we argue that DA alone is not sufficiently informative on its own: DA informs about which training datapoints are influential for a prediction, yet not *why* they are influential: A single datapoint may be relevant for various reasons, e.g., due to background patterns, parts of objects, or other semantic features. DA, however, does not provide any information to distinguish between these sources of information and their influence, resulting in potentially ambiguous explanations. To resolve this lack of clarity, we combine data attribution with feature attribution, leveraging the structure of our DualDA attributions with the introduction of XDA in order to obtain additional degrees of detail in our combined DualXDA explanations. This integrated perspective allows for deeper, more interpretable explanations of model behavior. While in principle, any FA approach could be levied to further explain the attributions of DualDA we choose to integrate the principles of Layer-wise Relevance Propagation (LRP) (Bach et al., 2015) to achieve XDA because like DualDA, it leverages the model's activations on its hidden layers. LRP is a feature attribution technique based on the decomposition of a neural network's prediction by backward propagating relevance scores layer-by-layer from the output to the input, following a relevance conservation principle that maintains and preserves the total amount of relevance throughout the decomposition process. As such, LRP is effectively striving towards the same goal as DualDA: the conservative attribution of model outputs to units of interpretation. However, the crucial difference between the two approaches lies in the orthogonal selection of the components targeted for interpretation, i.e., individual test features for LRP and training samples in their totality for DualDA. In the image domain, explanations derived from LRP are commonly represented as a heatmap over the corresponding input.

XDA combines the approaches of DualDA and LRP to further explain the data attribution scores $\tau_c^{\mathrm{DD}}(x_{\mathrm{test}}, i)$ w.r.t. the interaction of both $x_{\mathrm{test}}$ and the influential training datapoint $x_i$. An overview of the XDA relevance computation process is presented in Step ⑤ of Figure 1. Specifically, the DualDA surrogate model naturally produces the data attributions as an intermediate signal, each of which is given by a weighted dot product of features computed by the feature extractor part of the network. XDA focuses on this branch of the surrogate model, whereby the output of this branch is isolated and explained via LRP with initial relevance determined depending on class $c$ explained by DualDA as

$$R(x_{\mathrm{test}}, i)_c^L = \lambda_{ic} \cdot f_{\mathrm{test}}^{\top} f_i \ . \tag{9}$$

The algorithm then backpropagates this quantity from layer $L$ of the DualDA surrogate model backwards through the feature extractor towards the input sample $x_{\mathrm{test}}$ as well as the training sample $x_i$ in two separate backward passes. This procedure results in a pair of feature attributions $R^{\mathrm{test}}(x_{\mathrm{test}}, i)_c^l$ and $R^{\mathrm{train}}(x_{\mathrm{test}}, i)_c^l$ which together explain how – in terms of related features – the influential support vector $x_i$ in particular is affecting the model in the prediction of $x_{\mathrm{test}}$[3]. As such, XDA shares conceptual similarities with BiLRP (Eberle et al., 2022) and CRP (Achtibat et al., 2023), but differs by explaining the feature kernel of a surrogate SVM (rather than a deep similarity model, as in BiLRP) and by splitting propagation pathways across support vectors instead of latent concepts (as in CRP). Overall, XDA leverages the conservative properties of DualDA and LRP to achieve data attribution as well as feature attribution, combining the benefits of the two orthogonal approaches. The conservative property of XDA further implies that the generated feature attribution heatmaps are also conservative whereby the sum of XDA test feature attribution maps recovers the LRP feature attribution map, i.e. LRP explanation of $x_{\mathrm{test}}$ for class $c$ at layer $l$ is approximately equal to $\sum_{i=1}^{N} R^{\mathrm{test}}(x_{\mathrm{test}}, i)_c^l$.

## 4 Evaluating Data Attribution

To test the validity of DualDA, we compare the method against other state-of-the-art attribution methods introduced in Section 2 and defined in detail in Appendix D.1. Details about implementations are given in Section 4.3. We analyze three different vision models of varying sizes, each trained on a different dataset: a simple Convolutional Neural Network with 3 convolutional and 3 fully connected layers with ReLU non-linearities trained on the MNIST dataset (LeCun et al., 2010), ResNet-18 (He et al., 2016) trained

---

[3]Note that while we can derive such explanations in all layers $l \le L$, we will focus on input-layer explanations in pixel space with $l = 1$ in this paper.

on CIFAR-10 (Krizhevsky & Hinton, 2009) and ResNet-50 (He et al., 2016) trained on the Animals with Attributes 2 (AwA2) dataset (Xian et al., 2018) (training details can be found in Appendix D.2). For each dataset and model, we calculate the attribution score of all training datapoints for the predicted class for 2,000 randomly selected test points, using the attribution methods. We evaluate the quality of the derived attributions on seven different metrics outlined in Section 4.1, two of which are novel contributions. For detailed explanations of evaluation metrics, including formulas, we refer the reader to Appendix E. In our study, we rely on `quanda` (Bareeva et al., 2024), a toolkit designed for the systematic evaluation of DA methods. The results of our experiments are presented in Section 4.2. We further present the resource consumption of each attribution method in Section 4.3 and compare the sparsity of the corresponding explanations in Section 4.4. In Section 4.5, we analyze how close the surrogate model matches the original model for our DualDA method, as well as the Representer points method. All experiments have been executed on a single NVIDIA A100 Tensor Core-GPU.

## 4.1 Evaluation Metrics

Two simple sanity checks are proposed by Hanawa et al. (Hanawa et al., 2021): The **Identical Class Test** relies on the assumption that the most positively important training points for a class decision should be of the same class as the model prediction. We therefore measure for all test points the proportion of the highest ranking training data points according the DA methods outputs belonging to the same class as predicted for the given test sample. The **Identical Subclass Test** further assesses whether attributions can distinguish between different subpopulations within the same class. Following (Hanawa et al., 2021), we therefore artificially group dissimilar classes together into a superclass. Given a predictor trained on this new classification setting, we check whether the DA methods can still attribute influence to training samples from the same true subclass, rather than just the broader grouped class set, as the test sample.

Furthermore, two metrics are derived from common downstream tasks of DA: **Mislabeling Detection** (Koh & Liang, 2017; Yeh et al., 2018; Pruthi et al., 2020), which aims to discover incorrectly labeled training samples, and **Shortcut Detection** (Koh & Liang, 2017; Hammoudeh & Lowd, 2022), which tries to identify misleading associations in training data that allow models to achieve accuracy through dataset-specific biases rather than generalizable relationships.

Finally, retraining-based metrics test the counterfactual validity of training data attributions by retraining models on subsets of the training data and measuring whether the attributions align with observed changes in model behaviour. The **Linear Datamodeling Score (LDS)** (Park et al., 2023) tests whether attribution scores can predict changes in model outputs when retrained on random subsets of data. This is achieved by measuring the rank correlation between predicted attribution scores and actual model outputs. **Coreset Selection** evaluates whether attributions can identify the most valuable training samples by retraining models only on the highest-attributed data points and measuring how well performance is maintained. **Adversarial Data Pruning** tests the opposite of coreset selection by removing the most highly attributed training samples and measuring how much performance degrades. Unlike Coreset Selection, which evaluates whether attributions can identify a representative training subset, Adversarial Data Pruning assesses whether attributions correctly identify samples that contain unique information.

For all metrics except Coreset Selection, where superior performance is indicated by a small loss, higher scores indicate higher attribution quality.

## 4.2 Results

Figure 2 provides a comprehensive summary of our experimental findings. For each method and dataset, it displays the average rank across all evaluation metrics plotted against the total runtime required to attribute 2,000 test images. This analysis demonstrates that DualDA exhibits competitive performance relative to other methods: it achieves the fourth-best average rank on MNIST, second-best on CIFAR-10, and the best average rank on AwA2. Crucially, DualDA invariably requires the least computational time among all evaluated methods, which includes both the cache time required for generating explanations (such as caching hidden activations) as well as the explanation time for all 2,000 test samples. Compared to the most efficient version of the widely-employed Influence Functions framework, DualDA is 175 times faster on MNIST, 383

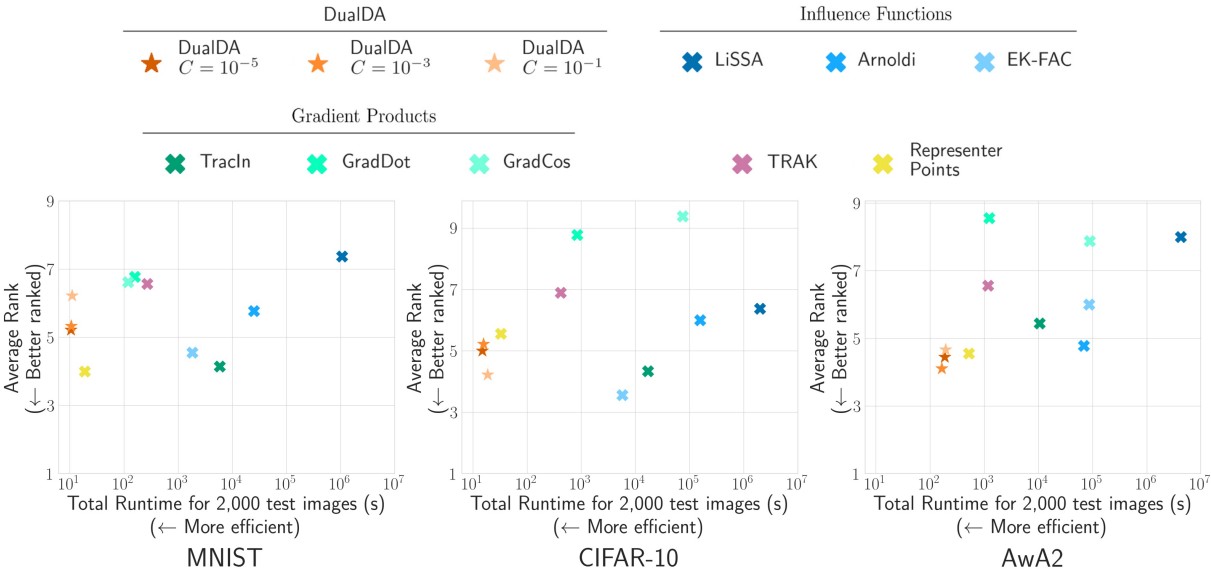

Figure 2: Average rank across all metrics plotted against the total runtime for calculating the attribution for 2,000 test images. DualDA demonstrates competitive performance with drastically reduced computational time requirements across datasets. The figure showcases the results for nine different methods, averaged over seven different metrics, on three different datasets and models.

times faster on CIFAR-10, and 520 times faster on AwA2. For explanation time only, this speedup increases to 4,000× on MNIST, 11,000× on CIFAR-10, and 8,000× on AwA2.

DualDA, with appropriately selected hyperparameter $C$ for each metric, ranks among the top three performing methods for 6 metrics on MNIST, all 9 metrics on CIFAR-10, and 8 metrics on AwA2. Furthermore, it achieves optimal performance on 5 out of 9 metrics on AwA2.

For a more detailed discussion, we present the scores of all metrics and methods for the AwA2 dataset in Figure 3 (results for the MNIST dataset and for the CIFAR-10 dataset can be found in Figure F.1 and Figure F.2 in Appendix F.1): On the Identical Class test, DualDA achieves a perfect result, while most methods except Representer Points, Arnoldi and TracIn achieve a score below 0.2. On the Identical Subclass test, no method achieves a score above 0.8, but DualDA with a low sparsity performs comparatively well and achieves the second-best score. On Mislabeling Detection, DualDA demonstrates excellent performance. Here, most methods perform reasonably well, except for Representer Points. We do not show results for LiSSA, which can not be computed in feasible time, and GradCos, which does return inconclusive attributions in this setting by design (c.f. Figure 3 and Appendix E). On Shortcut Detection, DualDA achieves a perfect score while the conceptually related Representer Points only attains a score of 0.2 and all other methods fail almost completely. Notably, despite not being explicitly designed to estimate outputs of counterfactually trained models, DualDA also performs strongly on the retraining-based metrics. On LDS, it achieves the third-highest rank. On Coreset Selection, it is the best performing method both at 10% and in the combined average. At the 10% data pruning level, DualDA does not rank among the top three performing methods. Nevertheless, the performance differences between all evaluated approaches remain relatively modest, highlighting the inherent difficulty of eliminating unique information when pruning only a small fraction of the dataset. When considering the weighted average across all pruning levels, DualDA demonstrates competitive performance, securing the second-highest score and substantially outperforming the majority of alternative methods.

From the experiments, we can also see how the choice of the sparsity hyperparameter $C$ affects the performance of DualDA on various metrics. This is particularly pronounced on Shortcut Detection, where a small sparsity is preferable, likely because it is preferable to have many training samples included in the support vectors to find all spuriously correlated samples. Conversely for Coreset Selection, higher sparsity yields markedly better

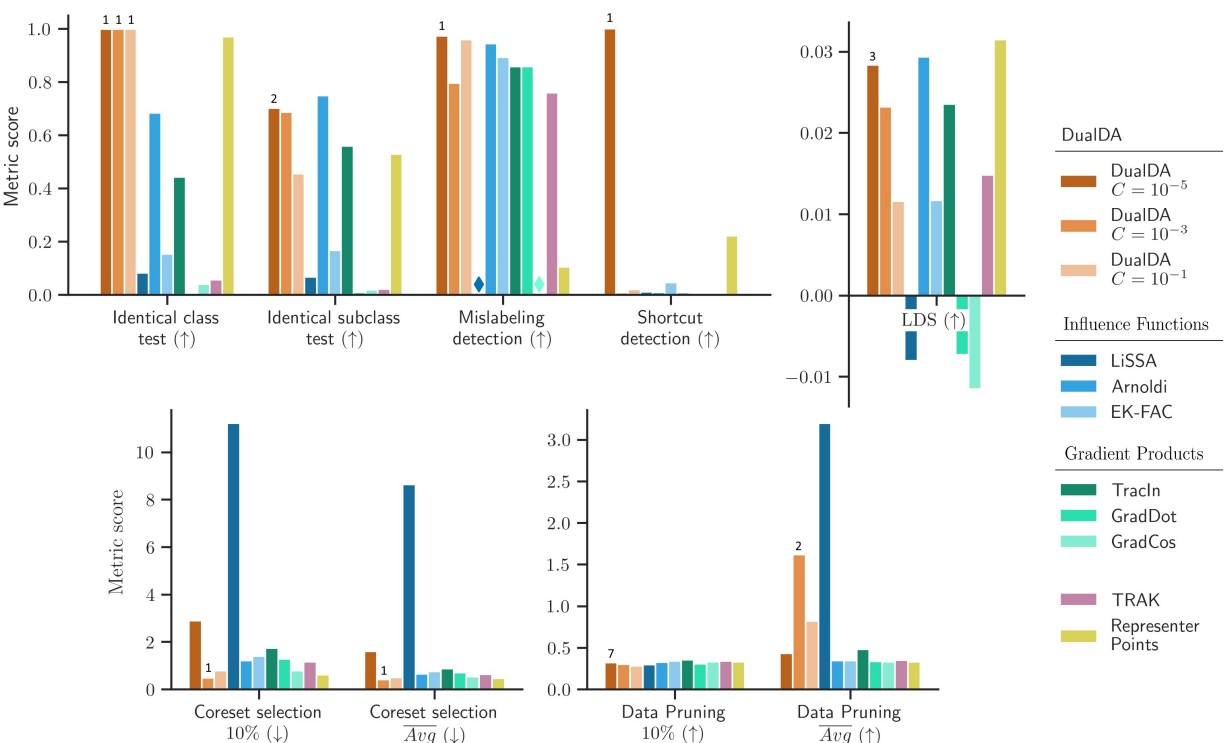

Figure 3: Evaluation results on the AwA2 dataset. The rank for DualDA with the best-performing hyperparameter $C$ is denoted over the corresponding bar. Note that Mislabeling Detection requires calculating the self-influence of the entire train set (see Appendix E). For LiSSA, calculating self-attributions for all training points would take roughly a year of runtime and is therefore computationally infeasible. As GradCos is defined as the cosine of the angle between the test and training sample's feature vectors, the self-attribution for GradCos is equal to 1 regardless of the sample.

results, plausibly because a representative subsample of the data is selected in the sparsification process. We extensively analyze the effect of the sparsity hyperparameter over a wide numerical range in Appendix F.3. Based on these results, we recommend a default setting of $C = 10^{-3}$ for image classification tasks with convolutional neural networks, which corresponds to the primary experimental configuration considered in this work.

### 4.3 Resource Consumption

Computation time and memory requirements are the main concerns for scaling DA approaches to big models and datasets. Early approaches such as LiSSA (Koh & Liang, 2017) require $\mathcal{O}(Np)$ time to attribute a new test point, where $p$ denotes the number of model parameters. This is indeed much slower than the explanation time of DualDA, which requires only $\mathcal{O}(d)$ time, as $d$ is the size of the penultimate layer of the network and therefore $d \ll p$ (for the ResNet-50 model, we have $d = 2{,}048$ and $p = 25{,}557{,}032$). Other approaches such as Arnoldi (Schioppa et al., 2022), EK-FAC (Grosse et al., 2023) and TRAK (Park et al., 2023) provide efficient approximations, which come at the cost of precomputing and and creating an information cache. In Figure 4, we report caching times, cache sizes and local explanations times (excluding caching times) for evaluations on 2,000 datapoints per dataset. The total runtime on a single GPU per method and dataset, which is the sum of caching and explanation time, is used as the $x$-coordinate in each plot of Figure 2. For TRAK, Representer Points and EK-FAC, we rely on official code releases of the original publications. For LiSSA, we use the implementation from a different publication (Bae et al., 2022) as it is written in the `PyTorch` (Paszke

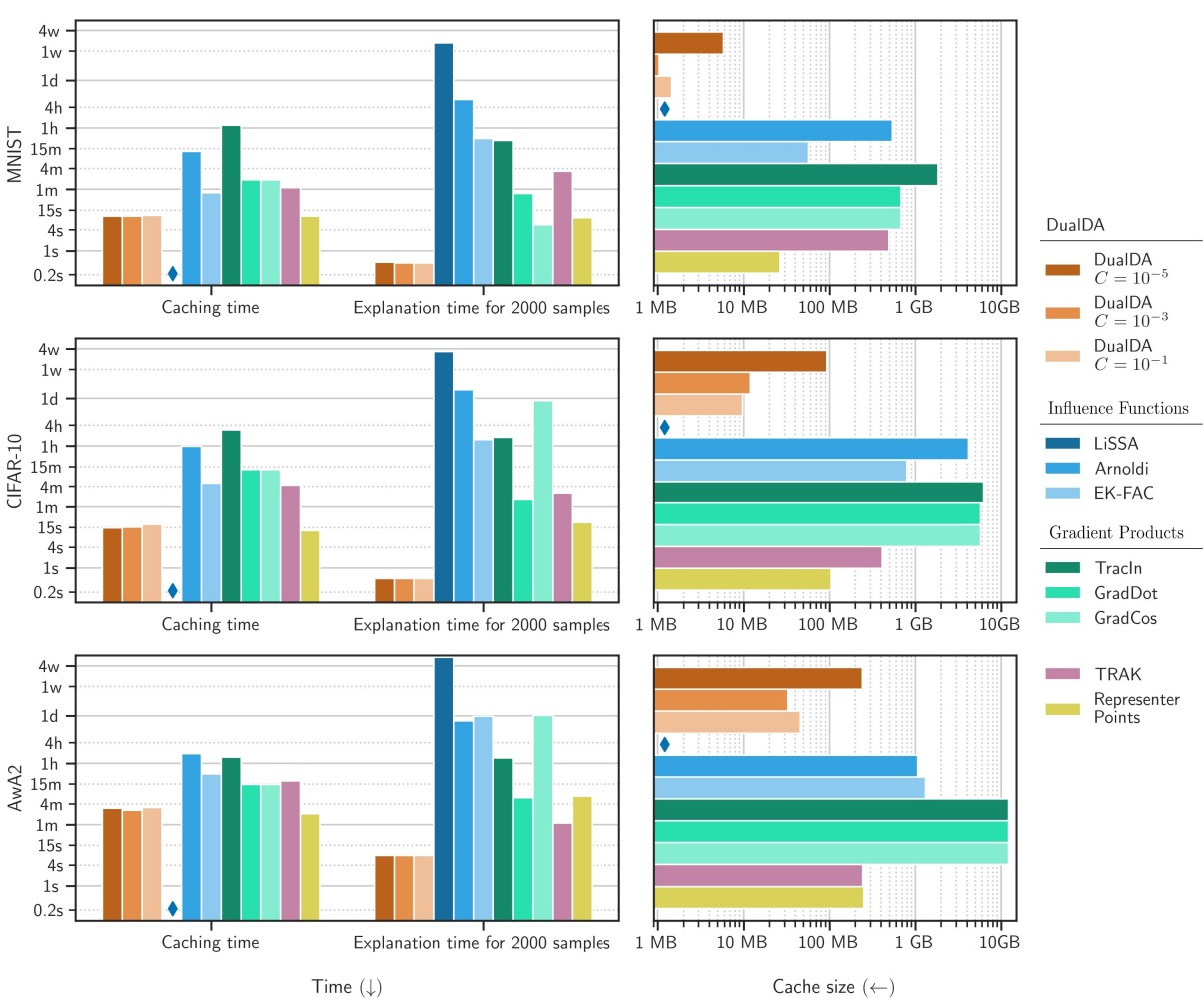

Figure 4: Caching times and cumulative explanation times over 2,000 test samples for all methods, mapped onto a logarithmic scale, as well as sizes of precomputed cache information per method. Note that LiSSA does not require any caching and thus has no caching time or cache size. As calculations are made on GPU, the sparsity level of DualDA has only very minor impact on the runtime.

et al., 2019) framework. Employing the same framework universally guarantees a just comparison between all methods. We have implemented GradCos, GradDot and TracIn ourselves, using the random projection trick explained in (Pruthi et al., 2020). For Arnoldi, we relied on the `captum` (Kokhlikyan et al., 2020) implementation. To implement DualDA in Python, we have modified[4] the multiclass SVM implementation from the Python package `scikit-learn` (Pedregosa et al., 2011).

**Cache Size** TracIn, GradDot and GradCos save the gradients of the fully trained model, and TracIn additionally saves gradients at different training epochs. To make cache requirements manageable, these methods further require storing a random projection matrix mapping model gradients to a subspace of manageable dimensionality $D$. This method can bring about memory-related challenges with larger models, as it requires storing a random vector of size $D$ for each model parameter. The Arnoldi and EK-FAC implementations of Influence Functions as well as TRAK also require a large cache. All of these methods rely on the introduction of random sampling and projections to ameliorate cache space and runtime requirements at the cost of approximation inaccuracies. Note that for TRAK, we use the GPU-level C++ implementation provided by the authors. Reported results concerning TRAK do not include any cache for random projection matrices, and the reported runtimes are obtained using a setup with maximal efficiency.

In comparison, Representer Points only saves the extracted latent features of the training data and the surrogate weights. DualDA further reduces memory requirements by storing only the extracted latent features for training samples that serve as support vectors. This substantially reduces cache size at higher sparsity levels: compared to the most cache-intensive method, TracIn, DualDA reduces the required cache by a factor of 1,750 on MNIST, 636 on CIFAR-10, and 370 on AwA2.

**Caching Time** TracIn requires the computation of gradients for model checkpoints, whereas Arnoldi, TRAK and EK-FAC require the computation of Hessian approximations and in some cases, projected model gradients. These computations take considerable time, compared to the surrogate based methods DualDA and Representer Points. Caching times for competing methods range from 5 times to 735 times longer than the kernel surrogate approaches.

**Explanation Time** To a substantial degree the slowest method at inference is LiSSA. Even though we have only included the final fully-connected layer in the Hessian and gradient computations, LiSSA takes 35 minutes for the explanation of a single test point of the AwA2 dataset, requiring almost 7 weeks for explaining the entire test set of 2,000 datapoints. This is the minimal runtime that is achieved by following the authors' suggestion (Koh & Liang, 2017) that the total number of iterations dedicated to the estimation of the inverse-Hessian-vector-product of Equation (D.1) be greater than the training dataset size. Arnoldi, TracIn and GradCos are moderately slow, requiring more than one second per test point on the AwA2 dataset. DualDA is the fastest method overall across all datasets, only requiring between 0.2ms and 4ms per test sample. Compared to the slowest method, LiSSA, DualDA is $2.4 \times 10^6$ times faster on MNIST, $4.1 \times 10^6$ times faster on CIFAR-10, and $5.4 \times 10^5$ times faster on AwA2. Compared to the most efficient Influence Functions method, DualDA provides a speedup of $4{,}000 - 11{,}000\times$, depending on the dataset. When generating explanations using a GPU, higher sparsity does not significantly decrease explanation time since the required matrix multiplication is already effectively parallelized and other parts of the code dominate the runtime. In our experiments, we observed, however, that when generating explanations on CPU, high sparsity significantly improves the computation time for explanations.

### 4.4 Sparsity of Attributions

We analyze the distribution of attribution scores over the training set for various DA methods, exemplified on the AwA2 dataset. Comparable patterns emerge across all datasets examined in this study. From a practical standpoint, sparsity can be quantified by determining the number of training points required to account for a given percentage of total attribution for a specific test sample. In Figure 5, we evaluate the sparsity characteristics of each method by analyzing how the cumulative sum of absolute attributions grows as a function of the fraction of most influential datapoints included. Training samples are incorporated in descending order of absolute attribution value. Methods exhibiting sparse attribution patterns demonstrate steep initial increases in this cumulative curve, while non-sparse methods show more gradual growth. Our

---

[4]Code will be made available upon acceptance.

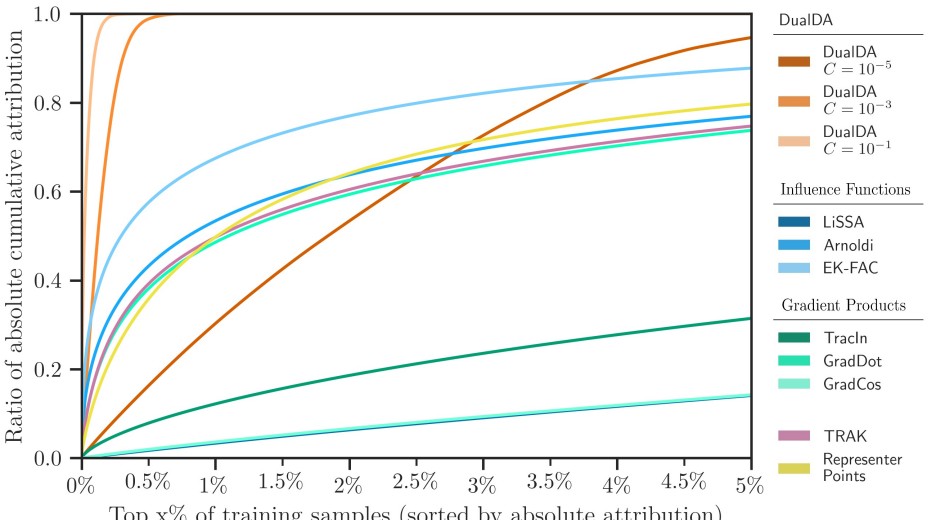

Figure 5: Analysis of cumulative distribution of positive and negative attributions for various DA methods on the AwA2 dataset. The $x$-axis represents the top x% of training samples when sorted by their absolute attribution scores, while the $y$-axis shows what fraction of the total absolute attribution is contained within those top samples. The cumulative curves are obtained by computing individual curves for each test sample and subsequently averaging across the complete test set.

findings reveal that the sparsity hyperparameter $C$ of DualDA effectively controls the sparsity of the resulting explanations. With $C = 10^{-1}$, the method reaches full attribution saturation when selecting only 0.2% of the training samples descendingly ranked by absolute attribution. EK-FAC exhibits faster accumulation than Arnoldi, TRAK, GradDot, and Representer Points. Although these latter methods initially surpass DualDA configured with low sparsity ($C = 10^{-5}$), their accumulation rate decreases more rapidly, enabling DualDA with low sparsity to still reach 90% cumulative absolute attribution quicker. TracIn, GradCos, and LiSSA demonstrate minimal sparsity, distributing similar absolute attribution values across numerous training samples.

## 4.5 Faithfulness of Surrogates

When using a surrogate model to obtain explanations, we must ensure that the surrogate in question closely represents the original model. To test this empirically, we observe three model similarity metrics for the surrogate models Representer Points and DualDA across all three datasets: For the original and the surrogate model, we record the cosine similarity between their last layer weight matrix (which are the only weights which we change in the model), the average cosine similarity of the output logits, and the multiclass Matthews correlation coefficient (MCC) of the predictions. All three metrics range from -1 to +1, where higher scores indicate greater similarity. The monochrome bars in Figure 6 present the results for DualDA and Representer Points on the AwA2 dataset, results for the remaining two datasets are shown in Figure F.3 in Appendix F.2. The overall faithfulness of the surrogates to the original model are high across methods, metrics and datasets. The similarity scores are overall comparable for DualDA and Representer Points. Interestingly, an increase in the hyperparameter $C$ can either lead to an increase or decrease in the scores, depending on the similarity metrics and dataset. However, most metrics seem to be highest at $C = 10^{-3}$, indicating that there exists an optimum which maximizes the similarity of the surrogate to the original model.

**Artificially Sparsifying Representer Points** Following the assumption that sparse attributions are preferable due to their lower complexity (Colin et al., 2022), it is only natural to decide to take a non-sparse attribution method and subsequently sparsify its results. As Representer Points is a method similar in spirit to DualDA, achieving promising results in attribution quality, we test the feasibility of this approach by post-hoc

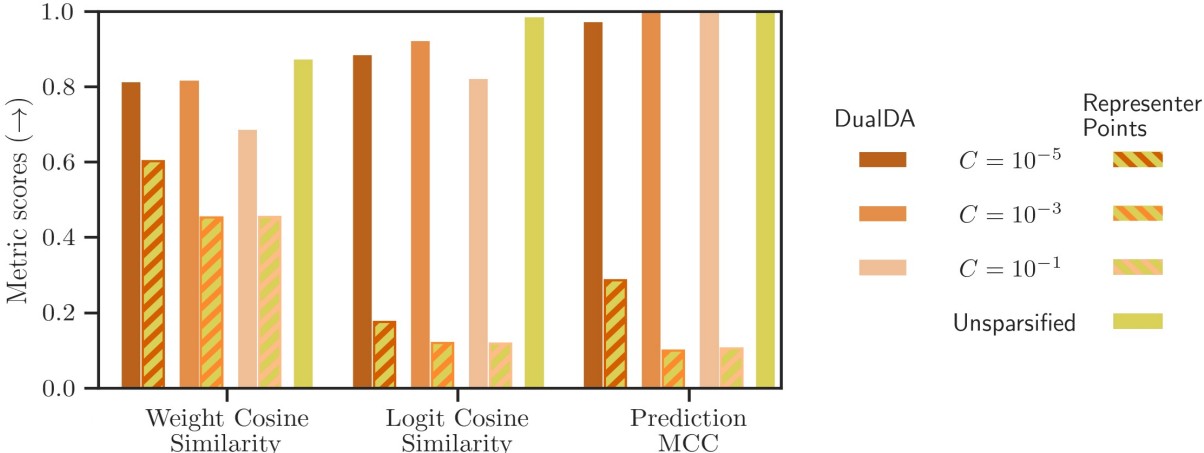

Figure 6: We test surrogate DA faithfulness metrics for DualDA with sparsity hyperparameter $C \in \{10^{-5}, 10^{-3}, 10^{-1}\}$ and Representer Points on the AwA2 dataset. The results are shown in the monochrome bars. In order to assess whether post-hoc artificially sparsified Representer Points surrogates will lead to faithful explainers, we induce the same sparsity levels as they naturally emerge for DualDA with the corresponding sparsity parameter $C$, and present the corresponding surrogate faithfulness scores via striped bars.

and artificially sparsifying the Representer Points surrogate model. We first sort the absolute coefficients $|\lambda_{ic}|$ for all classes $c$ and datapoints $z_i$. Afterwards, we set all coefficients except for those with highest absolute value to zero. The number of non-zero coefficients for each class is set to the number of support vectors defining a trained DualDA surrogate for $C \in \{10^{-1}, 10^{-3}, 10^{-5}\}$ (an overview over the number of support vectors for each class and dataset can be found in Figure F.7 in Appendix F).

The striped bars in Figure 6 present the results for these artificially sparsified Representer Points surrogates for the AwA2 dataset and in Figure F.3 in Appendix F.2 for the remaining datasets. For better readability, the striped bars are placed next to the bar corresponding to the DualDA setting at the same sparsity level, i.e. we maintain the same number of support vectors per class for Representer Points as in the corresponding DualDA configuration. The results indicate that the quality of the Representer Points surrogate decreases rapidly as the sparsification intensifies. The decrease in faithfulness metrics is less severe in CIFAR-10, however for the other datasets, we see that already the first sparsity level corresponding to $C = 10^{-5}$ has low surrogate faithfulness. This indicates that in order to achieve sparse data attribution, it does not suffice to choose an attribution method and post-hoc sparsify artificially – for optimal results it is required to choose a method that creates inherently sparse attributions.

## 5 XDA Experiments

XDA provides unprecedented insight into the interaction between training and test samples by simultaneously highlighting relevant features in both input domains, thereby illuminating the similarities and dissimilarities that drive their relationship. This approach transforms previously opaque attribution values into interpretable explanations, offers a more nuanced understanding of model predictions, and enables feature-level ablation of training or test samples to analyze and modify training dynamics. To demonstrate XDA's capabilities, we apply the approach to a small convolutional network trained on the MNIST dataset, and a VGG-16 network trained on AwA2 and the ImageNet dataset (Deng et al., 2009) respectively. Within the LRP framework, different formulas, or "rules", exist to distribute the relevance from the output back to the input space. For our experiments, we use the recommended rule composite `EpsilonPlusFlat`, as implemented in the `zennit` toolkit (Anders et al., 2021), which applies the `flat` rule for any linear first layer, the `z-plus` rule for all other convolutional layers and the `epsilon` rule for all other fully connected layers. For VGG-16, we additionally

apply the `flat` rule for the first 10 layers for better legibility of the corresponding relevance heatmaps. An explicit description of the corresponding formulas is given in (Montavon et al., 2019).

## 5.1 Qualitative Case Study: Gaining Insight with XDA

Using an exemplary sample from, the AwA2 dataset, we step-by-step guide the reader through the interpretation of an XDA explanation with the help of Figure 7: ① The trained model classifies this image as a wolf. The corresponding LRP heatmap shows which parts of the image were relevant for the decision: Red areas support the model's decision (the ears, the fur, and parts of the background), while the model considers blue areas (the eyes and the snout) contradictory to its classification. Surprisingly, the animal's characteristic facial features are considered as opposing the classification as a wolf. While LRP and other feature attribution methods can show us *where* informative features are picked up by the model for the classification, they cannot explain *why* the model has identified these areas based on the training data. We can answer this question by the characteristic of XDA of splitting the single LRP explanation into individual, training-data-specific attributions (which, when summed up, equate to the LRP explanation again). ② The right part of the middle row depicts the strongest proponents for the classification of the test sample as class wolf, together with their label and corresponding DualDA attribution value. All shown most influential training images depict wolves in a grassy or leafy habitat. ③ The XDA maps of the test sample attribute the corresponding DualDA attribution for each training sample onto the input test image. Note that all of these heatmaps are unique: For the first two training samples, where snouts of the animals are only hardly or not visible, the snout area of the test sample is negatively or only barely positively relevant according to the attribution. This is different for the third and fourth sample, where the eyes and snout are clearly visible, and the corresponding part of the test sample receives high positive attribution scores. Note that for all pairs of test and training image, the image areas depicting the wolves' ears and fur are attributed with positive relevance. ④ The XDA heatmaps computed in the input domain do not only connect DualDA attributions to the test sample features, but also to the features of the influential training samples. We observe that for the third and fourth most positively influential training samples, the snout and eyes are attributed most strongly, indicating a high responsibility of those features for the sample's influence on the prediction. ⑤ The center row, from the midpoint to the left, depicts the strongest opponents against the classification of the test sample as a wolf, identified by the strong negative attribution scores received from DualDA. Note that all samples depict instances of class "fox". ⑥ The XDA test heatmaps indicate that the eyes and snout are responsible for the high negative attributions for these training samples. We recognize from the shown training samples of class "fox", representing influential training data subsets where the depicted animals all share similar facial features as the wolf in the test image, that the model must have associated this characteristic canine facial structure more to class "fox" than to class "wolf", which in turn influences the currently analyzed prediction. ⑦ In the XDA heatmaps of the training samples, we again find that the corresponding eyes and noses are most relevant for the negative attribution. Altogether, XDA explains the a priori unanticipated result from the LRP heatmap in ① which causes the facial area of the wolf to be negatively relevant for the classification as a wolf: While there exist some other wolves in the training set with a similar muzzle, which the model correctly uses to detect these training samples as relevant for the classification of the test sample, the training corpus also contains multiple other canines such as foxes, making the snout an unreliable indicator for wolves in the AwA2 dataset.

Additional examples of DualXDA explanations for samples from the MNIST, AwA2, and Imagenet dataset are presented in Appendix G.

## 5.2 Ablation Study

To ascertain the capability of XDA to explain counterfactual effects, we additionally conduct a controlled ablation study on the MNIST dataset, the results of which are depicted in Figure 8. Figure 8a shows the original test sample together with its five most relevant training samples and the corresponding XDA heatmaps for the test sample. For the first three training samples with the highest relevance, the upper segment of the test sample is positively influential for their relevance. On the other hand, for the last two training samples, which are attributed with lower relevance and do not have a similar arc, the upper segment holds much less relevance. First, we perturb the test sample by removing the base of the digit "2" and keeping only the upper

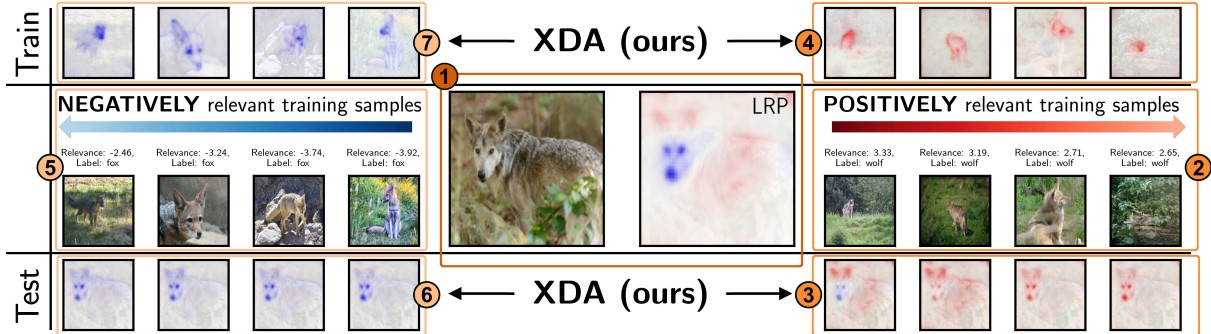

Figure 7: Overview figure showcasing the use of DualXDA on the AwA2 dataset. ① Test sample classified as wolf and corresponding LRP heatmap. ② Strongest proponents with corresponding label and DualDA attribution (attribution reduces in the direction of the arrows). ③ XDA test heatmaps relevance of proponents. ④ XDA training heatmaps relevance of proponents. ⑤ Strongest opponents with corresponding label and DualDA attribution. ⑥ XDA test heatmaps relevance of opponents. ⑦ XDA training heatmaps relevance of opponents.

loop of the test sample. We now recompute DualDA attributions for this perturbed sample and present them with the corresponding XDA heatmaps in Figure 8b, yet keep observing the training samples as previously identified via DualDA for the unperturbed test point. We can observe that for the two rightmost training samples the attribution scores are reduced by approximately 70%, which is proportionally higher than the reduction in attribution scores on the three highest attributed training samples at approximately 60% on average. Furthermore, the ordering of the training samples by attribution is not changed by the ablation.

In a second variant of our experiment, we intervene by removing the upper arc from the test sample and calculate the DualDA attribution for the same five training samples again, as depicted in Figure 8c. Now, the relevance of the first three training samples reduces approximately by 67% on average. In contrast, the relevance for the last two training samples decreases by only approximately 51% and the attribution scores remain at higher levels than in Figure 8b, where the base of the numeral was removed. The training samples which previously have received lower attribution scores are now the second and third most attributed sample respectively. After the ablation, the attribution maps of all five training points are similar, indicating that the increased relevance of the first three training points before ablation can be explained with the presence of the upper arc in the test sample. This is further supported by observations on the XDA heatmaps for the modified test sample, which now appear more similar across different training samples; any extent of positive attribution associated with the arc has vanished.

## 6 Limitations & Future Work

The current formulation of DualXDA employs a multiclass SVM as a surrogate to achieve Data Attribution, which restricts the demonstrated applications to classification settings. However, the framework does not rely on this restriction, and alternative sparse kernel-based models (e.g., Support Vector Regression (Drucker et al., 1996), One-Class SVMs (Schölkopf et al., 2001), Incremental SVMs (Cauwenberghs & Poggio, 2000), or RVMs (Tipping, 2001)) can be used to extend DualXDA to regression, outlier detection, active learning, or provide probabilistic formulations.

Furthermore, the evaluation of DualXDA in the present study is limited to image classification models of moderate scale, and does not yet include a comprehensive quantitative assessment on contemporary large-scale or multimodal foundation models beyond the exploratory analyses provided in Appendix I. The method's reliance on penultimate-layer representations suggests compatibility with other network families, including transformers. Our preliminary observations in Appendix I suggest that DualXDA may produce meaningful attributions for large transformer models in specific scenarios, although a systematic evaluation across model types, tasks, and data modalities remains to be conducted in future work.

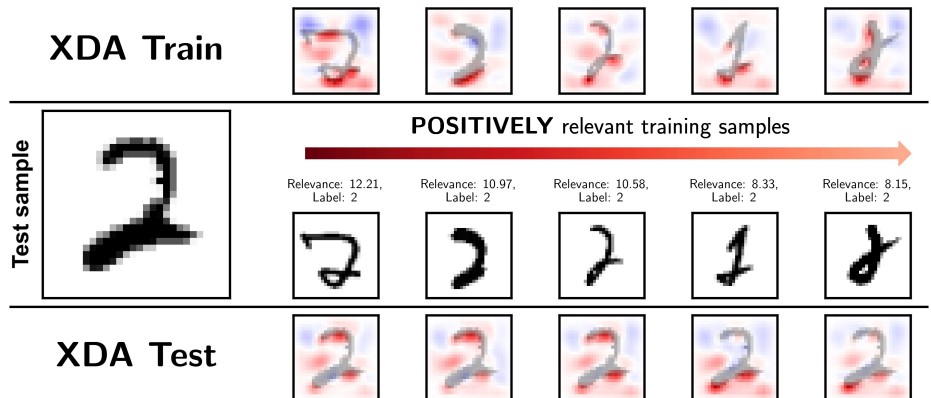

(a) DualDA attributions and XDA heatmaps for the unperturbed test sample. The three most attributed training points (left) share a similar top arc as the test sample. In the corresponding XDA heatmaps, this arc receives a large quantity of positive attribution scores. The other two training samples lack the arc, accumulating attribution at their base.

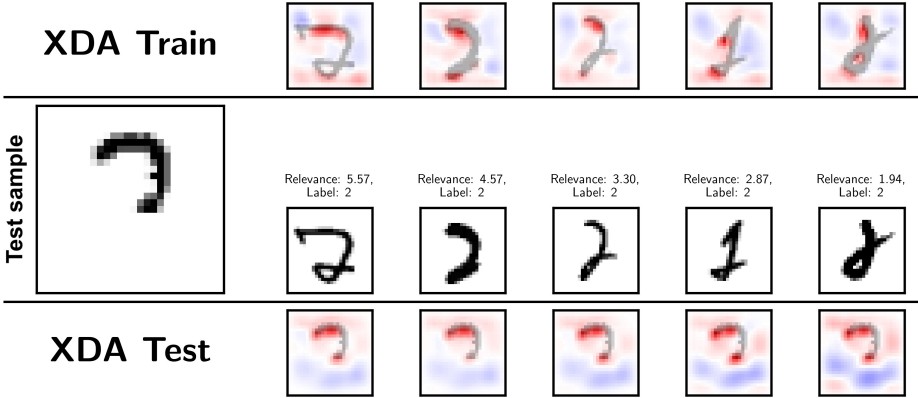

(b) When removing the base of the test sample and attributing the perturbed image that only consists of the top arc, all attributions are reduced, but the ordering induced by the data attribution scores remains the same. The XDA training sample heatmaps show a weakened attribution at the digit base. All positive relevance has vanished from the bottom of the test heatmaps.

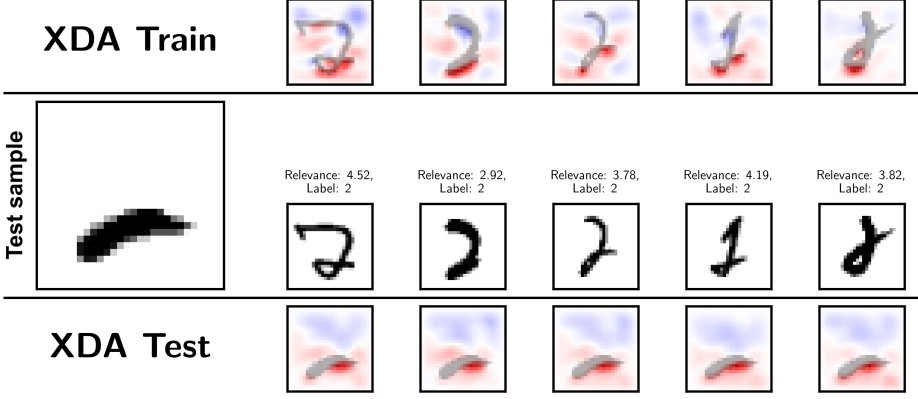

(c) When removing the upper half of the test sample and only leaving the base, the attribution of the three most attributed training points is significantly reduced. Conversely, the attribution for the remaining two training points is less affected. They now become the second and third highest attributed training samples respectively.

Figure 8: Ablation study of XDA explanations for the numeral "2" from the MNIST dataset. We take a test sample and the five most highly positively attributed training samples and evaluate how the attribution and the corresponding XDA heatmaps change when removing different parts of the digit.

The evaluation of XDA currently remains qualitative in nature. While the visualized attribution produced by XDA provide insight into feature-level interactions between training and test samples, quantitative evaluation metrics for such explanations are not yet established. Existing metrics for attribution map assessment have been developed for feature-attribution settings (Hedström et al., 2023) and do not directly apply to feature level attributions revealing interactions between training and test datapoints (instead of explaining classifier outputs), making the development of suitable evaluation measures an important direction for future work.

We note that the training of surrogate SVM models can introduce computational overhead for large training sets, although practical efficiency can be improved using early stopping or GPU-accelerated SVM implementations.

Finally, prior research suggests that combining Data Attribution methods over an ensemble of models may improve attribution efficacy. Considering the computational efficiency of DualXDA, this approach provides another potential avenue for extending DualXDA.

## 7 Discussion & Conclusion

In this work, we introduce DualXDA, a framework that enables efficient and sparse data attributions, coupled with feature-level explanations which clarify how specific training samples influence the prediction for a given test point. DualXDA consists of two parts: First, DualDA is an efficient surrogate-based DA method for explaining neural networks using the penultimate hidden representations to define a kernel map to fit a multiclass SVM. This strategy naturally provides global as well as local attributions as an approximate decomposition of the model output. Additionally, DualDA produces sparse attributions by virtue of the associated optimization process, which is beneficial for reducing the explanations' complexity and volume. Second, XDA combines data attribution with feature attribution by mapping data attribution values back to the train and test input domain simultaneously, to identify relevant corresponding features and explain previously opaque data attribution results. Together, our framework overcomes the computational inefficiency of popular DA approaches and allows the user to gain more fine-grained insights and novel analytical capabilities.

Moreover, we thoroughly evaluate our DualXDA framework using both qualitative analysis and quantitative metrics to assess its performance. First, we present the results of an extensive quantitative evaluation which compares eight state-of-the-art DA methods across seven evaluation metrics, building on existing evaluation approaches from literature, on three datasets and deep neural network architectures. We further report the memory and runtime requirements of all evaluated methods. To the best of our knowledge, this is the most extensive quantitative evaluation of DA methods to date. Our results strongly suggest that surrogate-based DA methods can help reduce the computational requirements over several orders of magnitude, without sacrificing attribution quality. Our proposed method DualDA in particular outperforms state-of-the art methods for numerous evaluation metrics while executing up to six orders of magnitude faster than competing approaches. DualDA also exhibits the lowest memory consumption among all methods that use caching. These improvements above the previous state-of-the-art can be significant in the context of applications under strict constraints on computational resources.

Additionally, we investigate the sparsity of the resulting attributions. DualDA achieves its remarkable performance while producing sparse attributions, which results in insightful attributions that are more easy to interpret than from competing methods. The resulting reduction in cognitive load is crucial to analyze and understand model behaviour in modern applications with a vast number of training samples. For use cases in which less sparse attributions may be desirable, we show how to effectively tune the sparsity using the hyperparameter $C$. Our experiments show that the hyperparameter $C$ is vital for the effectiveness of the attributions as features for downstream tasks, and can be chosen carefully by the user depending on the application at hand. As DualDA has only a single hyperparameter, it is more straightforward to apply and more user-friendly than competing methodologies.

Finally, we employ XDA to gain deeper insight into the DualDA attributions across three different datasets. Through qualitative analysis, we demonstrate how XDA explains nuances in model predictions and confounding factors in training data, while showing robustness in input ablation studies. To our knowledge, this is the first work to propose a methodology to explain the interaction of training and test datapoints in the inference

context of a trained model from both perspectives, on feature level. By unifying the attributions in two orthogonal approaches in XAI, feature attribution and data attribution, we obtain a method that leverages the benefits of both approaches, constituting a novel explanation paradigm.

## Acknowledgements

We would like to express our gratitude to Melina Zeeb for her work in the creation of Figure 1 as well as guidance on the visual representation of the results. This work was supported by the European Union's Horizon Europe research and innovation programme (EU Horizon Europe) as grants [ACHILLES (101189689), TEMA (101093003)], the Fraunhofer Internal Programs as grant PREPARE (40-08394), the Federal Ministry of Education and Research (BMBF) as grant BIFOLD (01IS18025A, 01IS180371I) and the German Research Foundation (DFG) as research unit DeSBi [KI-FOR 5363] (459422098).

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

# A Derivation of DualDA using the SVM framework

Given the soft-margin optimization problem of (Crammer & Singer, 2001)

$$
\begin{aligned}
\min_W \quad & \tfrac{1}{2}\|W\|_2^2 + C \sum_{i=1}^N \xi_i \\
\text{s.t.} \quad & \forall i \in [N], \forall c \in [K] \\
& (w_{y_i}^\top f_i) - (w_c^\top f_i) + \delta_{c y_i} \geq 1 - \xi_i
\end{aligned}
\tag{A.1}
$$

where $\delta_{c y_i} = \mathbb{1}(c = y_i)$ denotes the Kroenecker delta function, we derive the Lagrangian

$$
\mathcal{L}(W, \xi, \mathcal{D}, \alpha) = \frac{1}{2} \sum_{i=1}^K \|w_i\|_2^2 + C \sum_{i=1}^N \xi_i + \sum_{i=1}^N \sum_{c=1}^K \alpha_{ic}\big[(w_c - w_{y_i})^\top f_i + 1 - \delta_{c y_i} - \xi_i\big].
\tag{A.2}
$$

To enforce the stationarity condition (Kuhn & Tucker, 1951), we take derivatives with respect to the primal variables and set them to 0:

$$
\forall i \in [N]: \; \frac{\partial \mathcal{L}}{\partial \xi_i} = C - \sum_{c=1}^K \alpha_{ic} = 0 \implies \forall i \in [N]: \; \sum_{c=1}^K \alpha_{ic} = C
\tag{A.3}
$$

$$
\frac{\partial \mathcal{L}}{\partial w_k} = w_k + \sum_{i=1}^N \alpha_{ik} f_i - \sum_{i:y_i=k} \underbrace{\sum_{c=1}^K \alpha_{ic}}_{=C \text{ by (A.3)}} f_i = 0
\tag{A.4}
$$

$$
\implies \forall k \in [K]: \; w_k = \sum_{i=1}^N (C\delta_{y_i,k} - \alpha_{ik}) f_i
\tag{A.5}
$$

From Equations (A.3) and (A.5) we derive:

$$
\forall i \in [N]: \; \sum_{c=1}^K \alpha_{ic} = C \implies \forall i \in [N], C - \alpha_{i y_i} = \sum_{c:y_i \neq c} \alpha_{ic}
\tag{A.6}
$$

$$
\forall c \in [K]: \; w_c = \sum_{i=1}^N (C\delta_{y_i,c} - \alpha_{ic}) f_i = \sum_{i:y_i=c} (C - \alpha_{i y_i}) f_i - \sum_{i:y_i \neq c} \alpha_{ic} f_i
\tag{A.7}
$$

Combining these two results, we have:

$$
w_c = \sum_{i:y_i=c} \Big( \sum_{k:y_i \neq k} \alpha_{ik} \Big) f_i - \sum_{i:y_i \neq c} \alpha_{ic} f_i
\tag{A.8}
$$

As such, the output of the surrogate model on input $x$ is given by:

$$
\forall c \in [K]: \; \Phi(x; \vartheta, W)_c = w_c^\top f(x; \vartheta) = \sum_{i:y_i=c} \Big( \sum_{k:y_i \neq k} \alpha_{ik} \Big) f_i^\top f(x; \vartheta) - \sum_{i:y_i \neq c} \alpha_{ic} f_i^\top f(x; \vartheta)
\tag{A.9}
$$

Thus, the output of the network for any class is a linear combination of the inner products of the test point with training points. This motivates our choice in Equation (5).

To solve this problem, we first derive the dual problem by plugging Equation (A.8) in the Lagrangian and maximize it subject to the dual feasibility and complementary slackness conditions (Kuhn & Tucker, 1951):

$$
\begin{aligned}
\max_\alpha \quad & \tfrac{1}{2} \sum_{i,j=1}^N (f_i^\top f_j) \Big[ \sum_{c=1}^K (C\delta_{y_i,c} - \alpha_{i,c})(C\delta_{y_j,c} - \alpha_{j,c}) \Big] - \sum_{i=1}^N \alpha_{i,y_i} \\
\text{s.t.} \quad & \forall i \in [N], \\
& \sum_{c=1}^K \alpha_{i,c} = C \text{ and } \forall c \in [K], \; \alpha_{ic} \geq 0.
\end{aligned}
\tag{A.10}
$$

We refer the reader to (Crammer & Singer, 2001) for the derivations.

After solving this problem, we obtain a nonnegative real matrix $\mathbf{A} \in \mathbb{R}_+^{N \times K}$, consisting of the dual variables $\alpha_{i,c}$, which constitute the multiplier of each training datapoint in the final decision. Each row $i$ of $\mathbf{A}$ corresponds to a training datapoint $f_i$. Each entry in the matrix contains a positive scalar $\alpha_{ic}$ corresponding to class $c$.

Each datapoint $x_i$ *contributes negatively* to decisions for classes other than its own class $y_i$ (classes $c \neq y_i$), and the multiplier for this contribution is $\alpha_{ic}$.

Each datapoint $x_i$ *contributes positively* to decisions for its own classes ($c = y_i$), and the multiplier for this contribution is the *sum of the multipliers for other classes*: $\sum_{k \neq y_i} \alpha_{ik}$.

The variables $\alpha_{iy_i}$ are the values that satisfies the Karush-Kuhn-Tucker condition expressed in Equation (A.3), concerning positivity of slack variables.

Note that only a subset of the training data will have positive attributions, since all the datapoints outside the *margin* will have all zero values in the corresponding rows of $\mathbf{A}$.

## B Derivation of DualDA using the Representer Points Framework

To formulate the training of an SVM in the spirit of Representer Points Selection, we first reformulate the Multiclass SVM problem as an optimisation problem with the following loss function:

$$L(W; f_1^n, y_1^n) = \frac{1}{2}\|W\|_2^2 + C \sum_{i=1}^n \max_{c \in [K]} \left\{ 1 - \left[ \delta_{cy_i} + w_{y_i}^\top f_i - w_c^\top f_i \right] \right\} \tag{B.1}$$

$$\hat{W} = \arg\min_W L(W; f_1^n, y_1^n) \tag{B.2}$$

Consider data points that do not lie precisely on the decision boundary between classes, such that the argmax operation in the hinge loss function yields a unique class label. This is true for most datapoints, except a few which are situated exactly on the margin. Denote in the following

$$c^* := \arg\max_{c \in [K]} \left\{ 1 - \left[ \delta_{cy_i} + \hat{w}_{y_i}^\top f_i - \hat{w}_c^\top f_i \right] \right\} \tag{B.3}$$

and

$$\beta_c(f) = \hat{w}_c^\top f. \tag{B.4}$$

For class $c'$, we have

$$\frac{\partial}{\partial \beta_{c'}(f_i)} \ C \max_{c \in [K]} \left\{ 1 - [\delta_{cy_i} + \beta_{y_i}(f_i) - \beta_c(f_i)] \right\} = \begin{cases} -C & \text{if } c' = y_i \neq c^* \\ C & \text{if } c' = c^* \neq y_i \\ 0 & \text{if } c' = y_i = c^* \\ 0 & \text{if } c' \neq y_i, \ c' \neq c^* \end{cases}, \tag{B.5}$$

since an infinitesimal step does not change the output of the argmax operator. Setting $\ell(\beta_{c'}(f_i), y_i; \hat{W}) = C \ \max_{c \in [K]} \left\{ 1 - [\delta_{cy_i} + \beta_{y_i}(f_i) - \beta_c(f_i)] \right\}$, and replacing terms in Equation (D.6) we obtain the Representer Points attribution using the Hinge loss:

$$\tau_c^{\text{HingeRP}}(x_{\text{test}}, i) = -\frac{\partial \ell(\Phi(x_i; \theta), y_i)}{\partial \Phi(x_i; \theta)_c} f(x_i)^\top f(x_{\text{test}}) = \begin{cases} C f_i^\top f_{\text{test}} & \text{if } c = y_i \\ -C f_i^\top f_{\text{test}} & \text{if } c = c^* \\ 0 & \text{if } c = y_i = c^* \\ 0 & \text{if } c \neq y_i, \ c \neq c^* \end{cases} \tag{B.6}$$

We now establish that these attributions coincide with those of DualDA up to positive scaling. This equivalence is proven in the next section via an Influence Functions derivation of DualDA.

## C Derivation of DualDA using the Influence Functions Framework − Proof of Theorem 3.1

To prove Theorem 3.1, we again rely on the formulation given by Equations (B.1) and (B.2). We seek to quantify the effect of infinitesimally downweighting the loss contribution of the $i$-th training sample $(f_i, y_i)$ by an infinitesimal $\varepsilon$.

$$L^i(W; f_1^n, y_1^n) = L(W; f_1^n, y_1^n) - \varepsilon C \max_{c \in [K]} \left\{ 1 - \left[ \delta_{cy_i} + w_{y_i}^\top f_i - w_c^\top f_i \right] \right\} \tag{C.1}$$

$$\hat{W}^i = \arg\min_W L^i(W; f_1^n, y_1^n) \tag{C.2}$$

For a given test point $x_{\text{test}}$ with features $f_{\text{test}} = f(x_{\text{test}}; \vartheta)$ and an arbitrary class $c$, our objective is to determine how the class score $w_c^\top f_{test}$ responds to the infinitesimal downweighting of the $i$-th datapoint. Using Corollary 1 in (Naujoks et al., 2024), we derive

$$\frac{\hat{w}_c^\top f_{test} - \hat{w}_c^{i\top} f_{test}}{\varepsilon} = \nabla_{\hat{W}} \left( \hat{w}_c^\top f_{test} \right) \cdot H_{\hat{W}}^{-1} \cdot \nabla_{\hat{W}} \left( C \max_{c \in [K]} \left\{ 1 - \left[ \delta_{cy_i} + \hat{w}_{y_i}^\top f_i - \hat{w}_c^\top f_i \right] \right\} \right) + \mathcal{O}(\varepsilon), \tag{C.3}$$

where $H_{\hat{W}}$ is the Hessian of the loss function $L$, so $H_{\hat{W}}$ is equal to the identity matrix. As before, under the assumption that the maximum is achieved by a unique class index, denote in the following

$$c^* := \arg\max_{c \in [K]} \left\{ 1 - \left[ \delta_{cy_i} + \hat{w}_{y_i}^\top f_i - \hat{w}_c^\top f_i \right] \right\}. \tag{C.4}$$

For classes $c$ and $c'$, we have that

$$\nabla_{\hat{w}_{c'}} \hat{w}_c^\top f_{test} = \begin{cases} f_{\text{test}}^\top & \text{if } c = c' \\ 0 & \text{else} \end{cases} \tag{C.5}$$

and furthermore

$$\nabla_{\hat{w}_{c'}} \max_{c \in [K]} \left\{ 1 - \left[ \delta_{cy_i} + \hat{w}_{y_i}^\top f_i - \hat{w}_c^\top f_i \right] \right\} = \begin{cases} -f_i & \text{if } c' = y_i \neq c^* \\ f_i & \text{if } c' = c^* \neq y_i \\ 0 & \text{if } c' = y_i = c^* \\ 0 & \text{if } c' \neq y_i, \ c' \neq c^* \end{cases} \tag{C.6}$$

So overall

$$\frac{\hat{w}_c^\top f_{test} - \hat{w}_c^{i\top} f_{test}}{\varepsilon}$$

$$= \sum_{c'} -\nabla_{\hat{W}} \left( \hat{w}_c^\top f_{test} \right) \cdot \nabla_{\hat{W}} \left( C \max_{c \in [K]} \left\{ 1 - \left[ \delta_{cy_i} + \hat{w}_{y_i}^\top f_i - \hat{w}_c^\top f_i \right] \right\} \right) + \mathcal{O}(\varepsilon)$$

$$= \begin{cases} C f_i^\top f_{\text{test}} & \text{if } c = y_i \\ -C f_i^\top f_{\text{test}} & \text{if } c = c^* \\ 0 & \text{if } c = y_i = c^* \\ 0 & \text{if } c \neq y_i, \ c \neq c^* \end{cases} \tag{C.7}$$

We now prove that this is indeed the same as the attributions calculated by DualDA, which are given by

$$\tau_c^{\text{DD}}(x_{\text{test}}, i) = \begin{cases} \sum_{c \neq y_i} \alpha_{ic} f_i^\top f_{\text{test}} & \text{if } y_i = c \\ -\alpha_{ic} f_i^\top f_{\text{test}} & \text{else} \end{cases} \tag{C.8}$$

To establish this result, we consider two distinct cases. For correctly identified training points, we have $y_i = c^*$. By complementary slackness of the SVM, we have for the dual variables $\alpha_{ic} = 0$ for $c \neq y_i$ and therefore $\sum_{c \neq y_i} \alpha_{ic} = 0$. So

$$\tau_c^{\text{DD}}(x_{\text{test}}, i) = \begin{cases} \sum_{c \neq y_i} \alpha_{ic} f_i^\top f_{\text{test}} &= 0 & \text{if } c = y_i = c^* \\ -\alpha_{ic} f_i^\top f_{\text{test}} &= 0 & \text{if } c \neq y_i = c^* \end{cases} \tag{C.9}$$

For incorrectly identified training points, we have $y_i \neq c^*$, then by complementary slackness $\alpha_{ic} = 0$ for $c \neq c^*$ and since $\sum_{c \in [C]} \alpha_{ic} = C$, it follows that $\alpha_{ic^*} = C$. Therefore

$$\tau_c^{\text{DD}}(x_{\text{test}}, i) = \begin{cases} (C - \alpha_{ic})f_i^\top f_{\text{test}} & = & Cf_i^\top f_{\text{test}} & \text{if } c = y_i \neq c^* \\ -\alpha_{ic}f_i^\top f_{\text{test}} & = & -Cf_i^\top f_{\text{test}} & \text{if } c \neq y_i, \ c = c^* \\ -\alpha_{ic}f_i^\top f_{\text{test}} & = & 0 & \text{if } c \neq y_i, \ c \neq c^* \end{cases} \tag{C.10}$$

# D   Technical Details of Experiments

## D.1   Data Attribution Methods

As DualDA is already defined in Section 3, we now specify the other DA methods we compare our approoach to in this paper. In all experiments, we employ GPU implementations and follow the best practices as suggested by original authors of the papers or code bases, as explained in the individual sections. This includes the use of random projections and CUDA-level implementations to efficiently perform these projections for TRAK. For the caching phase of DualDA, the extraction of latent features is run using GPU capabilities, while the training of the SVM uses CPU. The sequential nature of the underlying SVM optimization algorithm does not allow for a straightforward GPU parallelization of existing implementations (Keerthi et al., 2008; Chiang et al., 2016).

### D.1.1   Influence Functions

Influence Functions (IF) originates from the field of robust statistics, where it was initially used for the analysis of the effects of outliers on linear models (Srikantan, 1961; Hammoudeh & Lowd, 2022). Koh and Liang (Koh & Liang, 2017) proposed using this method for local data attribution of deep neural networks. It approximates the effect of discarding a training point using a second-order Taylor approximation. However, this approximation is conditioned on the current trained parameters of the network and hence includes information about the decision making strategies employed by the trained model. Data attribution by IF is given by

$$\tau_c^{\text{IF}}(x, i) = -\nabla_\theta \ell(\Phi(x; \hat{\theta}), c)^\top H_\theta^{-1} \nabla_\theta \ell(\Phi(x_i; \hat{\theta}), y_i) \tag{D.1}$$

and $H_\theta = \nabla_\theta^2 \ell_{\text{train}}(\hat{\theta}, \mathcal{D})$ is the Hessian of the training loss $\ell_{\text{train}}(\hat{\theta}, \mathcal{D}) = \sum_{(x_i, y_i) \in \mathcal{D}} \ell(\Phi(x_i; \hat{\theta}), y_i)$ where $\ell(\cdot, \cdot)$ is a sample-wise loss function.

Computation of the inverse Hessian is computationally prohibitive for modern architectures with millions of parameters. For this reason, (Koh & Liang, 2017) propose to use the **LiSSA** algorithm (Agarwal et al., 2017) which approximates the inverse Hessian vector product $\ell(\Phi(x; \theta), c)^\top H_\theta^{-1}$ for each test point using an iterative procedure. While this mitigates the computational cost, it does not entirely solve the problem: computing explanations for a single test point can still take a long time, possibly in the order of hours (Hammoudeh & Lowd, 2022). In our experiments, we have followed the authors' suggestions to determine hyperparameters which produce meaningful influence values while minimizing the required computation time. First, we have used only the final layer of deep neural networks to compute gradients for influence estimation with LiSSA. Secondly, we have set the total number of iterations dedicated to the estimation of the inverse Hessian vector product to the training dataset size in each experiment.

Another estimation of the inverse Hessian is achieved by using the **Arnoldi** iteration algorithm to approximate its largest eigenvalues and the corresponding eigenvectors. These can then be used for an approximative diagonalization of the Hessian, which is simple to invert. For Arnoldi explanations, we have used 128 as the number of random projections and 150 as the Arnoldi space dimensionality. We have used 10,000 randomly sampled datapoints from the training dataset to estimate the Hessian matrix.

Finally, the authors of Grosse et al. (Grosse et al., 2023) propose to solve the inverse-Hessian-vector product using an Eigenvalue-Corrected Kronecker-Factored Approximate Curvature **(EK-FAC)**, which is derived from an approximation of the Fisher information of the network. As hyperparameters, we have used the default values given in the original code release.

### D.1.2 TracIn

TracIn circumvents the computational problem of calculating the inverse Hessian by estimating the effect of discarding a training sample with a first-order Taylor approximations. However, instead of only considering the final trained model, it aggregates this approximation over the entire training procedure using training checkpoints and the optimised training parameters $\hat{\theta}_e$ at each epoch $e$, weighted by the step size $\eta_e$:

$$\tau_c^{\text{TracIn}}(x, i) = \sum_{\text{epochs } e} \eta_e \nabla_\theta \ell(\Phi(x; \hat{\theta}_e), c)^\top \nabla_\theta \ell(\Phi(x_i; \hat{\theta}_e), y_i) \tag{D.2}$$

Two more attribution methods can be viewed as simplified version of the former. **GradDot** functions like TracIn but only considers the final training epoch:

$$\tau_c^{\text{GradDot}}(x, i) = \nabla_\theta \ell(\Phi(x; \hat{\theta}), c)^\top \nabla_\theta \ell(\Phi(x_i; \hat{\theta}), y_i) \tag{D.3}$$

As a small variation, **GradCos** uses the cosine similarity between gradients instead of the unnormalized inner product:

$$\tau_c^{\text{GradCos}}(x, i) = \cos(\nabla_\theta \ell(\Phi(x; \hat{\theta}), c), \ \nabla_\theta \ell(\Phi(x_i; \hat{\theta}), y_i)) \tag{D.4}$$

In the implementation of these methods, we have employed random projections on gradients to be able to cache training gradients, following (Pruthi et al., 2020). This allowed the attributions to be achieved in a practical time scale. We have used 128 random projections.

### D.1.3 TRAK

TRAK is originally motivated by averaging the attribution over multiple models trained on the same data to derive more stable attributions that are model-agnostic and instead only focus on the data. However, it can also be used as a single-model estimator. In this case, it is similar to Influence Functions but uses a projected version of the generalized Gauss-Newton approximation to the Hessian.

Let $p_i$ denote the prediction probability corresponding to the ground truth label of data point $z_i$. Let further $G = [g_1; g_2; \ldots; g_N]$ be a matrix with columns $g_i = \nabla_\theta \log\left(\frac{p_i}{1-p_i}\right)$ and $Q$ be a diagonal matrix with entries $Q_{i,i} = 1 - p_i$. Then the TRAK attribution for a multiclass classification task is given by:

$$\tau_c^{\text{TRAK}}(x, i) = (\nabla_\theta \Phi(x; \theta)_c^\top (G^\top G)^{-1} G^\top Q)_i. \tag{D.5}$$

Similar to TracIn, TRAK operates on random projections of gradients to achieve feasible runtimes. We use the official code release, following best practices including the GPU level implementation of random projections to dramatically decrease runtimes and memory requirements. Following the original publication, we use 2,048 random projections.

### D.1.4 Representer Points

Representer Points trains the final (fully-connected) layer until convergence with an added $\ell_2$ weight decay regularization. The authors prove a representer theorem in this setup, showing that the retrained last layer can be written as a linear combination of final hidden features of training datapoints. Therefore, the model output corresponding to class $c$ of the new model can be written as the sum of the following individual contributions. Letting $\ell(\cdot, \cdot)$ denote a sample-wise loss function, which accepts a test sample and a target class, Representer Points is defined as

$$\tau_c^{\text{RP}}(x_{\text{test}}, i) = -\frac{\partial \ell(\Phi(x_i; \theta), y_i)}{\partial \Phi(x_i; \theta)_c} f_i^\top f_{\text{test}}. \tag{D.6}$$

### D.2 Model Training

For all datasets we use stochastic gradient descent training without any learning rate scheduling. Experiments are run on Rocky Linux 8.

For MNIST, we use a 6-layer convolutional neural network with 0.001 a learning rate of 0.001.

For CIFAR-10, we train a ResNet-18 (He et al., 2016) with random cropping and flipping as data augmentations, with a learning rate of 0.0003. We further include a weight decay term with a coefficient of 0.01 as part of the loss term.

For AwA2, we have use a learning rate of 0.001 and augment the data with random horizontal flipping of the images as data augmentation for training a ResNet-50.

## E Evaluation Metrics

For evaluating attribution quality, we employ a variety of metrics from the literature and three newly proposed metrics, all of which will be introduced in this Section. These metrics are derived from an intuitive notion of data attribution or from emulating downstream applications that utilize data attribution.

Consider a test sample $x \in \mathcal{D}_{\text{test}}$, and a target class $c$ to be attributed. This means that we use DA methods to attribute the corresponding model output to the training dataset. We denote by $\pi_x^c$ a permutation of training data indices *sorted decreasingly by attribution* $\tau_c(x, \cdot)$ as assigned by a DA method such that $i > j \implies \tau_c(x, \pi_x^c(i)) \leq \tau_c(x, \pi_x^c(j))$. We let $\pi_x^c(i)$ denote the $i^{\text{th}}$ element in this list, where we start indexing at 1 (i.e. the first value corresponding to the highest attribution is $\pi_x^c(1)$). Finally, we explicitly denote the Kroenecker delta function $\delta_{a,b} = \delta(a, b)$, and we denote the prediction class for test sample $x$ with $c(x) = \arg\max_k \Phi(x; \theta)_k$ .

### E.1 Sanity Checks

Hanawa et al. (Hanawa et al., 2021) suggest two sanity checks to determine the quality of DA methods.

**Identical Class Test** The authors first posit that a reasonable explanation method should produce permutations $\pi_x^c$ that satisfy the following:

$$\forall (x, y) \in \mathcal{D}_{\text{test}}, \forall c \in [K], c = y_{\pi_x^c(1)},$$

where $\mathcal{D}_{\text{test}}$ denotes the test dataset. In plain language, the authors posit that the strongest proponents for the classification of a test sample as class $c$ should be training samples of that same class. In practice, we consider the top five attributed samples for the class predicted by the model and report

$$\frac{1}{|\mathcal{D}_{\text{test}}|} \sum_{x,y \in \mathcal{D}_{\text{test}}} \sum_{i=1}^{5} \frac{1}{5} \delta\left(c(x), y_{\pi_x^{c(x)}(i)}\right) \tag{E.1}$$

**Identical Subclass Test** Hanawa et al. further suggest that attributions should be able to detect different subpopulations in the same class (Hanawa et al., 2021). To create this condition, they propose using an arbitrary partitioning of classes, and group each set in the partition into a super group. In practice, we have selected pairs of classes to group. These pairs of classes are chosen to be dissimilar to ensure that different strategies are learned for samples belonging to different subclasses. After training a neural network on the modified dataset, a successful DA methods should attribute highly to training samples that belong to the same subclass as the original class of the test sample.

To express this condition formally, following (Hanawa et al., 2021), let $s(\cdot)$ be an oracle function that returns the true (sub)class of the datapoint given as its input. Similar to the Identical Class Test, we report

$$\frac{1}{|\mathcal{D}_{\text{test}}|} \sum_{x,y \in \mathcal{D}_{\text{test}}} \sum_{i=1}^{5} \frac{1}{5} \delta\left(s(x), s\left(x_{\pi_x^{c(x)}(i)}\right)\right) \tag{E.2}$$

### E.2 Metrics Emulating Downstream Tasks

Many downstream applications, which initially served as the motivation to several DA methods, can be used for evaluating the effectiveness of DA methods as estimators of model behaviour in counterfactual training setups. We include two metrics that measure the effectiveness of DA methods in downstream tasks.

**Mislabeling Detection**  The following metric mimics the downstream task of identifying mislabeled samples in the training data (Koh & Liang, 2017; Yeh et al., 2018; Pruthi et al., 2020). Correctly labelled datapoints can be classified by a model by relying on similar training samples of the same class. In contrast, mislabeled samples diverge from the typical feature patterns of their labeled class and consequently require individual memorization rather than generalization. Therefore, mislabelled datapoints are highly influential on *their own model prediction*, allowing the user to leverage DA approaches to detect mislabeled datapoints. To simulate this scenario, we randomly change the labels of a subset of the training data and retrain the model. We then use the attribution method to calculate each training sample's self-influence, $\tau_{y_i}(x_i, i)$. To quantify the performance of the mislabeling detection by a DA method, we traverse the training data set in descending order of self-influence and record the cumulative density function of the percentage of mislabeled samples we have identified. We report the area under the curve of this cumulative density function, linearly transformed to a score between 0 and 1, where 0 corresponds to the minimum achievable area, 1 represents the maximum achievable area, and random attribution by sampling from a uniform distribution yields an expected score of 0.5.

**Shortcut Detection**  Another candidate downstream application for DA is to identify the causes of shortcut learning and backdoor poisoning attacks. If we modify the training dataset to introduce an artificial shortcut to trigger a certain model prediction, DA methods should attribute the responsible training images highly whenever the shortcut feature is present in the test sample. Similarly to previous work (Koh & Liang, 2017; Hammoudeh & Lowd, 2022), we formulate a shortcut detection metric: determine a perturbation $\omega(\cdot)$ which takes a sample $x_i$ and adds a shortcut perturbation on it. Importantly, $\omega$ applies a consistent and detectable perturbation to all chosen samples. We use a small box in the center, and a frame around the images in our experiments.

We select a class $c'$ as the shortcut class, and randomly sample a subset of datapoints from that class to apply $\omega$. By retraining the model on this perturbed training data, the model learns a connection between the artifact and the class $c'$. We now create a particular dataset for evaluation by applying the perturbation to all samples and collecting all perturbed samples, which are not originally from class $c'$ but who, after the perturbation is applied, are classified as class $c'$, i.e. $\mathcal{D}_{\text{eval}} = \{(\omega(x), y) \mid (x, y) \in \mathcal{D}, \ y \neq c', \ \arg\max c(x) = c'\}$. This ensures that if a test point is now classified as class $c'$, it is primarily due to the influence of the perturbation and therefore, the perturbed training samples should be most influential for the classification as class $c'$. For all such test points, we report the Area Under the Precision-Recall Curve (AUPRC) score for the detection of perturbed training samples, following (Hammoudeh & Lowd, 2022).

### E.3 Retraining-based Metrics

The counterfactual relevance of training data can be empirically assessed through model retraining under an identical training regime with modified training sets. As explained in Section 2.2, this approach is the motivation for methods based on Influence Functions (Koh & Liang, 2017). The subsequent metrics evaluate the correspondence between observed changes in model behaviour and the attributions of the training data.

**Linear Datamodeling Score (LDS)**  LDS measures the effectiveness of attributions as an estimator for the effect of leaving out training samples by retraining the model on general subsets of the training data (Park et al., 2023). Given attributions $\tau_c(x, \cdot)$, and a subset of training data indices $\mathcal{S} \subset [N]$, we define

$$\xi(\mathcal{T}_c, \mathcal{S}, x) = \sum_{i \in \mathcal{S}} \mathcal{T}_c(x, i)$$

to be the overall attribution of the subset for the output $c$. We know want to measure how strongly this overall attribution is correlated with the model performance of a new counterfactual model, which was trained

only on the subset $S$. To compute the LDS, we randomly sample subsets $\{\mathcal{S}_1, \mathcal{S}_2, \mathcal{S}_3, \ldots, \mathcal{S}_m\}$. We train models $\{\Phi_1, \Phi_2, \ldots, \Phi_m\}$ independently on these subsets. Finally, we compute the average rank correlations of actual outputs and predicted outputs over the test set:

$$\frac{1}{|\mathcal{D}_{\text{test}}|} \sum_{(x,y) \in \mathcal{D}_{\text{test}}} \rho(\{\Phi_j(x)_{c(x)} \mid j \in [m]\}, \{\xi(\mathcal{T}_{c(x)}, \mathcal{S}_j, x) \mid j \in [m]\}) \tag{E.3}$$

where $\rho(\cdot, \cdot)$ denotes Spearman rank correlation.

**Corseset Selection** A downstream application of data attribution is dataset distillation: limiting the training dataset to fewer examples, while maintaining high test accuracy (Guo et al., 2022; Joaquin et al., 2024). In contrast to LDS, we are not interested in the model performance for randomly chosen subsets of the training data, but instead for the highest attributed subset. To measure this in a metric, we need a strategy to determine the global importance of a training point for the model performance on the entire test set. To achieve this, we compute the average value of the absolute attributions of a training point over the test set. We then restrict the training data to the most globally relevant datapoints and measure the test loss of a model retrained on the restricted training set. We report the test loss when retrained on 10% of the data as well as a weighted average

$$\overline{CS} = \frac{\sum_{p \in \mathcal{P}} (\frac{1}{p}) \cdot \ell_{test}^{CS}(p)}{\sum_{p \in \mathcal{P}} \frac{1}{p}},$$

where $\ell_{test}^{CS}(p)$ is the test loss when retrained only on the most relevant $p\%$ of the training data and we use steps of 10%, i.e. $\mathcal{P} = \{10, \ldots, 90\}$. This is the only metric for which lower values indicate better performance. Additionally, we report model accuracy variations across different training subsets in Appendix F.4.

**Adversarial Data Pruning** Inspired by previous work, which investigates the sensitivity of model predictions to ablation of the training data (Ilyas et al., 2022), we measure the test loss of a model that was retrained on a train set which had its most influential datapoints removed. Similarly to Coreset Selection, we consider the highest attributed subset. But in contrast, we aim to judge how much exclusive relevant information this subset holds that would induce an increase in the loss when the subset is removed. We use the same strategy as in Coreset Selection to determine the global importance of training samples. We report the test loss when retrained without the 10% most influential data as well as a weighted average

$$\overline{DP} = \frac{\sum_{p \in \mathcal{P}} (\frac{1}{p}) \cdot \ell_{test}^{DP}(p)}{\sum_{p \in \mathcal{P}} \frac{1}{p}},$$

where $\ell_{test}^{DP}(p)$ is the test loss when retrained without the most relevant $p\%$ of the training data and we again choose $\mathcal{P} = \{10, \ldots, 90\}$. As for the Coreset Selection metric, we also report model accuracy across different training subsets in Appendix F.5.

## F Experiments: Additional Findings

### F.1 Metric Scores for MNIST and CIFAR-10

Figures F.1 and F.2 show the evaluation results for MNIST and CIFAR-10 datasets respectively.

### F.2 Surrogate Faithfulness Scores

Figure F.3 shows the surrogate faithfulness scores for DualDA and Representer Points, as well as artificially sparsified Representer Points surrogate models.

### F.3 DualDA: Effect of sparsity hyperparameter $C$

We evaluate how different choices of the sparsity hyperparameter affect DualDA attribution performance across the evaluation metrics. We therefore evaluate all metrics by choosing $C$ over an expanded range from

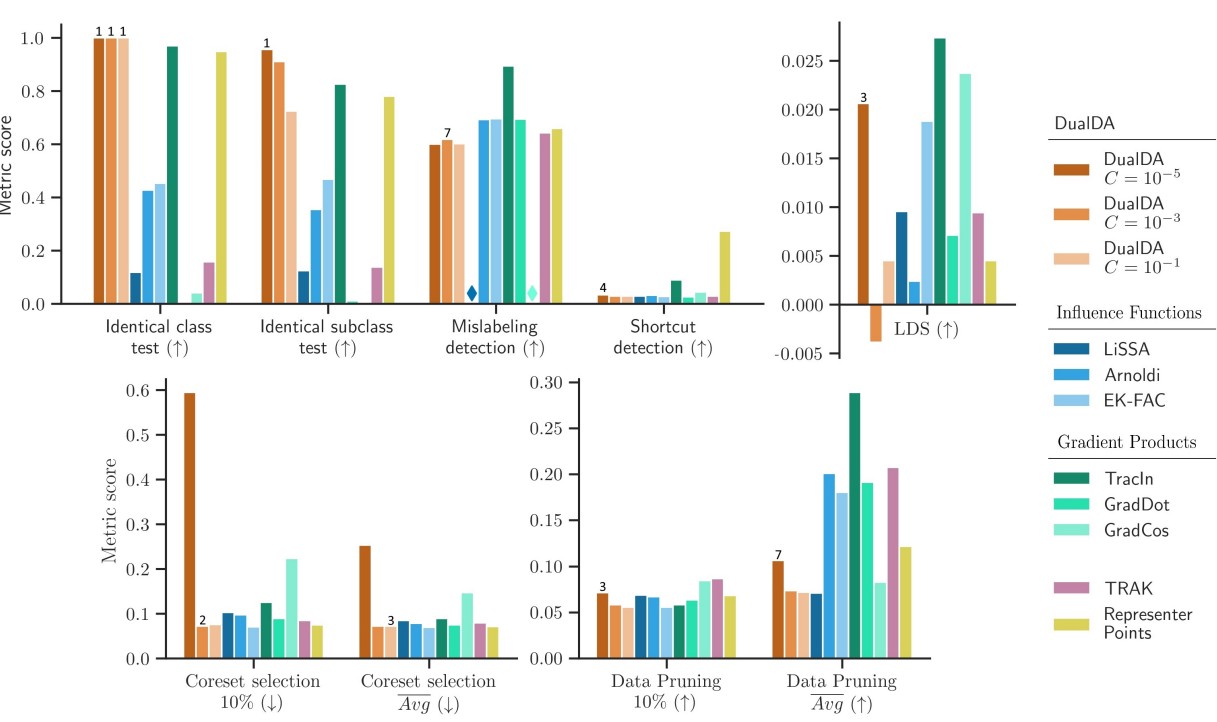

Figure F.1: Evaluation results on the MNIST dataset. LiSSA and GradCos do not have a score for the mislabeling detection metric. Note that Mislabeling Detection requires calculating the self-influence of the entire train set (see Appendix E). For LiSSA, calculating self-attributions for all training points is computationally infeasible. As GradCos is defined as the cosine of the angle between the test and training sample's feature vectors, the self-attribution for GradCos is equal to 1 regardless of the sample.

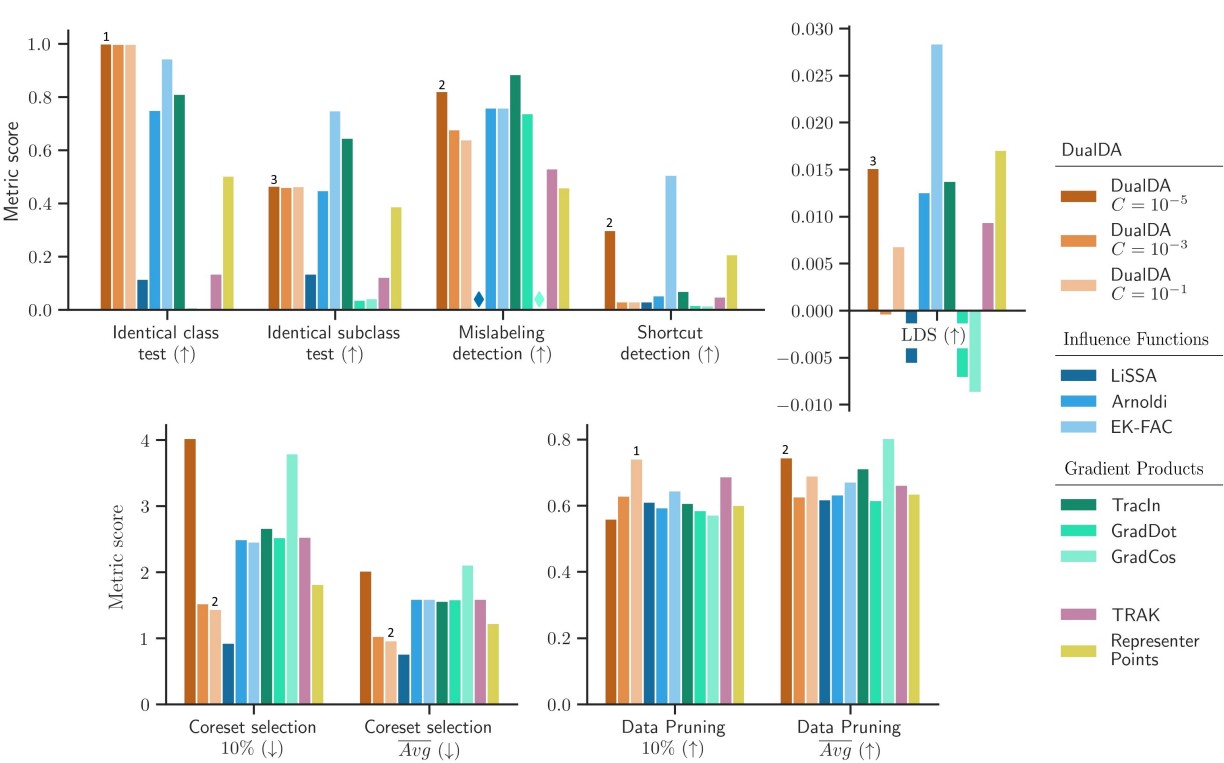

Figure F.2: Evaluation results on the CIFAR-10 dataset. LiSSA and GradCos do not have a score for the mislabeling detection metric. Note that Mislabeling Detection requires calculating the self-influence of the entire train set (see Appendix E). For LiSSA, calculating self-attributions for all training points is computationally infeasible. As GradCos is defined as the cosine of the angle between the test and training sample's feature vectors, the self-attribution for GradCos is equal to 1 regardless of the sample.

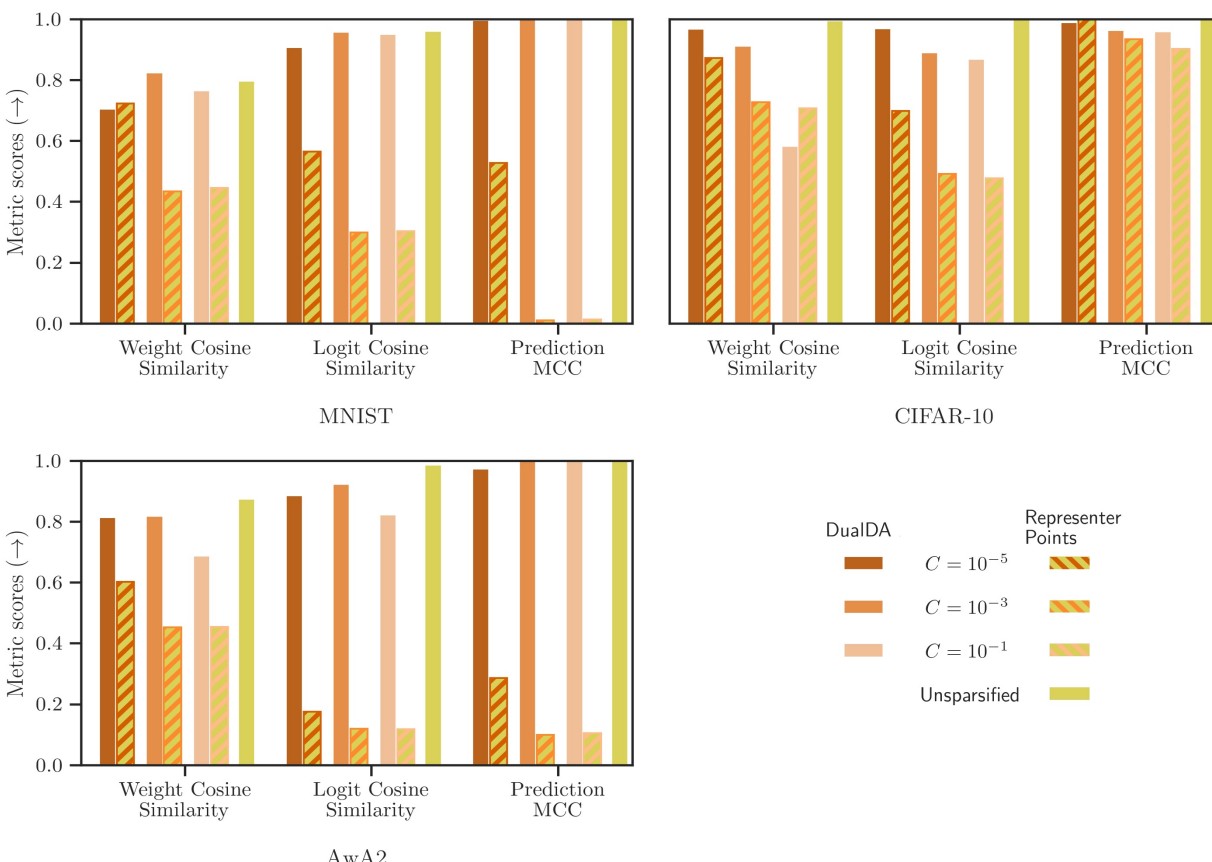

Figure F.3: Surrogate similarity metrics for DualDA with sparsity hyperparameter $C \in \{10^{-5}, 10^{-3}, 10^{-1}\}$ and Representer Points on the MNIST, CIFAR-10, and AwA2 dataset. Additionally, we sparsified Representer Points artificially to the same sparsity levels as DualDA (in terms of the number of support vectors) with the corresponding sparsity parameter $C$ and present corresponding surrogate similarity metrics as well.

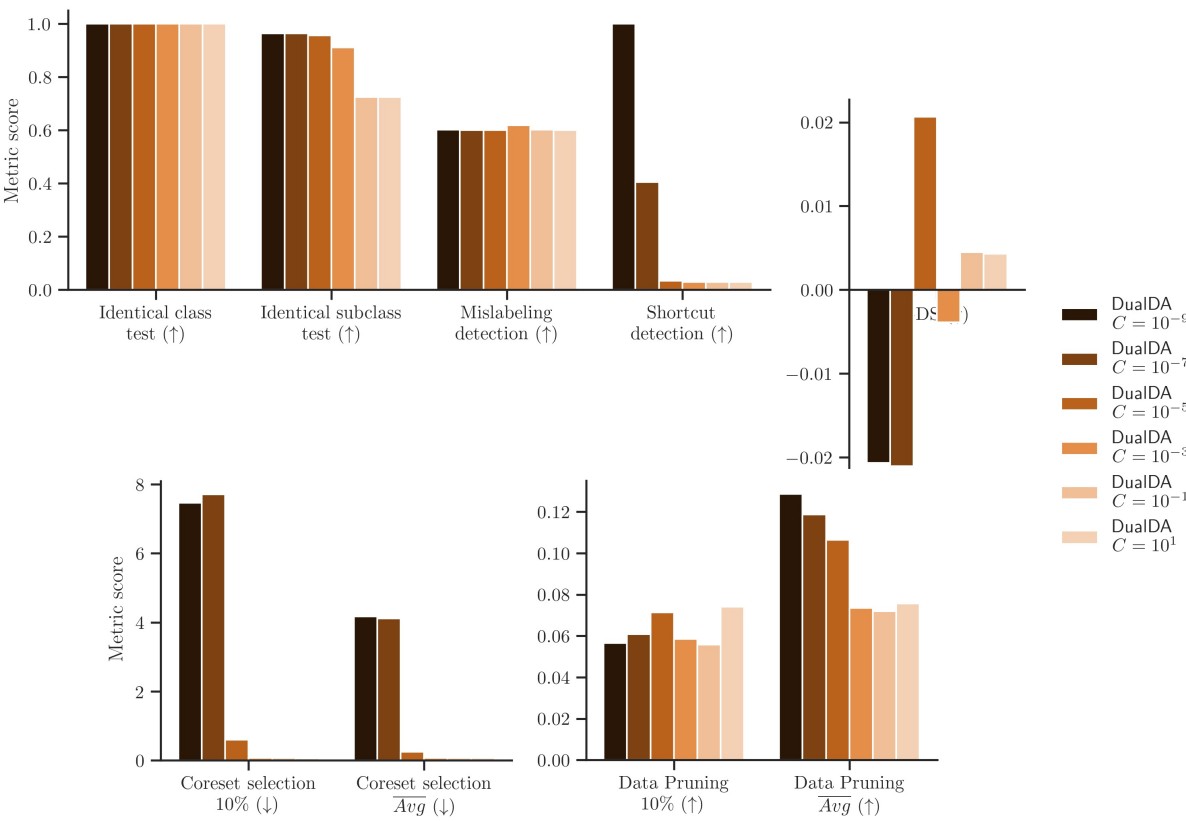

Figure F.4: Evaluation results on the MNIST dataset for DualDA with different sparsity hyperparameters.

$10^{-9}$ to $10^1$ for DualDA. Figure F.4 shows the results for the MNIST dataset, Figure F.5 for CIFAR-10, and Figure F.6 for the AwA2 dataset. DualDA performs best in the Identical Class Test, irrespective of the chosen hyperparameter. For the Identical Subclass Test, lower sparsity appears preferable for better performance, boosting scores from 0.72 to 0.96 on the MNIST dataset and from a low of 0.45 to a maximum of 0.70 on AwA2. However, on CIFAR-10, scores remain largely unaffected by the choice of $C$. Mislabeling Detection is not affected by the sparsity level on MNIST and CIFAR-10, but on AwA2, higher sparsity allows the method to achieve near-perfect performance. Shortcut Detection benefits substantially from lower sparsity levels. For example, on MNIST, where DualDA with $C \in \{10^{-5}, 10^{-3}, 10^{-1}\}$ fails at the task as presented in Figure F.1, for $C = 10^{-9}$, DualDA achieves a perfect score. For the LDS metric, no clear pattern emerges: on MNIST, lower sparsity yields worse results, but on CIFAR-10 and AwA2, it increases scores. Coreset Selection is improved by higher sparsity on all tasks, likely because the SVM selects more important training points as support vectors. This selection effect is reduced when we decrease sparsity and therefore select more support vectors. The effect on Adversarial Data Pruning is also unclear: for MNIST and CIFAR-10, lower sparsity yields better results, but on AwA2, the opposite is true.

These findings reveal that optimal sparsity settings are both metric- and dataset-dependent. While some metrics show consistent patterns across datasets, others exhibit dataset-specific behaviors that require hyperparameter selection for optimal performance.

Furthermore, we analyze how the sparsity hyperparameter affects the fit of the surrogate SVM. The top row of plots in Figure F.7 displays the training and test accuracies of DualDA surrogate models for $C \in \{10^{-6}, 10^{-5}, 10^{-4}, 10^{-3}, 10^{-2}, 10^{-1}, 10^0, 10^1, 10^2\}$ in comparison to the test accuracy of the original model. The middle row shows the training time in black. The final row displays the number of support vectors per training class, where each coloured line represents one class. In general, the figure suggests that an optimal choice for $C$ always exists, satisfying the requirements for fast training to fit a DualDA surrogate,

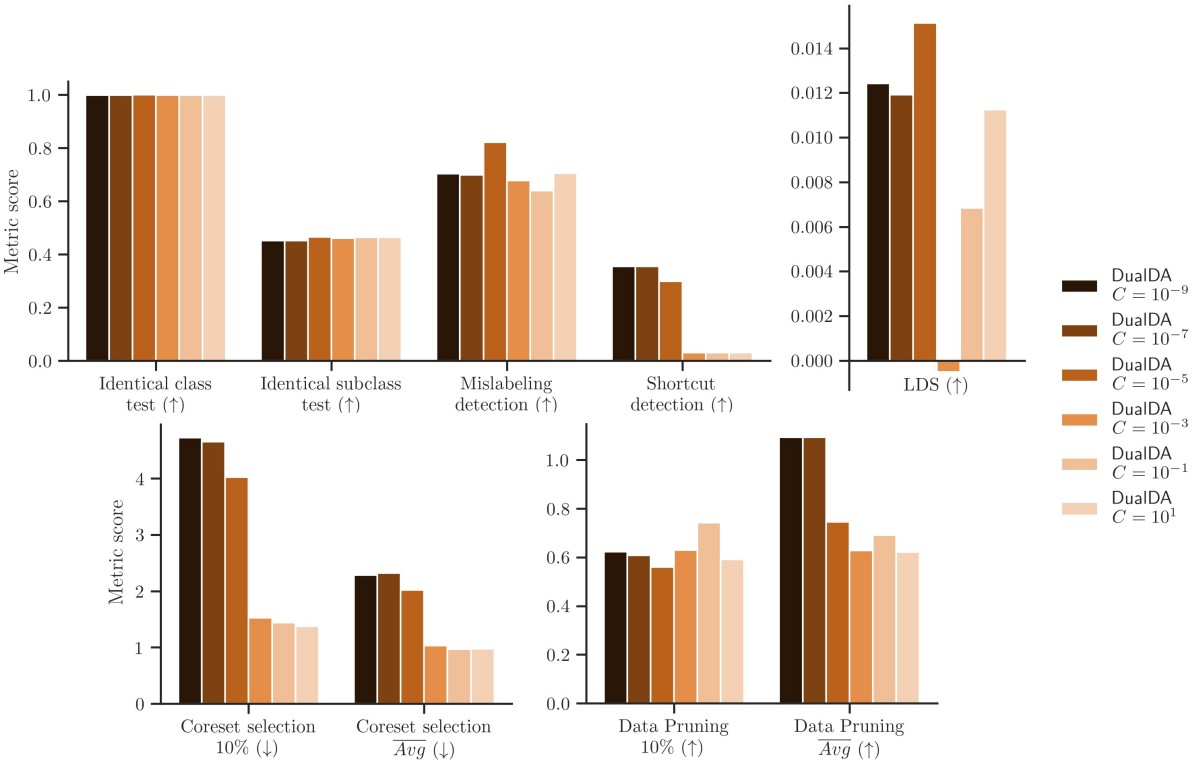

Figure F.5: Evaluation results on the CIFAR-10 dataset for DualDA with different sparsity hyperparameters.

closely matching the original model, and resulting in an adequate amount of support vectors to assure a faithful DualDA surrogate generating sparse and faithful data attributions. Our extended results further show that with an optimal choice of hyperparameter $C$, DualDA can be used as a highly effective method for solving several DA-related downstream tasks.

These analyses of the sparsity and accuracy of the surrogate SVM models, alongside corresponding evaluation results indicate that the choice of $C = 10^{-3}$ yields a sparse surrogate while preserving strong fidelity to the original model and stable attribution performance in the image classification task with convolutional neural networks. However, under alternative setups, where the task, data modality and model architecture differ, the optimal hyperparameter might change. In such cases, an independent hyperparameter optimization process may be necessary under these evaluation criteria. This process is considerable more tractable compared to competing approaches, due to the computational efficiency provided by DualDA.

### F.4 Accuracy Rates for Coreset Selection Metric

Accuracy rates for the Coreset Selection task are presented in Table F.1 for the MNIST dataset, in Table F.2 for the CIFAR-10 dataset, and in Table F.3 for the AwA2 dataset. We observe that increasing the sparsity hyperparameter enhances the performance of DualDA on this task, establishing it as the top-performing method across multiple sparsity levels on all three datasets.

### F.5 Accuracy Rates for Adversarial Data Pruning Metric

Accuracy rates for the Adversarial Data Pruning task are presented in Table F.4 for the MNIST dataset, in Appendix F.5 for the CIFAR-10 dataset, and in Table F.6 for the AwA2 dataset. In contrast to the Coreset Selection task, no consistent pattern emerges regarding the optimal sparsity level for DualDA: reduced sparsity produces superior results on MNIST and CIFAR-10, whereas increased sparsity yields better performance

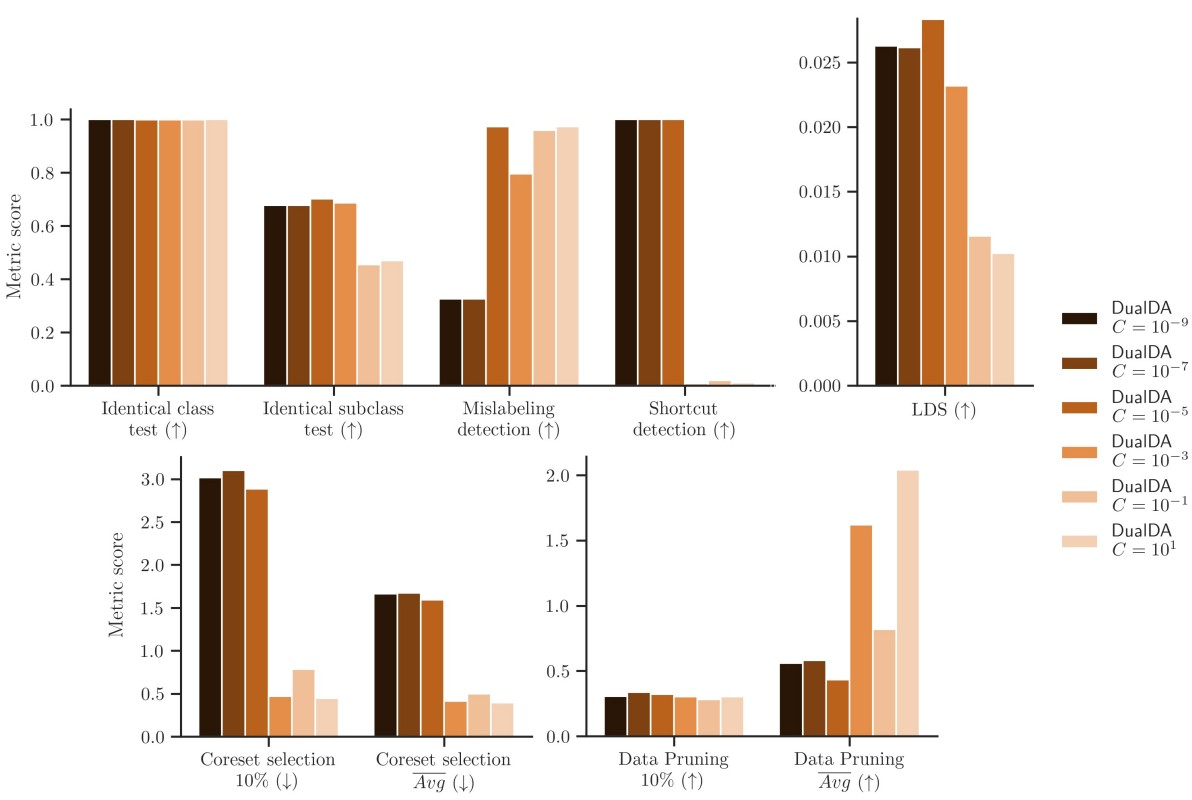

Figure F.6: Evaluation results on the AwA2 dataset for DualDA with different sparsity hyperparameters.

| | 10% | 20% | 30% | 40% | 50% | 60% | 70% | 80% | 90% |
|---|---|---|---|---|---|---|---|---|---|
| DualDA $C = 10^{-9}$ | 40.81% | 56.88% | 70.75% | 77.49% | 82.50% | 87.21% | 89.40% | 98.66% | 98.44% |
| DualDA $C = 10^{-7}$ | 43.20% | 57.33% | 69.64% | 78.86% | 83.89% | 86.91% | 89.73% | 98.28% | 98.68% |
| DualDA $C = 10^{-5}$ | 83.23% | 98.84% | 98.78% | 98.70% | 98.79% | 98.69% | 98.58% | 98.80% | 98.65% |
| DualDA $C = 10^{-3}$ | 98.24% | 98.41% | 98.56% | 98.74% | 98.75% | 98.68% | 98.68% | **98.89%** | 98.73% |
| DualDA $C = 10^{-1}$ | 98.31% | 98.49% | 98.38% | 98.73% | 98.95% | 98.71% | 98.76% | 98.70% | **98.83%** |
| DualDA $C = 10^{1}$ | 98.35% | 98.41% | 98.54% | 98.56% | 98.56% | 98.64% | 98.75% | 98.63% | 98.54% |
| LiSSA | 97.09% | 98.18% | 98.19% | 98.71% | 98.86% | 98.70% | 98.64% | **98.89%** | 98.68% |
| Arnoldi | 98.18% | 98.84% | 98.84% | 98.65% | **98.99%** | 98.80% | 98.58% | 98.76% | 98.58% |
| EK-FAC | 98.41% | 98.61% | 98.69% | 98.78% | 98.79% | 98.64% | **98.86%** | 98.74% | 98.73% |
| TracIn | 97.20% | 98.53% | 98.60% | **98.90%** | 98.74% | **98.89%** | 98.56% | 98.79% | 98.56% |
| GradDot | 98.31% | 98.75% | 98.80% | 98.65% | 98.54% | 98.70% | 98.73% | 98.59% | 98.73% |
| GradCos | 94.50% | 96.41% | 98.04% | 98.40% | 98.39% | 98.39% | 98.68% | 98.81% | 98.59% |
| TRAK | 98.38% | **98.86%** | 98.75% | 98.83% | 98.59% | 98.73% | 98.84% | 98.70% | 98.75% |
| Representer Points | **98.43%** | 98.76% | **98.96%** | 98.86% | 98.84% | 98.59% | 98.83% | 98.81% | 98.71% |

Table F.1: Coreset Selection accuracy rates on MNIST dataset (model retrained on top $x\%$ of training data, sorted by attribution values of corresponding DA method). Higher accuracy rates indicate better DA performance on the task of the metric. Best values are highlighted in bold.

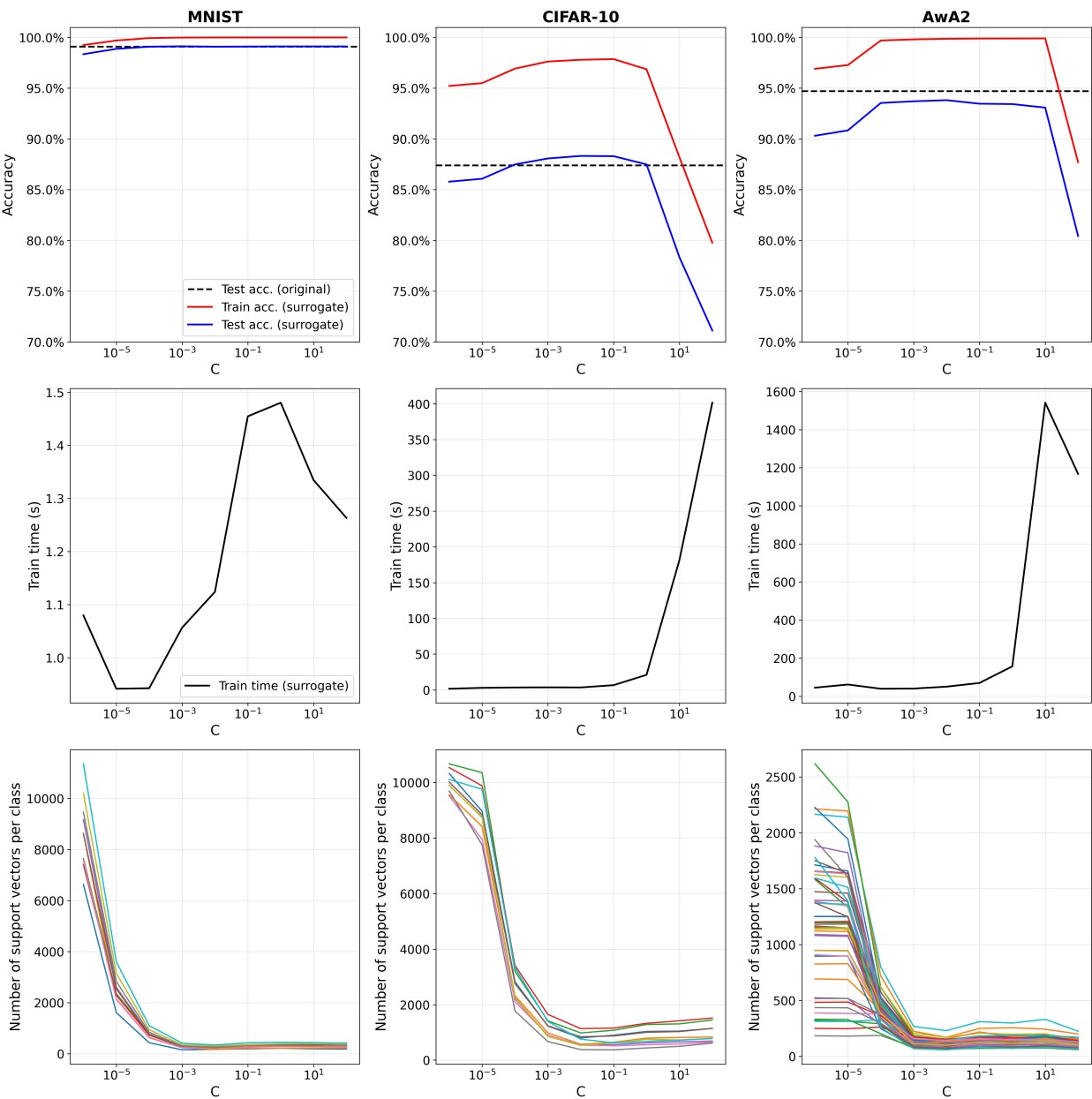

Figure F.7: Further analysis of the effect of the hyperparameter $C$ on the training of the SVM. First row: Accuracy rates on the train and test set are compared to the accuracy rate of the original model on the test set for all three datasets. The SVM performs well over a large range of hyperparameters, reaching accuracy rates on the test set in line with the performance of the original models. However, overtly strong regularisation substantially degrades performance on the CIFAR-10 and AwA2 datasets. Second row: The training time for the surrogate is displayed for all three datasets. Overall, the training time is very short, but exceedingly strong regularisation leads to a remarkable increase in training time. Third row: Each line plots the number of support vectors, i.e. samples that have a non-zero contribution to the fit of the SVM, for one class of the train set. Stronger regularisation leads to a reduction in support vectors.

| | 10% | 20% | 30% | 40% | 50% | 60% | 70% | 80% | 90% |
|---|---|---|---|---|---|---|---|---|---|
| DualDA $C = 10^{-9}$ | 22.71% | 57.38% | 68.79% | 73.63% | 77.16% | 78.33% | 78.58% | 79.75% | 81.16% |
| DualDA $C = 10^{-7}$ | 24.10% | 54.31% | 66.96% | 72.14% | 75.20% | 78.35% | **81.25%** | 80.48% | 81.73% |
| DualDA $C = 10^{-5}$ | 28.60% | 59.55% | 71.43% | 73.08% | 79.04% | 80.08% | 79.93% | 80.98% | 81.28% |
| DualDA $C = 10^{-3}$ | 47.15% | 69.35% | 74.78% | 77.98% | 79.50% | 77.16% | 80.05% | 80.48% | 81.38% |
| DualDA $C = 10^{-1}$ | 55.60% | 73.18% | 77.31% | 77.13% | 79.86% | 81.15% | 81.24% | 80.10% | 80.44% |
| DualDA $C = 10^{1}$ | 56.90% | 70.74% | 76.23% | **78.46%** | 79.16% | 80.38% | 80.19% | 78.24% | 78.64% |
| LiSSA | **68.71%** | **76.68%** | **79.71%** | 75.63% | **80.40%** | 79.73% | 81.13% | 79.04% | 81.14% |
| Arnoldi | 40.39% | 50.61% | 64.50% | 75.69% | 75.99% | 77.53% | 78.10% | 81.16% | 80.56% |
| EK-FAC | 35.79% | 51.74% | 62.26% | 73.46% | 74.91% | 79.74% | 80.14% | 80.49% | 80.98% |
| TracIn | 28.18% | 47.41% | 72.23% | 78.10% | 78.76% | **81.18%** | 80.10% | 81.35% | 80.96% |
| GradDot | 38.95% | 51.83% | 66.76% | 75.23% | 75.79% | 76.63% | 78.80% | 79.80% | 79.43% |
| GradCos | 27.48% | 57.81% | 65.86% | 71.35% | 77.89% | 77.45% | 80.20% | 80.28% | **81.79%** |
| TRAK | 38.23% | 51.97% | 65.44% | 72.78% | 75.96% | 77.61% | 80.26% | **81.75%** | 81.10% |
| Representer Points | 45.89% | 60.93% | 71.59% | 77.03% | 79.51% | 80.83% | 80.26% | 80.94% | 79.71% |

Table F.2: Coreset Selection accuracy rates on CIFAR-10 dataset (model retrained on top $x\%$ of training data, sorted by attribution values of corresponding DA method). Higher accuracy rates indicate better DA performance on the task of the metric. Best values are highlighted in bold.

| | 10% | 20% | 30% | 40% | 50% | 60% | 70% | 80% | 90% |
|---|---|---|---|---|---|---|---|---|---|
| DualDA $C = 10^{-9}$ | 57.01% | 77.53% | 80.87% | 87.88% | 89.44% | 90.46% | 92.48% | 92.90% | 93.38% |
| DualDA $C = 10^{-7}$ | 57.89% | 74.54% | 83.64% | 88.08% | 89.21% | 90.21% | 91.80% | 92.88% | 91.56% |
| DualDA $C = 10^{-5}$ | 50.90% | 78.39% | 84.56% | 89.18% | 90.76% | 90.81% | 91.27% | 91.40% | 93.38% |
| DualDA $C = 10^{-3}$ | 85.18% | 90.59% | 90.59% | 90.52% | 91.89% | 91.91% | 93.43% | 92.75% | 93.38% |
| DualDA $C = 10^{-1}$ | 77.62% | **91.47%** | 92.39% | 91.75% | **93.10%** | 91.87% | 92.92% | 92.41% | 93.87% |
| DualDA $C = 10^{1}$ | **87.09%** | 90.65% | 90.35% | 91.91% | 90.72% | 89.25% | 92.26% | 91.62% | 92.68% |
| LiSSA | 16.27% | 33.93% | 42.74% | 44.79% | 50.86% | 61.67% | 67.50% | 79.00% | 87.00% |
| Arnoldi | 63.79% | 89.99% | 92.39% | **93.05%** | 92.70% | 92.46% | 92.30% | 93.01% | 93.38% |
| EK-FAC | 59.59% | 84.04% | **92.57%** | 91.91% | 92.88% | 92.28% | 92.90% | 93.38% | 93.84% |
| TracIn | 65.54% | 86.08% | 91.40% | 92.26% | 93.07% | 93.32% | 92.41% | **93.53%** | 93.43% |
| GradDot | 56.75% | 83.71% | 89.60% | 90.98% | 92.75% | **93.40%** | 93.12% | 92.79% | 93.76% |
| GradCos | 82.41% | 84.57% | 87.77% | 89.42% | 90.79% | 92.74% | 93.34% | 92.57% | 92.52% |
| TRAK | 67.41% | 89.80% | 92.37% | 91.67% | 92.30% | 92.83% | **93.86%** | **93.53%** | 93.20% |
| Representer Points | 72.29% | 90.22% | 91.86% | 92.79% | 92.79% | 93.31% | 92.98% | 93.09% | **93.93%** |

Table F.3: Coreset Selection accuracy rates on AwA2 dataset (model retrained on top $x\%$ of training data, sorted by attribution values of corresponding DA method). Higher accuracy rates indicate better DA performance on the task of the metric. Best values are highlighted in bold.

| | 10% | 20% | 30% | 40% | 50% | 60% | 70% | 80% | 90% |
|---|---|---|---|---|---|---|---|---|---|
| DualDA $C = 10^{-9}$ | **53.28%** | 83.51% | 90.00% | 97.23% | 97.10% | 98.28% | 98.66% | 98.69% | 98.74% |
| DualDA $C = 10^{-7}$ | 63.35% | 81.85% | 94.39% | 96.39% | 97.75% | 98.60% | 98.68% | 98.79% | 98.76% |
| DualDA $C = 10^{-5}$ | 97.15% | 97.31% | 97.59% | 97.88% | 97.60% | 97.20% | 97.90% | 97.96% | 98.39% |
| DualDA $C = 10^{-3}$ | 97.74% | 98.36% | 98.41% | 98.49% | 98.54% | 98.39% | 98.50% | 98.49% | 98.71% |
| DualDA $C = 10^{-1}$ | 97.94% | 97.94% | 98.38% | 98.01% | 98.71% | 98.48% | 98.54% | 98.73% | 98.63% |
| DualDA $C = 10^1$ | 97.74% | 98.40% | 98.06% | 98.65% | 98.53% | 98.31% | 98.70% | 98.70% | 98.63% |
| LiSSA | 98.18% | 98.54% | 98.56% | 98.70% | 98.66% | 98.65% | 98.50% | 98.60% | 98.86% |
| Arnoldi | 74.96% | 80.03% | 89.20% | 92.95% | 96.83% | 97.41% | 97.89% | 98.24% | 98.46% |
| EK-FAC | 74.76% | 77.24% | 86.80% | 94.51% | 96.98% | 98.04% | 98.29% | 98.08% | 98.54% |
| TracIn | 78.38% | 84.08% | 86.48% | **89.23%** | **90.96%** | **93.48%** | **96.35%** | **96.85%** | **97.88%** |
| GradDot | 72.66% | **74.76%** | 86.89% | 94.18% | 97.08% | 96.68% | 97.78% | 98.11% | 98.65% |
| GradCos | 93.93% | 95.69% | 98.28% | 98.34% | 98.46% | 98.59% | 98.76% | 98.45% | 98.80% |
| TRAK | 82.69% | 81.34% | **80.51%** | 94.04% | 96.93% | 97.60% | 97.84% | 98.08% | 98.56% |
| Representer Points | 76.83% | 93.61% | 95.80% | 96.93% | 96.96% | 97.09% | 97.74% | 97.96% | 98.59% |

Table F.4: Adversarial Data Pruning accuracy rates on MNIST dataset (model retrained on bottom $x$% of training data, sorted by attribution values of corresponding DA method). Lower accuracy rates indicate better DA performance on the task of the metric. Best values are highlighted in bold.

| | 10% | 20% | 30% | 40% | 50% | 60% | 70% | 80% | 90% |
|---|---|---|---|---|---|---|---|---|---|
| DualDA $C = 10^{-9}$ | 30.84% | 39.05% | 55.66% | 64.90% | 67.63% | **68.43%** | 69.25% | **71.13%** | 78.70% |
| DualDA $C = 10^{-7}$ | **29.40%** | **36.18%** | **54.19%** | **62.94%** | **66.98%** | 69.06% | **68.78%** | 72.59% | **78.39%** |
| DualDA $C = 10^{-5}$ | 64.96% | 68.09% | 69.68% | 72.31% | 71.99% | 74.35% | 76.35% | 78.04% | 79.39% |
| DualDA $C = 10^{-3}$ | 71.94% | 70.41% | 77.48% | 77.46% | 80.03% | 81.56% | 78.80% | 81.75% | 80.74% |
| DualDA $C = 10^{-1}$ | 72.53% | 78.48% | 78.71% | 80.18% | 80.53% | 80.61% | 81.43% | 81.05% | 81.08% |
| DualDA $C = 10^1$ | 70.69% | 73.94% | 78.68% | 77.48% | 79.71% | 78.48% | 81.34% | 81.04% | 81.89% |
| LiSSA | 71.19% | 75.44% | 78.25% | 77.94% | 79.40% | 80.11% | 80.51% | 81.20% | 81.28% |
| Arnoldi | 65.75% | 70.56% | 76.51% | 77.74% | 77.08% | 78.78% | 81.69% | 80.44% | 81.68% |
| EK-FAC | 63.28% | 69.44% | 74.23% | 73.88% | 77.66% | 78.50% | 79.86% | 80.95% | 81.08% |
| TracIn | 60.69% | 62.95% | 69.33% | 71.99% | 75.20% | 77.09% | 79.60% | 79.74% | 80.13% |
| GradDot | 61.31% | 66.90% | 76.69% | 77.71% | 79.04% | 78.58% | 81.06% | 80.43% | 80.84% |
| GradCos | 44.51% | 57.80% | 59.44% | 63.89% | 68.49% | 69.90% | 74.74% | 78.98% | 82.31% |
| TRAK | 65.79% | 72.56% | 75.08% | 79.10% | 78.36% | 78.20% | 80.14% | 82.71% | 81.74% |
| Representer Points | 67.93% | 75.59% | 77.36% | 78.46% | 77.08% | 77.98% | 80.55% | 78.19% | 82.14% |

Table F.5: Adversarial Data Pruning accuracy rates on CIFAR-10 dataset (model retrained on bottom $x$% of training data, sorted by attribution values of corresponding DA method). Lower accuracy rates indicate better DA performance on the task of the metric. Best values are highlighted in bold.

on AwA2. With optimized hyperparameters, DualDA achieves substantial performance gains: when models are retrained on only 10% of the data, accuracy is reduced to 53.28% compared to 72.66% achieved by the second-best method. On CIFAR-10, DualDA similarly attains the best performance at the same data retention level, with an accuracy of 29.40% versus 44.51% for the second-best approach. On AwA2, it reduces accuracy to 12.07%, performing competitively with the top method, LiSSA, which achieves 9.19%.

### F.6 Details of Mislabeling Detection

In Figure F.8, we present the curves produced by using different DA methods to detect mislabeled training samples as explained in Appendix E under the paragraph about the Mislabeling Detection metric.

### F.7 Details of Sparsity Experiment

In Figure 5, we analyze the distribution of absolute attribution scores over the training set for various DA methods, exemplified on the AwA2 dataset. Since positive and negative attributions carry distinct semantic

|  | 10% | 20% | 30% | 40% | 50% | 60% | 70% | 80% | 90% |
|---|---|---|---|---|---|---|---|---|---|
| DualDA $C = 10^{-9}$ | 53.43% | 70.60% | 75.28% | 77.95% | 84.90% | 87.66% | 91.45% | 93.29% | 93.69% |
| DualDA $C = 10^{-7}$ | 46.50% | 71.40% | 73.26% | 79.48% | 83.16% | 89.01% | 91.34% | 92.33% | 93.43% |
| DualDA $C = 10^{-5}$ | 65.59% | 73.95% | 80.63% | 83.84% | 86.35% | 88.59% | 89.10% | 92.79% | 92.52% |
| DualDA $C = 10^{-3}$ | 13.90% | 28.30% | 42.00% | 56.31% | 67.46% | 81.58% | 91.67% | 92.75% | 93.38% |
| DualDA $C = 10^{-1}$ | 19.94% | 42.61% | 63.13% | 81.69% | 91.58% | 91.14% | 93.36% | 91.36% | 94.19% |
| DualDA $C = 10^{1}$ | 12.07% | 26.08% | 36.98% | 50.64% | 61.83% | 73.17% | 83.95% | 93.84% | 92.81% |
| LiSSA | **9.19%** | **20.38%** | **31.55%** | **40.22%** | **48.22%** | **62.53%** | **73.17%** | **80.59%** | **85.95%** |
| Arnoldi | 79.95% | 89.64% | 86.92% | 91.82% | 92.52% | 92.20% | 93.43% | 93.76% | 93.56% |
| EK-FAC | 78.06% | 88.28% | 88.35% | 91.56% | 92.02% | 93.20% | 93.05% | 93.64% | 92.79% |
| TracIn | 54.84% | 69.59% | 80.96% | 82.59% | 90.85% | 90.19% | 91.36% | 93.60% | 93.42% |
| GradDot | 84.48% | 88.92% | 88.08% | 90.11% | 91.45% | 92.19% | 92.83% | 93.49% | 92.98% |
| GradCos | 73.64% | 88.61% | 88.33% | 90.46% | 91.60% | 92.13% | 93.10% | 94.02% | 93.27% |
| TRAK | 85.99% | 88.79% | 89.18% | 91.40% | 90.15% | 92.66% | 93.32% | 93.82% | 93.23% |
| Representer Points | 86.21% | 90.08% | 87.64% | 90.04% | 92.20% | 92.06% | 93.89% | 93.67% | 93.42% |

Table F.6: Adversarial Data Pruning accuracy rates on AwA2 dataset (model retrained on bottom $x$% of training data, sorted by attribution values of corresponding DA method). Lower accuracy rates indicate better DA performance on the task of the metric. Best values are highlighted in bold.

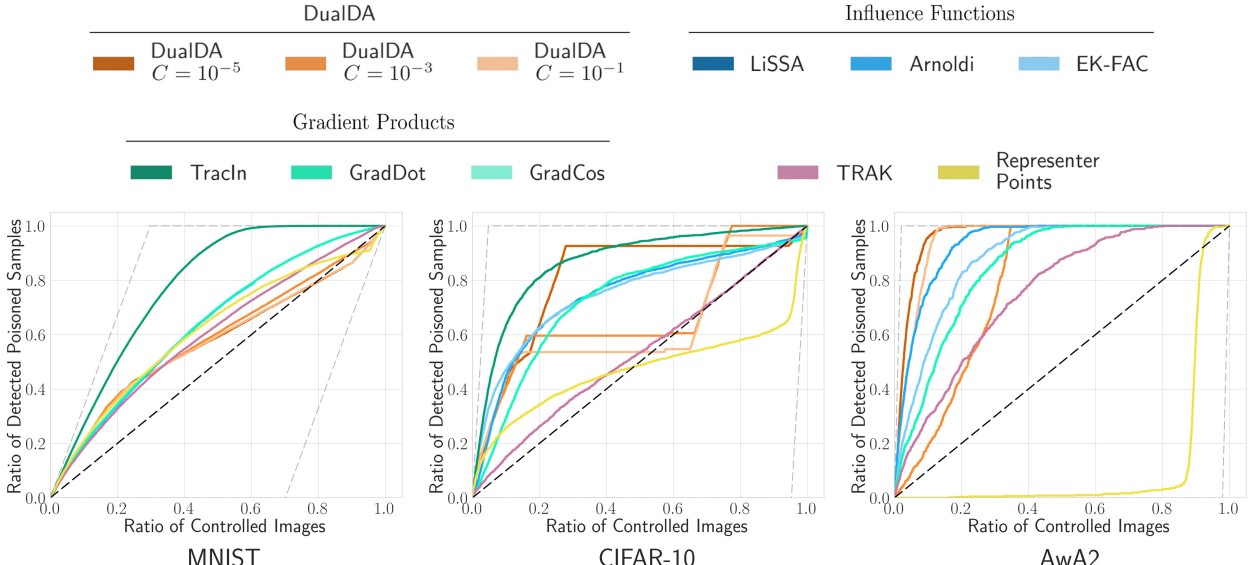

Figure F.8: Mislabeling detection curves for the three datasets. Dashed grey lines indicate the best and worst possible curves for a DA method, depending on the amount of poisoned samples for each dataset. The score is the area under each curve which falls inside area inside the grey dashed curves, normalized to be between 0 and 1. Note that for the AwA2 dataset, the curves for TracIn and GradDot overlap, such that only the curve corresponding to GradDot is visible.

interpretations, it is worthwhile to investigate the distribution of positive and negative attributions for different method. In this appendix, we provide a more detailed view into the attributions, taking into account the distribution of negative and positive attributions. Initially, we examine the overall ratio of cumulative positive attribution versus negative attribution, displayed on the left side of the figure. Positive attributions show that a training point carries information corroborating the model prediction being explained, as such, they are expected to be concentrated on fewer datapoints from the corresponding class, compared to negative attributions which carry information about different kinds of evidence contradicting the prediction at hand. As expected, most explainers – with the exception of GradDot and GradCos – exhibit greater positive than negative cumulative attribution.

The proportion of negative to positive cumulative attribution varies significantly across methods. The LiSSA implementation of Influence Functions shows the lowest ratio at only 53.7%, while TRAK demonstrates the highest at 98.7%.

For DualDA, we observe that the sparsity hyperparameter $C$ directly affects this ratio – increasing values of $C$ (i.e., increasing sparsity) bring the positive cumulative relevance increasingly closer to the negative cumulative relevance.

On the right side of Figure F.9, we analyze the growth of the cumulative sum of both positive and negative attribution as a function of the fraction of most influential datapoints included, to assess the sparsity characteristics of attributions with distinct semantics. When comparing the development of the cumulative sum between positive and negative attribution, we observe that DualDA maintains consistent sparsity across both types. In contrast, Arnoldi and EK-FAC show increased sparsity for negative attribution, while Representer Points and GradDot demonstrate significantly reduced sparsity for negative attribution compared to positive attribution.

## G   Additional DualXDA Results

We present additional DualXDA explanations in Figures G.1 to G.6.

## H   Unifying Concept-Level Explanations, Feature Attribution and Data Attribution

The formulation of DualDA can be combined with concept-level explanations to understand the impact that individual concepts – which may be present or absent in the training and test data – have on their combined DualDA attribution value. For a given concept $k$, assume through a concept-discovery method such as linear probes we have obtained a Concept Activation Vector (CAV) $\hat{v}_k$ (Kim et al., 2018). The larger the dot product between the hidden activations $f_{\text{test}}$ and the vector $v_k$, the more likely it is that the concept is present in the test sample. We further normalize the CAV $v = \frac{\hat{v}_k}{\|\hat{v}_k\|_2}$. Then, the amount of the DualDA attribution that is explained by concept $k$ can be quantified as

$$\tau_c^{\text{DD}}(x_{\text{test}}, i \mid k) = \begin{cases} (C - \lambda_{iy_i}) \cdot (f_i^\top v_k)(f_{\text{test}}^\top v_k) & \text{if } y_i = c \\ -\lambda_{ic} \cdot (f_i^\top v_k)(f_{\text{test}}^\top v_k) & \text{else} \end{cases} \tag{H.1}$$

Furthermore, if given a set of concepts $\mathcal{K}$ of which the corresponding CAVs $\{v_k \mid k \in \mathcal{K}\}$ form an orthogonal basis of the final hidden layer, then this concept decomposition fulfills a **conservation property**:

$$\tau_c^{\text{DD}}(x_{\text{test}}, i) = \sum_{k \in \mathcal{K}} \tau_c^{\text{DD}}(x_{\text{test}}, i \mid k) \tag{H.2}$$

An example for such a basis is given by the final layer neurons, whose CAVs are the unit vectors of the Cartesian coordinate system. Furthermore, the concept-attribution values $\tau_c^{\text{DD}}(x_{\text{test}}, i \mid k)$ can be propagated back through the network, similarly to the per-support-vector attributions originating from DualDA via XDA. This will provide us for each concept with attribution maps for both training and test samples in which the attribution that is mediated through the concept is localized in the input space.

An illustrative application of this methodology to an ImageNet sample is presented in Figure H.1. The attribution decomposition is performed with respect to concepts encoded by neurons in the penultimate

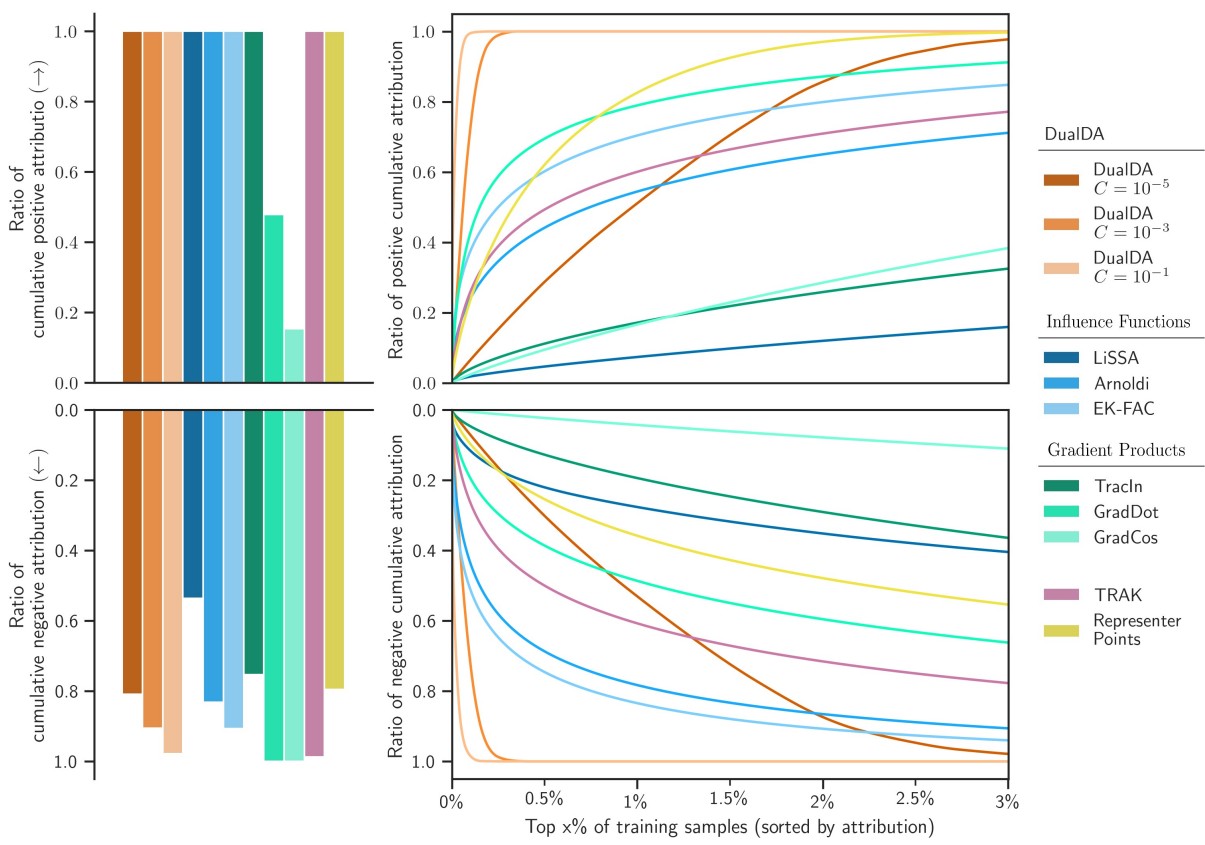

Figure F.9: Analysis of cumulative distribution of positive and negative attributions for various DA methods on the AwA2 dataset. The *y*-axis represents the cumulative sum of attributions over the dataset, ordered descendingly by attribution magnitude.

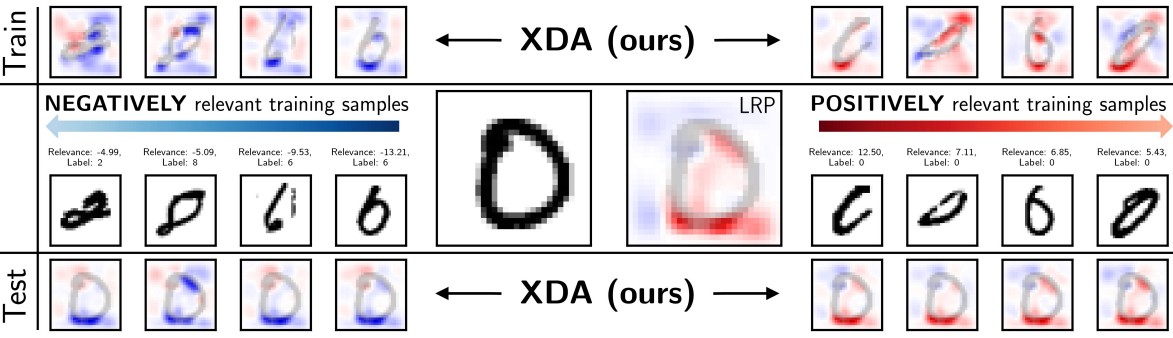

Figure G.1: DualXDA explanations for the numeral "0" from the MNIST dataset. The XDA heatmaps indicate, that in the strongest proponent the part missing part of the digit to close the loop is negatively relevant for the classification as class "0".

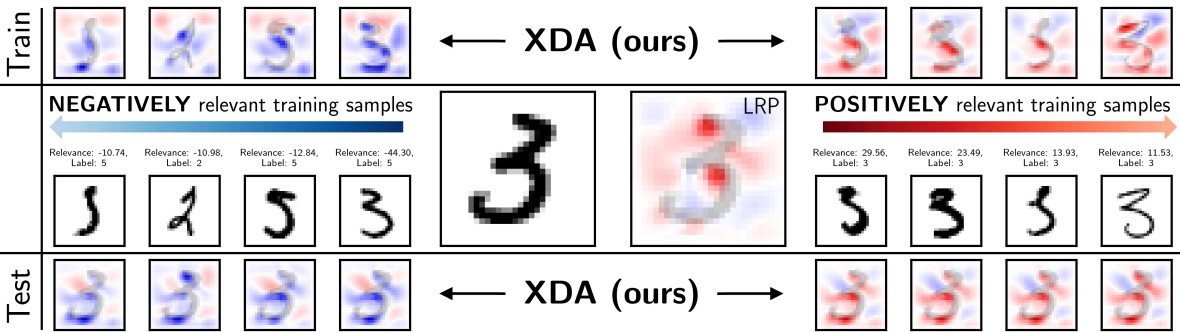

Figure G.2: DualXDA explanations for the numeral "3" from the MNIST dataset. This example shows how DualDA can be used to detect and understand mistakes in the training data: The most negatively relevant training sample is an image of a "3" mistakenly labeled as a "5".

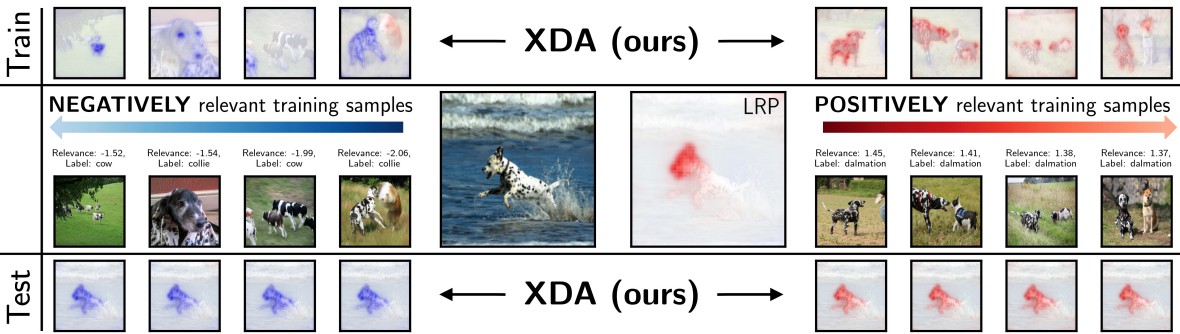

Figure G.3: DualXDA explanations for a dalmatian from the AwA2 dataset. Note that the most negatively relevant training sample is a picture labelled as "collie" that also contains a dalmatian next to a collie breed dog. This the interaction of those class-dependent features with the model can be observed in the XDA heatmap for the training sample: The part of the image that contains the dalmatian looks similarly to the test sample, but because the image is labeled "collie", this area is negatively relevant for the classification of the test sample as a dalmatian. On the other hand, the part that contains the collie looks different from the test sample and is therefore not evidence contrary to the model decision. Therefore this area of the image is red. In general, XDA marks black and white spotted areas as influential, either for the decision (if this is the fur of dalmatians) or against the decision (when it is the fur of cows or the pattern of a blanket).

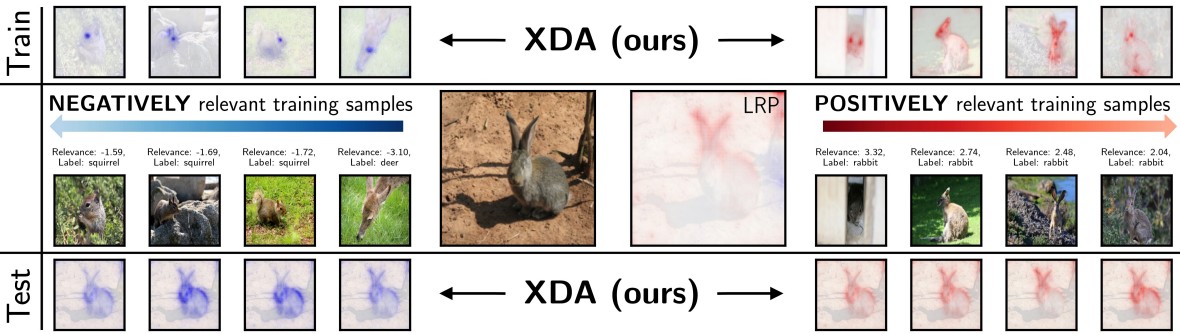

Figure G.4: DualXDA explanations for a rabbit from the AwA2 dataset. For the proponents, the entire body of the rabbit including the ears are relevant. However, the fur is also negatively relevant due to many opponents having a similar fur colour. Yet because few other animals have similarly long ears, the hare's ears are the main identifying feature in the overall XDA map.

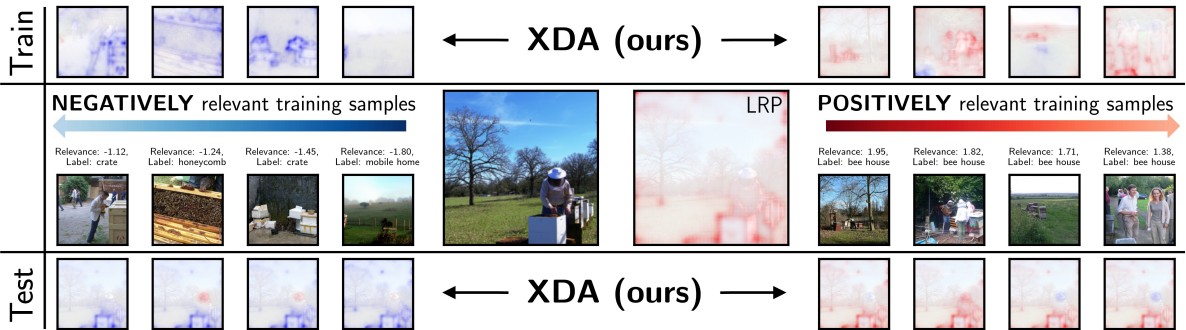

Figure G.5: DualXDA explanations for a bee house from the ImageNet dataset. Opponents feature box-like structures that look similar to apiaries. For the proponents, XDA considers the featured beekeeper hat positive evidence towards the classification only if a similar hat is also shown in the training image.

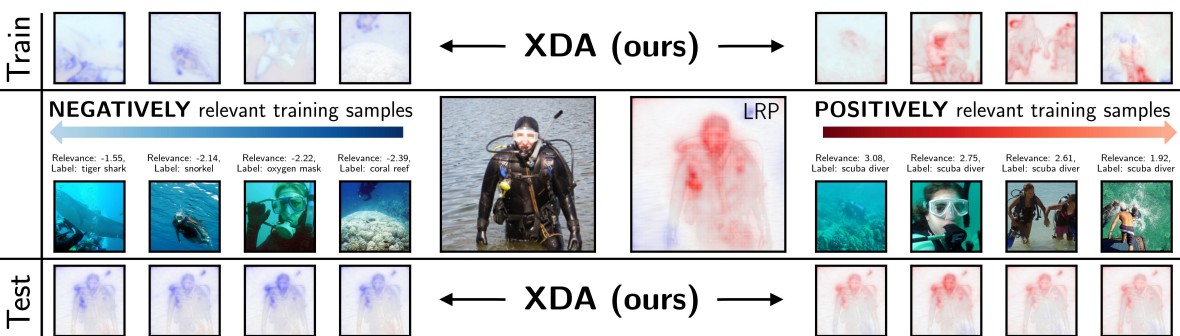

Figure G.6: DualXDA explanations for a scuba diver from the ImageNet dataset. Relevance is predominantly placed on the oxygen mask if it is featured prominently in the testing image.

network layer. Heatmaps are visualized for the three neurons contributing the highest relevance scores. To determine their corresponding conceptual representations, we present the nine training samples that maximize the activation of each neuron: the first neuron responds predominantly to turtle imagery, the second exhibits selectivity for hemispherical structures oriented upward such as shells, domes, helmets, and lamps, while the third demonstrates sensitivity to surfaces exhibiting scaled or dimpled textures as on golf balls, reptilian skin, and artichokes. Although neuron-level decomposition provides valuable interpretable insights, individual neurons frequently encode multiple concepts simultaneously, or encode concepts incompletely, which complicates conceptual understanding of the neurons and results in more diffuse, uniformly distributed activation patterns in the corresponding heatmaps. Future research should investigate decomposition approaches utilizing linear probing techniques to identify more semantically coherent and interpretable representational subspaces. Alternatively, methods such as PURE (Dreyer et al., 2024) allow us to first disentangle neurons into multiple monosemantic units, which improves the clarity of neuron-level decompositions.

# I Preliminary Experiments on Large Language Models

To assess the applicability of our method to Large Language Models, we conduct preliminary experiments that will be expanded in future work. We fine-tune a pre-trained Llama-3.2-1B model (Dubey et al. (2024), released under the Llama 3.1 Community License, available on Huggingface[5]) on the AG News dataset for news topic classification (Zhang et al. (2015), available on Huggingface[6]). The dataset contains 120,000 training samples, each consisting of a news description and a label indicating one of four categories. We assess both the resource consumption for attribution creation and the attribution quality through a Shortcut Detection experiment. All experiments have been run on a NVIDIA H200 Tensor Core GPU.

## I.1 Resource Consumption

In this section, we report the computational resources required by DualDA for $C \in \{10^{-5}, 10^{-3}, 10^{-1}\}$ and compare them to EK-FAC (Grosse et al., 2023), the most computationally efficient variant of Influence Functions available to date. Table I.1 presents the results on the caching time, cache space, average explanation time and VRAM usage for both methods.

Generating attribution scores with EK-FAC for the Llama model demands substantial VRAM. The official implementation[7] highlights the parameter `per_device_train_batch_size`, which allows the user to trade-off VRAM consumption against minor changes in explanation times. To ensure a fair comparison, we report EK-FAC's results using its most efficient configurations for each metric. Specifically, the VRAM usage reported in Table I.1 corresponds to the absolute minimal setting (i.e. `per_device_train_batch_size=1`), while the explanation time corresponds to the most efficient setting we obtained with an available VRAM of 145 GB (i.e. `per_device_train_batch_size=32`).

The precomputation phase of DualDA involves two steps: (1) caching final hidden features and (2) training the SVM surrogate. Feature extraction dominates both caching time and cache size. Table I.1 reports total caching time and cache size, while the training durations for the SVM surrogate models are as follows: 18 minutes 27 seconds for $C = 10^{-1}$, 48 seconds for $C = 10^{-3}$, and 32 seconds for $C = 10^{-5}$. The storage required for the trained SVM parameters requires only 1.9 MB across all values of $C$.

The results show that DualDA achieves improvements of 42–116× in caching time, 35× in cache size, at least $108,000\times$ in average explanation time and at least 3× in VRAM usage.

## I.2 Shortcut Detection Experiment

We adapt the task of Shortcut Detection (Koh & Liang, 2017; Hammoudeh & Lowd, 2022), which identifies spurious correlations in training data that enable models to exploit dataset-specific biases, from the visual

---

[5]`https://huggingface.co/meta-llama/Llama-3.2-1B`
[6]`https://huggingface.co/datasets/sh0416/ag_news`
[7]`https://github.com/pomonam/kronfluence`

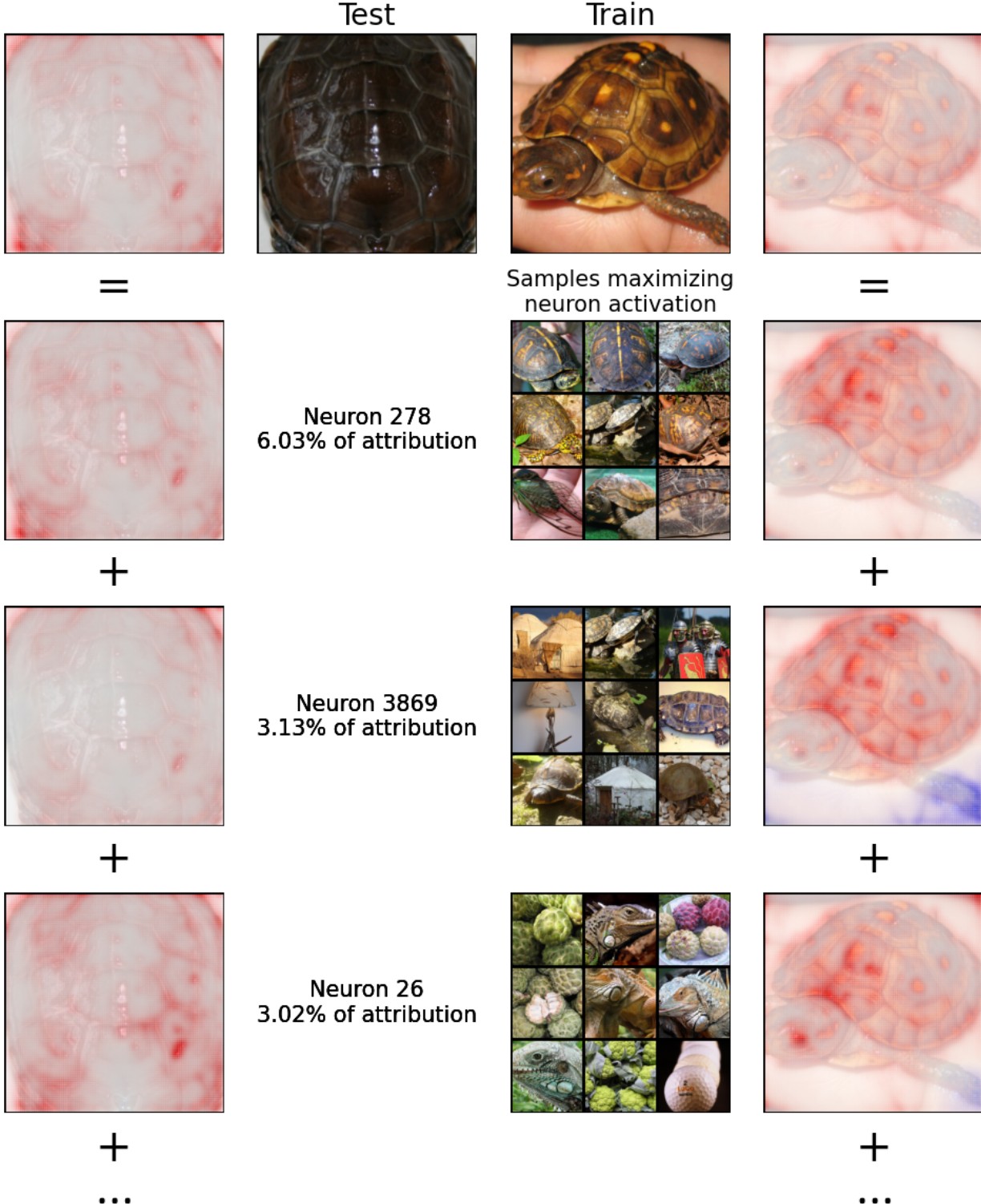

Figure H.1: Attribution decomposition for a turtle image and its highest-attributed training sample, with neuron-specific contributions ranked in descending order of attribution.

|  | DualDA | EK-FAC | Factor of Improvement |
|---|---|---|---|
| Caching Time (s) | $620 - 1695$ | 72460 | $42\times - 116\times$ |
| Cache Size (GB) | 0.99 | 34.60 | $35\times$ |
| Explanation Time per test sample (s) | 0.04 | $> 4345$ | $> 108,000\times$ |
| VRAM Usage (GB) | 8.7 | $> 29.5$ | $> 3\times$ |

Table I.1: Computational resource comparison between DualDA and EK-FAC for Llama-3.2-1B trained on the AG News dataset. For DualDA, we report the range of caching times, including feature extraction and SVM training, over three tested settings ($C \in \{10^{-5}, 10^{-3}, 10^{-1}\}$). Results for EK-FAC are reported using its most efficient `per_device_train_batch_size` for each metric. DualDA demonstrates substantially lower caching time, cache size, explanation time, and VRAM usage across all settings.

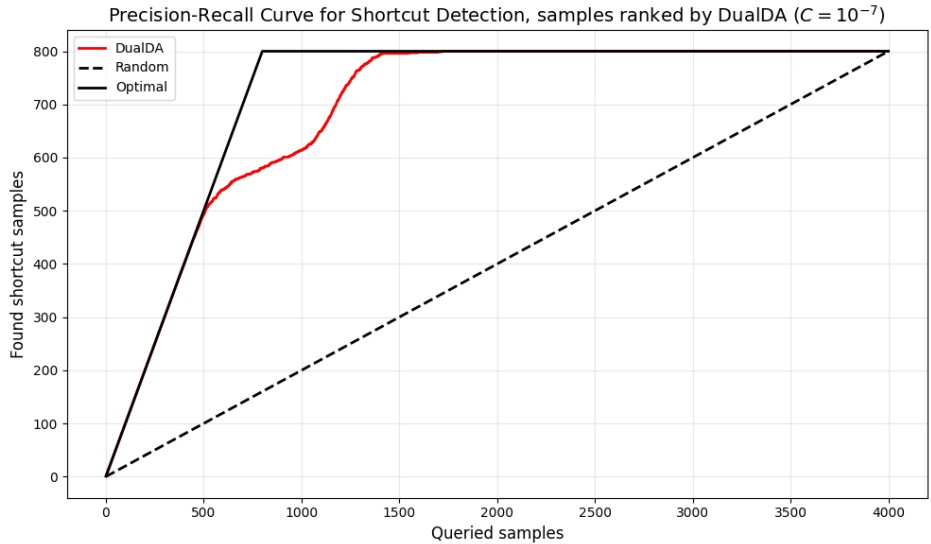

Figure I.1: Precision-Recall Curve for the task of identifying shortcutted training samples from attribution values.

domain to our language setting. We construct a training set of 4,000 randomly selected samples and introduce a shortcut into 20% of them by randomly reassigning the label to a class *cls* and inserting the corresponding keyword '$<cls>$' at a random position in the sample text. We apply the same manipulation to create a shortcutted test set consisting of 1,000 test samples. Since models exploiting these shortcuts should rely heavily on the shortcutted training samples, we hypothesize that Data Attribution methods assign high attribution to these samples when explaining predictions on shortcutted test instances. We quantify the overall effect of each training sample on the entire test set by summing its attributions across all shortcutted test instances.

Consistent with our vision domain experiments, lower sparsity improves Shortcut Detection performance. At sparsity levels of $C = 10^{-5}$ or higher, DualDA fails to meaningfully differentiate between shortcutted and non-shortcutted samples. However, at $C = 10^{-7}$ or lower, DualDA successfully identifies shortcuts: it assigns the highest overall attribution to nearly half of all shortcutted samples, and all shortcutted samples are retrieved within the top 50% of ranked training samples. The corresponding Precision-Recall curve (Figure I.1) achieves an AUPRC score of 0.88.

