# OpenReview forum: "Sparse, Efficient and Explainable Data Attribution with DualXDA"
_TMLR — Accepted by TMLR_

### Review · Reviewer_tgqp · 2025-09-02

**Summary Of Contributions:**

This paper talked about the data attribution issue in the LLM context. Specifically, they proposed a novel approach for efficient and effective DA, leveraging Support Vector Machine theory to provide fast and naturally sparse data attributions for AI predictions.

**Additional Comments:**

If the authors make substional major revision by reducing the text and complex illustrative figures. I could recheck this paper.

**Audience:**

Yes

**Audience Explanation:**

I would think the contributions are still valuable for the community
-  A fast, efficient calculation in data attribution is important in an explainable LLM.
-  This paper leverages several co-exiting methods and empirical results seem promising.

**Claims And Evidence:**

No

**Claims Explanation:**

Frankly, I am NOT successfully completing reading this paper, particularly in section 2 and 3. I found many parts are seemingly missing from my viewpoint. The followings are the reading notes

**s.2 related work**

- definition 2.1 Is not clear from my viewpoint. For example, the

> Global DA methods assign each pair of training sample and class c a real-valued attribution which indicates the relevance or influence of the training datapoint on the fit of the model Φ(·;θ)c

This sentence does not rigorously describe what is the global DA exactly, for example, what is each pair of training samples? Φ is a model rather than score? What is the relevance or influence of the training datapoint?

> Local DA methods produce a function \tau for each class c which accepts as input a test sample and the index for a training point. The value \tau indicates the relevance or influence of the training datapoint zi on the model output.

So what is \tau, as input to a test sample and the index for a training point ?

> advocates for a different standpoint, aiming to estimate the changes induced by fine-tuning an already trained model with an altered training set instead.

I would suggest using formal math equations rather than wording descriptions.

- In addition, the gradient based, surrogate based methods require better math equations.

**s 3.1 Using Support Vector Machines for Sparse and Efficient Data Attribution**

In equation (1), the formulation is standard SVM and equation (5) is about test data. I would suggest a clear consistency check in s.2.

I think the logic transition from the (1) to (7) becomes unclear for me.

**about paper organizations and figures**

Indeed, I think this may be the main reason why I cannot continue reading the paper. The beginning of the paper is filled with large amounts of text. Figure 1 seems intuitive, but it's actually counterproductive because I have to keep checking the figure and the equation back and forth. In particular, certain subtle arrows and shapes make me repeatedly guess the actual meaning of the figure. Combined with the unclear equations, I am finally lost.

**Requested Changes:**

I would strongly suggest a major revision for s 1-3 to make clear

- the math definition
- a simplified figure without subtle details to quickly understand the overall information. The current figure is counterproductive.
- the consistent and concise math notation and equation.

---

> ### Author Response · Authors · 2025-10-09
> **Response to Reviewer tgqp**
>
> We thank the reviewer for their constructive feedback and kind remarks on the potential impact of our work.
>
> **Regarding application to LLMs**
> As written in the abstract and throughout the paper, application to LLMs is a possibility that becomes feasible thanks to the efficiency of DualXDA. In our paper, however, we focus on applications to the visual domain, using models and datasets of moderate scale. This is because we are evaluating our method against 8 other methods from the literature, most of which are infeasible for modern LLM architectures. Therefore we leave the investigation of LLMs for follow-up work. We will adapt the mentions of LLM in the paper to make this clearer. Finally, as the interest in application to LLMs has also been raised by other reviewers, we are conducting an experiment on text classification with LLMs, which will be added to the appendix.
>
> **Regarding Section 2** Thank you for your detailed feedback regarding the readability of Section 2. We will shorten and improve the conciseness of this part in the final version.
> In the following, we want to address the concrete questions about the notation, and will make changes to the paper to include these explanations:
> - Global DA is a function $[N] \times K \to \mathbb{R}$ which assigns a real valued attribution score to each pair $(x_i, c)$, consisting of a training sample $x_i$ and a class $c$. This attribution indicates the relevance or influence of the training datapoint on the fit of the model, i.e. how the inclusion of the training sample in the train set changes the function $\Phi(\cdot; \theta)_c$ that assigns class scores to each training sample. “Influence” expresses that a large Global DA value means, the model output for class $c$ would have changed drastically if the training sample was not included in the train set (i.e. the norm of the delta of these functions is large), whereas a small Global DA values means that the function assigning class scores would have changed only little. We have opted to give this explanation in plain text, as opposed to mathematical formulas, because formulas would require us to introduce complicated notation for retraining etc., which would not be used elsewhere in the paper.
> - Local DA produces, for each class $c$, a function $\tau_c: \mathbb{R}^d \times [N] \to \mathbb{R}$. It produces a real-valued attribution score for any pair of test and train samples $(x_{\text{test}}, x_i)$. This attribution score indicates how the inclusion of the training sample in the train set changes the class score $\Phi(x_{\text{test}}, \theta)\_c$ for one specific test sample $x_{\text{test}}$. A positive value indicates that the inclusion of the training sample increased the class score, a negative value indicates that the inclusion decreased the class score. The absolute value of the attribution indicates the magnitude of the effect.
> - Our comment regarding the recent work by Wei et al. was intended to be a helpful point to better understand the different meanings that attribution values were given in the literature (either retraining from the beginning with a changed training set, or continued finetuning with a changed training set). We see how this comment can confuse readers, and will adjust the phrasing accordingly in the final version.
>
> Since we include 8 different methods from the literature, giving explicit formulas and explanations for each one would result in a very long Related Works section. Therefore, we have chosen to move the detailed explanations incl. formulas for the methods in Section 2.2 into the Appendix D.1, to improve readability and shorten the main part of the paper. Similarly, we have presented formulas for evaluation criteria in Appendix E. We will change the text to make sure that references to the Appendices are given at the beginning of related sections, in order to make it easy for the interested reader to reach the formulas.

---

> ### Author Response · Authors · 2025-10-09
> **Response to Reviewer tgqp (continuation)**
>
> **Regarding mathematical equations (1) - (8)** We have checked the consistency of the mathematical equations again. Below, we give an explanation of the formulas, please leave a comment in case some points remain unclear.
> Equation (1) is the standard optimization problem of a multi-class SVM, Equations (2) and (3) describe the solution to the optimization problem in Equation (1) (the detailed derivation is given in Appendix A).
> As described in the text, we are now interested in a test sample $f_{\text{test}}$ and the class score it receives from the SVM for class $c$, which is given as $w_c^\top f_{\text{test}}$.
> Applying both Equations (2) and (3) to this score yields Equation (4), which fulfills the purpose of highlighting the individual contributions of each training point $f_i$ to the class score.
> Motivated by this, Equation (5) defines the DualDA attribution as exactly these summands of Equation (4).
> Equation (6) is a reformulation of Equation (4): As the DualDA attributions are equal to the summands, the sum of all DualDA attributions is equal to the class score $w_c^\top f_{\text{test}}$. Further, as the SVM is a close fitting surrogate to the final linear layer of the original model, the SVM class score is approximately equal to the class score $\Phi(x_{\text{test}};\theta)_c$ given by the original model.
> Finally, Equations (7) and (8) belong to Theorem 3.1, the theoretical contribution of our paper, which aligns previous work in Data Attribution (specifically the method Influence Functions) with DualDA: The Influence Functions attribution is motivated by infinitesimally upweighting a single training point and approximating its effect on the class score for a test sample. Similarly, we show that when we upweight a single sample $f_i$ in our SVM optimisation problem (Equation (7)), for almost all test samples the effect of the upweighting on the class score is exactly equal to the DualDA attribution (Equation (8)), thus aligning Influence Functions and DualDA.
>
> **Regarding overview Figure 1** We appreciate the reviewer’s feedback regarding Figure 1. While we understand the concern, other reviewers have not flagged the figure as confusing. Therefore, we will retain the current design but refine the caption and add clarifying text in the main paper to further aid interpretation. Furthermore, we have made sure that all relevant equations are included in the figure: The solution to the SVM (Equation (2) and (3) is contained in Subfigure 2. The formula for the class score of a test sample (Equation (4)) is contained in Subfigure 3. The global DualDA attribution values (Equation (3)) and local DualDA attribution values (Equation (5)) are on the right side of Subfigure 4.
>
> **Regarding presentation of the paper** We thank the reviewer for their feedback regarding clarity. Following their suggestion, we will shorten the introductory sections of the paper, refine the exposition, and improve transitions to enhance readability in the final version.
>
> We thank the reviewer once again for their review, and would like to ask them to read our paper again with these explanations and reconsider their evaluation. We are also happy to clarify any points that remain unclear.

---

> > ### Comment · Reviewer_tgqp · 2025-10-20
> >
> > Thank u. My major issues in the clarity have been properly addressed.

---

> > > ### Author Response · Authors · 2025-10-23
> > > **Revision submitted**
> > >
> > > We sincerely thank the reviewers for their thorough and constructive feedback on our manuscript. We have carefully considered all comments and have prepared a revised version of the manuscript that addresses the concerns raised and incorporates the requested changes. Below, we provide an overview of the principal revisions made to the manuscript:
> > >
> > > - **Added hyperparameter recommendation (Section 4.2):** Included guidance on the default setting of $C=10^{-3}$ as the recommended hyperparameter choice for image classification tasks
> > > - **Restructured sparsity analysis (Section 4.4):** Rewrote this section to unify the perspective on positive and negative attributions by considering the sparsity of absolute attributions, thereby streamlining the analysis of DualDA's sparsity properties
> > > - **Added detailed sparsity analysis (Appendix F.7):** Provided an expanded analysis examining sparsity for positive and negative attributions separately (a reworked version of the former Section 4.4)
> > > - **Added preliminary LLM results (Appendix I):** Included preliminary findings regarding the application of DualXDA to large language models, specifically addressing resource efficiency and shortcut detection experiments
> > > - **Refined claims regarding LLM applicability:** Adjusted the language throughout the manuscript to clarify that while preliminary results for LLMs are promising, a comprehensive evaluation remains necessary and will be pursued in future work
> > > - **Enhanced readability:** Made minor revisions in teh Abstract, Section 1, and Section 2 to improve clarity and conciseness

---

> > > > ### Author Response · Authors · 2025-10-24
> > > > **Minor correction**
> > > >
> > > > We identified a minor reporting error in Table I.1: EK-FAC caching time was computed in half precision, whereas DualDA, all other values, and the main paper results used full precision. We have corrected the two affected values (caching time and factor of improvement). EK-FAC caching time is longer than previously reported, increasing DualDA’s competitive edge. This does not affect the qualitative conclusions and comparisons in the paper.

---

### Review · Reviewer_ev3s · 2025-09-15

**Summary Of Contributions:**

In this paper, the authors present a framework for scalable and interpretable data attribution in large AI models, called DualXDA. The objective is to identify which training samples influence model predictions both globally, i.e., their effect on the model parameters obtained during training, and locally, i.e., the influence of a training instance on the prediction of a test sample. The literature already provides methods for this task, but existing approaches are computationally expensive and yield dense, hard-to-interpret results.

DualXDA addresses these issues with two components: DualDA, which uses multiclass kernel SVMs as surrogates to provide sparse, faithful, and highly efficient attributions that can be both global and local; and XDA, which integrates feature attribution methods such as Layer-wise Relevance Propagation to explain why certain training samples matter by highlighting relevant input features in the prediction of a test sample.

The paper presents extensive experiments on three well-known computer vision datasets, state-of-the-art computer vision models, and seven established performance metrics for explainability. Moreover, it includes four guided examples showing how the feature attribution component of the framework can highlight its significance. The experimental results demonstrate that DualDA matches or surpasses state-of-the-art attribution quality while achieving massive speedups, and that XDA provides richer, human-understandable, feature-level interpretability.

**Strengths**
- Comprehensive review of previous data attribution approaches. The paper also provides detailed explanations of the connections between DualDA and existing methods such as Influence Functions and Representer Points, with formal proofs.
- Clear motivation and design of the framework.
- All theoretical results are formally proven in the Appendix.
- Rich and curated experimental evaluation that strongly motivates the effectiveness of the framework. Moreover, the additional explanations about the metrics used makes a lot more clear the experimental comparison with the baselines.
- Clear advantages of the proposed framework: the surrogate SVMs are faithful, the achieved speedup is notable, and the framework enables sparse data and feature attributions that are more interpretable.

**Weaknesses**
- I had a hard time to understand the purpose and the meaning of the first experiment presented in Section 4.4: why is it relevant to evaluate the `overall ration of cumulative positive attribution versus negative attribution` as a measure of attribution sparsity?
- One of the main goals of the paper is to enable explainable data attributions for large AI models. However, the evaluation focuses on standard computer vision models such as ResNet-50, VGG-16, and ResNet-18, which are also considered in prior work, showing the speed-up of the proposed method against previously proposed attribution methods. However, it would have been interesting to consider even larger models, such as large language models. This limitation is acknowledged by the authors as future work.

**Audience:**

Yes

**Audience Explanation:**

The findings of this paper are certainly of interest to the TMLR community. Explainability is a hot topic in artificial intelligence, and providing explainable data attribution is essential for better understanding the training and prediction dynamics of models that are not easily interpretable, such as neural networks. Since this work proposes a method that delivers faithful and explainable data and feature attributions with improved efficiency, it has the potential to become the new state-of-the-art approach for this task and to generate significant impact.

**Broader Impact Concerns:**

No broader impact concerns.

**Claims And Evidence:**

Yes

**Claims Explanation:**

In my opinion, the evidence provided by the paper to support its claims is accurate, clear, and convincing, with the exception of one point regarding the evaluation of the sparsity of the attributions, which the authors should clarify. Apart from this, all theoretical results are formally proven, and the experimental evaluation is sufficiently comprehensive to demonstrate the effectiveness of the proposed framework, its efficiency compared to state-of-the-art methods, and the usefulness of the approach. Moreover, an intriguing aspect of the paper is the explanation of the connections between the proposed framework, the Representer Points method, and the Influence Functions method.

**Requested Changes:**

**Changes critical for acceptance**
- I suggest that the authors provide an explanation about the relevance of the `overall ration of cumulative positive attribution versus negative attribution` to evaluate the sparsity of the attributions in Section 4.4.

**Changes that would strengthen the work**
- Even though the authors indicate this point as a limitation to be addressed in future work, it could be interesting to provide results about the scalability of the framework when applied to models that are even larger than the ones considered before in other papers, like large-language models.

---

> ### Author Response · Authors · 2025-10-17
>
> We thank the reviewer for their constructive feedback and positive assessment of our work.
>
> **Regarding Sparsity Analysis in Section 4.4** We thank the reviewer for their feedback regarding Section 4.4 and Subfigure 5 (left). The curves presented in Subfigure 5 (right) display how many training samples are needed to explain a desired amount of overall positive or negative attribution. Additionally, we compared the overall positive and negative attribution for each method so that the reader understands their respective proportions to contextualise the curves on the right. Given that positive and negative attributions carry distinct semantic interpretations, representing evidence supporting versus contradicting a class prediction, a rigorous analysis of their respective distributions is worthwhile. However, as noticed by the reviewer, this analysis contains complementary information that is not directly needed to evaluate the level of sparsity of each method. In order to display a concise evaluation, we will show a single plot displaying the ratio of training samples accounting for a given percentage of overall **absolute** attributions, and move the current detailed plots to a dedicated appendix. We hope this will improve the readability of the main paper and put into better focus the results about sparsity of the methods.

---

> ### Author Response · Authors · 2025-10-17
>
> **Regarding applications to LLMs** As multiple reviewers expressed interest, we decided to investigate the scalability of the DualDA approach in the context of Large Language Models in an additional section in the appendix, comparing its resource consumption to EK-FAC, which is the most efficient gradient-based variant included in our study.
>
> **Set-Up** We finetune a Llama 3.2 1B model [1] for the task of topic classification on the AG's News Topic Classification Dataset [2], consisting of 120,000 training samples and 7,600 test samples.
>
> **Memory Consumption** DualDA requires 938MBs for caching the penultimate features for the training dataset and an additional 1.9 MB for attribution related variables for all hyperparameters, while EK-FAC requires 33 GBs of caching space, 11$\times$ more memory. Alternatively, it provides the option of using half precision for floating point values.
>
> **VRAM Usage** When creating attributions for a test batch size of 4, DualDA requires 8.7 GB of VRAM. EK-FAC’s official implementation lets the user configure the batch size for computing training gradients, allowing to reduce the VRAM usage. However, since the computation time cost is mostly due to gradient computations and fixed size matrix operations, this does not result in substantial reductions in the overall computation time, as will be reported in the next two sections. With the same test batch size of 4, EK-FAC requires 149.9 GB of VRAM with a training batch size of 32, 86.9 GB with a training batch size of 16, 54.1 GB with a training batch size of 8, and 37.6 GB with a training batch size of 4, 29.5 GB with a training batch size of 2, and in the minimal setup, 25.6 GB with a batch size of 1.
>
> **Cache time** After caching all training features by running a single forward pass of the model for the entire train set, training the SVM for DualDA attributions takes 18m 26s  ($C=10^{-1}$) / 48s ($C=10^{-3}$) / 33s ($C=10^{-5}$), respectively. EK-FAC requires 110m 27s of cache time irrespective of the training batch size, making DualDA 6 to 200$\times$ faster.
>
> **Explanation Time** The generation of DualDA attributions after caching takes on average 0.04s per test sample for all choices of hyperparameter $C$. In comparison, EK-FAC requires on average 73m 23s per test sample with a training batch size of 32, and 72m 25s with a training batch size of 16, thus slower than DualDA by a factor of 110000$\times$ and 108000 $\times$ respectively.
>
> **Qualitative Experiments**  To evaluate attribution quality, we conduct a Shortcut Detection experiment: for half of the training samples, we randomly reassign the label to a class *cls* and insert a corresponding shortcut keyword  `<cls>` at a random position within the sample text. We apply the same manipulation to create a shortcutted test set. Since models exploiting these shortcuts should rely heavily on the shortcutted training samples, we hypothesize that data attribution methods identify these samples when explaining predictions on shortcutted test instances. We quantify the overall effect of each training sample on the entire test set by summing its attributions across all shortcutted test instances. With appropriate hyperparameters, DualDA assigns the top-500 attribution scores exclusively to shortcutted training samples These preliminary results demonstrate that DualDA effectively identifies training samples responsible for shortcut learning in LLM contexts. We will provide detailed findings in the appendix and plan to extend this investigation in future work.
>
>
> We thank the reviewer once again for their thoughtful review and hope for a positive assessment, as all requested changes have been addressed. We are happy to clarify any remaining points if needed.
>
> [1] https://huggingface.co/meta-llama/Llama-3.2-1B
> [2] https://huggingface.co/datasets/sh0416/ag_news

---

> > ### Comment · Reviewer_ev3s · 2025-10-18
> > **Thank you for the response**
> >
> > I thank the authors for their response, that partially clarifies the points raised in my review. The additional experimental results are convincing in demonstrating the effectiveness of DualDA in a specific setting involving an LLM. However, I suggest clearly stating as a limitation that the applicability of the method to other, more real-world-oriented scenarios will be explored in future work.

---

> > > ### Author Response · Authors · 2025-10-20
> > > **Response to Reviewer ev3s**
> > >
> > > We thank the reviewer for their response. We are actively finalizing a revised version of the manuscript including the requested changes and will upload it as soon as possible. In the updated version, it will be clearly stated as a limitation that the applicability of DualXDA to real-world-scenarios in which LLMs are used will be analysed thoroughly in future work.
> > >
> > > We hope that all remaining questions have been clarified adequately and remain happy to address any remaining issues.

---

> > > ### Author Response · Authors · 2025-10-23
> > > **Revision Submitted**
> > >
> > > We sincerely thank the reviewers for their thorough and constructive feedback on our manuscript. We have carefully considered all comments and have prepared a revised version of the manuscript that addresses the concerns raised and incorporates the requested changes. Below, we provide an overview of the principal revisions made to the manuscript:
> > >
> > > - **Added hyperparameter recommendation (Section 4.2):** Included guidance on the default setting of $C=10^{-3}$ as the recommended hyperparameter choice for image classification tasks
> > > - **Restructured sparsity analysis (Section 4.4):** Rewrote this section to unify the perspective on positive and negative attributions by considering the sparsity of absolute attributions, thereby streamlining the analysis of DualDA's sparsity properties
> > > - **Added detailed sparsity analysis (Appendix F.7):** Provided an expanded analysis examining sparsity for positive and negative attributions separately (a reworked version of the former Section 4.4)
> > > - **Added preliminary LLM results (Appendix I):** Included preliminary findings regarding the application of DualXDA to large language models, specifically addressing resource efficiency and shortcut detection experiments
> > > - **Refined claims regarding LLM applicability:** Adjusted the language throughout the manuscript to clarify that while preliminary results for LLMs are promising, a comprehensive evaluation remains necessary and will be pursued in future work
> > > - **Enhanced readability:** Made minor revisions in teh Abstract, Section 1, and Section 2 to improve clarity and conciseness

---

> > > > ### Author Response · Authors · 2025-10-24
> > > > **Minor correction**
> > > >
> > > > We identified a minor reporting error in Table I.1: EK-FAC caching time was computed in half precision, whereas DualDA, all other values, and the main paper results used full precision. We have corrected the two affected values (caching time and factor of improvement). EK-FAC caching time is longer than previously reported, increasing DualDA’s competitive edge. This does not affect the qualitative conclusions and comparisons in the paper.

---

### Review · Reviewer_9qGX · 2025-10-02

**Summary Of Contributions:**

The paper proposes DualDA a new approach to data attribution for deep learning models. DualDA trains a SVM on top of the features extracted by the penultimate layer of a classifiers, and then uses the dual variables of the SVM problem to build an attribution value for each training sample given a test sample and a class. Moreover, the paper introduces XDA, a method based on LRP to explain the data attribution scores. In particular, XDA can visualize as heatmaps in the pixel space which features of a pair of training and test images contribute most to the attribution scores. In the experimental evaluation on three image classification datasets, the proposed DualDA achieves competitive or better results on average across various metrics than existing methods, while being more efficient. Finally, the paper provides qualitative examples showing that XDA can provide interpretable heatmaps for the attribution scores.

Strengths
- The proposed DualDA leverages the structure of SVMs to obtain simple and effective data attribution scores for deep networks, with only a single hyperparameter.

- The experimental evaluation, spanning multiple datasets and architectures, supports the effectiveness of DualDA, which achieves good results across different metrics. Moreover, it is significantly more efficient than the baselines.

- While XDA is tested only via qualitative examples, these suggest that it can capture meaningful explanations for the attribution scores.

Weaknesses
- The best performing method varies across metrics, so the most suitable approach may depend on the task and metric of interest. Similarly, among the variants of DualDA with different values of $C$, the best performing one varies, as well as their comparison to the baselines (Fig. 3 and Fig. 5).

- A few examples are presented for XDA, but it's hard to get an idea of its effectiveness in the average case, or what are its failure cases, without more extensive (quantitative) evaluations.

- The text suggests that the proposed method is applicable to state-of-the-art scenarios, including LLMs, but the experiments are limited to relatively small datasets and architectures, and only to image classification tasks (and it's not clear how it can be extended to others).

- The presentation is a bit lengthy at times (for example, implementation details can be deferred to the appendix).

**Audience:**

Yes

**Audience Explanation:**

Data attribution and explainability are relevant topics, and an efficient method can be of interest especially if extended to more complex tasks and  models.

**Broader Impact Concerns:**

No concerns.

**Claims And Evidence:**

Yes

**Claims Explanation:**

The experimental evaluation supports the effectiveness of DualDA, which achieves good results across different metrics. Moreover, it is significantly more efficient than the baselines. XDA examples are promising, but a more extensive evaluation would be more convincing. Finally, the text seems to suggest that the proposed method is applicable to state-of-the-art scenarios, including LLMs, but the experiments are limited to relatively small datasets and architectures, and only to image classification tasks.

**Requested Changes:**

- Since all other baselines are presented with a single set of hyperparameters, I think it would be good if a default value for $C$ in the proposed method was suggested, and convenient for potential users. Studying the effect of hyperparameter is clearly interesting, but having a standard configuration might make the main comparison to the baselines easier.

- I think the claims about the scalability to SOTA setups should be clarified, since no experiments or technical insights are provided in this direction, while there are works tackling them, e.g. [A].

- (minor) I think the presentation could be made more concise.

[A] https://arxiv.org/abs/2504.16430

---

> ### Author Response · Authors · 2025-10-17
>
> We thank the reviewer for their constructive feedback.
>
> **Regarding applications to LLMs** As multiple reviewers expressed interest, we decided to investigate the scalability of the DualDA approach in the context of Large Language Models in an additional section in the appendix, comparing its resource consumption to EK-FAC, which is the most efficient gradient-based variant included in our study..
>
> **Set-Up** We finetune a Llama 3.2 1B model [1] for the task of topic classification on the AG's News Topic Classification Dataset [2], consisting of 120,000 training samples and 7,600 test samples.
>
> **Memory Consumption** DualDA requires 938MBs for caching the penultimate features for the training dataset and an additional 1.9 MB for attribution related variables for all hyperparameters, while EK-FAC requires 33 GBs of caching space, 11$\times$ more memory. Alternatively, it provides the option of using half precision for floating point values.
>
> **VRAM Usage** When creating attributions for a test batch size of 4, DualDA requires 8.7 GB of VRAM. EK-FAC’s official implementation lets the user configure the batch size for computing training gradients, allowing to reduce the VRAM usage. However, since the computation time cost is mostly due to gradient computations and fixed size matrix operations, this does not result in substantial reductions in the overall computation time, as will be reported in the next two sections. With the same test batch size of 4, EK-FAC requires 149.9 GB of VRAM with a training batch size of 32, 86.9 GB with a training batch size of 16, 54.1 GB with a training batch size of 8, and 37.6 GB with a training batch size of 4, 29.5 GB with a training batch size of 2, and in the minimal setup, 25.6 GB with a batch size of 1.
>
> **Cache time** After caching all training features by running a single forward pass of the model for the entire train set, training the SVM for DualDA attributions takes 18m 26s  ($C=10^{-1}$) / 48s ($C=10^{-3}$) / 33s ($C=10^{-5}$), respectively. EK-FAC requires 110m 27s of cache time irrespective of the training batch size, making DualDA 6 to 200$\times$ faster.
>
> **Explanation Time** The generation of DualDA attributions after caching takes on average 0.04s per test sample for all choices of hyperparameter $C$. In comparison, EK-FAC requires on average 73m 23s per test sample with a training batch size of 32, and 72m 25s with a training batch size of 16, thus slower than DualDA by a factor of 110000$\times$ and 108000 $\times$ respectively.
>
> **Qualitative Experiments**  To evaluate attribution quality, we conduct a Shortcut Detection experiment: for half of the training samples, we randomly reassign the label to a class *cls* and insert a corresponding shortcut keyword `<cls>` at a random position within the sample text. We apply the same manipulation to create a shortcutted test set. Since models exploiting these shortcuts should rely heavily on the shortcutted training samples, we hypothesize that data attribution methods identify these samples when explaining predictions on shortcutted test instances. We quantify the overall effect of each training sample on the entire test set by summing its attributions across all shortcutted test instances. With appropriate hyperparameters, DualDA assigns the top-500 attribution scores exclusively to shortcutted training samples These preliminary results demonstrate that DualDA effectively identifies training samples responsible for shortcut learning in LLM contexts. We will provide detailed findings in the appendix and plan to extend this investigation in future work.

---

> ### Author Response · Authors · 2025-10-17
>
> **Regarding the choice of hyperparameter $C$**
> We thank the reviewer for constructive suggestions to improve the ease of application of DualXDA for users. In the paper, we report the number of support vectors, training times and surrogate accuracies for different choices of hyperparameter. These analyses suggest that $C=10^{-3}$ achieves an effective balance between low training time, high sparsity, and strong surrogate faithfulness for convolutional neural networks on image classification tasks. However, for some evaluation criteria, excessive sparsity can degrade performance (e.g. for Shortcut Detection on the AwA2 dataset). Moreover, the optimal hyperparameter setup might change for different architectures and data modalities.
>
> We will add these considerations to our paper, recommending $C=10^{-3}$ as a default value for image classification with CNN’s, while noting that hyperparameter optimization may be required for different architectures and data modalities. Importantly, DualXDA's computational efficiency makes such optimization substantially more tractable than for alternative attribution methods.
>
> **Regarding quantitative assessments of XDA** We agree that more extensive quantitative experiments for XDA would further illuminate the method’s effectiveness. Due to the lack of ground-truth explanations in the field of Feature Attribution [3], detailed quantitative evaluations require a detailed look at a variety of metrics (see e.g. [4] for a collection of common metrics), or even controlled user studies.
> In this paper, we focus on presenting the approach and demonstrating its usefulness through qualitative examples, leaving a comprehensive quantitative analysis for future work. Upon acceptance, we will release our code to enable the community to explore and evaluate XDA further.
>
> **Regarding presentation of the paper** We thank the reviewer for their feedback regarding the conciseness of the paper. Following their suggestion, we will shorten the paper and move less relevant details to the appendix.
>
> **Regarding limitation to classification**
> As DualXDA builds upon Support Vector Classifiers, the current implementation is indeed limited to classification tasks. However, the framework can be naturally extended to other problem settings by leveraging analogous sparse kernel methods, such as Support Vector Regression. We defer such extensions to future work.
>
> We thank the reviewer once again for their thoughtful review and hope for a positive assessment, as all requested changes have been addressed. We are happy to clarify any remaining points if needed.
>
>
>
> [1] https://huggingface.co/meta-llama/Llama-3.2-1B
>
> [2] https://huggingface.co/datasets/sh0416/ag_news
>
> [3] Wickstrøm, Kristoffer, Marina Höhne, and Anna Hedström. "From Flexibility to Manipulation: The Slippery Slope of XAI Evaluation." In European Conference on Computer Vision, pp. 233-250. Cham: Springer Nature Switzerland, 2024.
>
> [4] Hedström, Anna, Leander Weber, Daniel Krakowczyk, Dilyara Bareeva, Franz Motzkus, Wojciech Samek, Sebastian Lapuschkin, and Marina M-C. Höhne. "Quantus: An explainable ai toolkit for responsible evaluation of neural network explanations and beyond." Journal of Machine Learning Research 24, no. 34 (2023): 1-11.

---

> ### Comment · Reviewer_9qGX · 2025-10-17
>
> I thank the authors for the response and additional experiments. I think the new evaluation on Llama-3.2-1B further shows the computationally efficiency of the proposed approach. However, the setup is still substantially simplified (i.e., using a shortcut). Therefore, I think it'd still be important to clarify the claims about effectiveness in SOTA setups. This and the other concerns may be resolved in a revision of the manuscript, which has not been provided.

---

> > ### Author Response · Authors · 2025-10-20
> >
> > We thank the reviewer for their response. We are actively finalizing a revised version of the manuscript including the requested changes and will submit it as soon as possible.
> >
> > **On the Shortcut Evaluation Setup**
> > Although Data Attribution is formally defined as approximating the effect of retraining a model on different subsets of the training data, this baseline is inherently noisy due to stochastic factors during training such as random initialization and batch ordering [1,2]. Consequently, prior work often evaluates Data Attribution methods through downstream tasks that serve as surrogates for retraining-based evaluation [3]. In line with this established practice, we incorporated both sanity checks (Identical Class and Identical Subclass Tests) and downstream tasks (Mislabeling Detection, Shortcut Detection) in our study.
> > In response to multiple reviewers' interest in the applicability of DualDA to LLMs, we have added two components to the revised paper's appendix: (1) an analysis of computational efficiency of our approach and (2) a shortcut detection experiment for text classification. This experimental setup follows established evaluation protocols for data attribution in computer vision tasks [4,5]. We note that analyzing model behavior through fine-tuning pre-trained LLMs to exhibit specific responses when triggered by shortcuts or backdoors is a well-established methodology in the field of LLM research [6,7]. Moreover, mitigating backdoor attacks represents a fundamental challenge for scaling LLM adoption in real-world applications, underscoring the relevance of our analysis.
> > However, a comprehensive analysis of DualDA's application to LLMs would require substantial additional investigation beyond the scope of this discussion period. We will clarify in the revised manuscript that our current experiments focus on the setup described above, and that detailed evaluation of LLM applications will be the subject of future work.
> >
> > [1] Nguyen, Elisa, Minjoon Seo, and Seong Joon Oh. "A bayesian approach to analysing training data attribution in deep learning." Advances in Neural Information Processing Systems 36 (2023): 64155-64180.
> > [2] Søgaard, Anders. "Revisiting methods for finding influential examples." arXiv preprint arXiv:2111.04683 (2021).
> > [3] Nguyen, Elisa, et al. "Towards user-focused research in training data attribution for human-centered explainable AI." arXiv preprint arXiv:2409.16978 (2024).
> > [4] Koh, Pang Wei, and Percy Liang. "Understanding black-box predictions via influence functions." International conference on machine learning. PMLR, 2017.
> > [5] Hammoudeh, Zayd, and Daniel Lowd. "Identifying a training-set attack's target using renormalized influence estimation." Proceedings of the 2022 ACM SIGSAC Conference on Computer and Communications Security. 2022.
> > [6] Betley, Jan, et al. "Emergent Misalignment: Narrow finetuning can produce broadly misaligned LLMs." arXiv preprint arXiv:2502.17424 (2025).
> > [7] Souly, Alexandra, et al. "Poisoning Attacks on LLMs Require a Near-constant Number of Poison Samples." arXiv preprint arXiv:2510.07192 (2025).

---

> > ### Author Response · Authors · 2025-10-23
> > **Revision submitted**
> >
> > We sincerely thank the reviewers for their thorough and constructive feedback on our manuscript. We have carefully considered all comments and have prepared a revised version of the manuscript that addresses the concerns raised and incorporates the requested changes. Below, we provide an overview of the principal revisions made to the manuscript:
> >
> > - **Added hyperparameter recommendation (Section 4.2):** Included guidance on the default setting of $C=10^{-3}$ as the recommended hyperparameter choice for image classification tasks
> > - **Restructured sparsity analysis (Section 4.4):** Rewrote this section to unify the perspective on positive and negative attributions by considering the sparsity of absolute attributions, thereby streamlining the analysis of DualDA's sparsity properties
> > - **Added detailed sparsity analysis (Appendix F.7):** Provided an expanded analysis examining sparsity for positive and negative attributions separately (a reworked version of the former Section 4.4)
> > - **Added preliminary LLM results (Appendix I):** Included preliminary findings regarding the application of DualXDA to large language models, specifically addressing resource efficiency and shortcut detection experiments
> > - **Refined claims regarding LLM applicability:** Adjusted the language throughout the manuscript to clarify that while preliminary results for LLMs are promising, a comprehensive evaluation remains necessary and will be pursued in future work
> > - **Enhanced readability:** Made minor revisions in teh Abstract, Section 1, and Section 2 to improve clarity and conciseness

---

> > > ### Author Response · Authors · 2025-10-24
> > > **Minor correction**
> > >
> > > We identified a minor reporting error in Table I.1: EK-FAC caching time was computed in half precision, whereas DualDA, all other values, and the main paper results used full precision. We have corrected the two affected values (caching time and factor of improvement). EK-FAC caching time is longer than previously reported, increasing DualDA’s competitive edge. This does not affect the qualitative conclusions and comparisons in the paper.

---

### Comment · Editors_In_Chief · 2025-12-18

On December 18, upon request of the authors, the EiCs replaced the camera ready PDF with a new one. This includes acknowledgments that were previously omitted. Additionally, the order of the first two authors on OpenReview was swapped (to match the previous arXiv version) as compared to the original submission's author order.

---

### Decision · Action_Editor_ERwu · 2025-11-19

**Recommendation:** Accept with minor revision

**Additional Comments:**

To consider the paper for final acceptance, I would like to ask the authors to address the remaining reviewer concerns, which I share. In particular:

- Please add an explicit **Limitations** section that clearly discusses what the study covers and where it falls short. I noted the addition of Appendix I to address some concerns about applicability to LLMs, but this is not yet a satisfactory solution. The Limitations section should also highlight limitations in the evaluation (e.g., where more quantitative comparisons would be required). This should incorporate remaining reviewer comments (e.g., “more real-world-oriented scenarios should be explicitly discussed as a limitation and explored in future work”)
- Given the final set of experiments, the reference “in Large AI Models” in the title and in the first sentence of the conclusion is highly misleading and should be edited. If the authors wish to keep this phrase, it would be necessary to move (and expand -- to more models and tasks) the quantitative results and the results from Appendix I into the main paper and discuss them -- which I think is neither feasible nor advisable as it would require a major revision and extension of the results. While the method has a lot of merit, this phrasing currently overclaims the contributions, given that all main results are still generated on comparatively small ResNet models for image classification. As an easier fix, I suggest removing the “in Large AI Models” reference from the title and the corresponding sentences. The scaling aspect can be clearly positioned as future work, referencing the initial results.
- Please carefully revise the Conclusion section. Sentences like “In this work, we introduce DualXDA, a framework that allows to scale interpretable data attribution to large AI models” and “Our contributions establish a foundation for explainable AI at unprecedented scale, enabling transparent and efficient analysis of large-scale neural architectures.” should be toned down in line with the actual scope of the experiments. In general, this minor revision could be used as an opportunity to refine the Conclusions, potentially shorten them and better outline Contributions vs. Limitations of the study.

Beyond these items, the authors should carefully revisit all promises made to reviewers during the discussion phase and ensure that the paper has been updated accordingly wherever such changes were promised.

**Audience:**

Yes

**Audience Explanation:**

I agree with the reviewers that the method has a lot of potential and is highly interesting to the TMLR audience.

**Claims And Evidence:**

Yes

**Claims Explanation:**

The authors present DualXDA, an SVM-based approach for data attribution in neural network models. Compared to previous methods, this approach improves substantially in terms of speed and memory performance with implications to perform data attribution also in larger, state of the art AI models.

**Strengths** of the paper highlighted by the reviewers are “simple and effective data attribution scores for deep networks, with only a single hyperparameter” (9qGX). The reviewer also noted that the “experimental evaluation, spanning multiple datasets and architectures, supports the effectiveness of DualDA, which achieves good results across different metrics. Moreover, DualXDA is significantly more efficient than the baselines.” In this context, ev3s also highlights the positioning in the literature, and connections drawn to other methods like Influence Functions and Representer Points (in terms of experiments and theoretical connections).

The **most important weakness** both 9qGX, ev3s pointed out are some exaggerated claims about the applicability of DualXDA to large scale models. The investigation in the paper was originally mostly focused on small scale methods (e.g. up to ResNet50), plus limited to image classification as a relatively narrow application setting compared to the breadth of tasks foundation models are able to perform nowadays. In the current version of the paper -- as also noted by reviewers in their final assessment -- this point is not fully addressed, requiring a minor revision of the paper.

Concerns about the overall clarity and organization of the paper, were sufficiently addressed. Following the discussion period all reviewers were satisfied with the applied changes to the paper and recommend to accept the paper. The remaining concerns can be worked in during a minor revision.

---

> ### Author Response · Authors · 2025-12-01
> **Camera Ready Revision is submitted**
>
> We thank the Action Editor for their comments. We have revised the manuscript to accommodate the requested changes, and we summarize the changes below.
>
> **Changes in the camera-ready version**
>
> - **Limitations & Future Work section**.
>   We added a dedicated *Limitations & Future Work* section, since the limitations are naturally connected to future research directions. This new section explicitly addresses the three limitations indicated by the reviewers: the scale of current evaluation experiments, the restriction to classification settings in the current formulations, and the request for quantitative evaluation measures for XDA heatmaps.
>
> - **Clarification of experiments scale and removal of large-model claims**
>   – We updated the title to **"Sparse, Efficient and Explainable Data Attribution with DualXDA"**.
>   – We kept the mention of large models in the **abstract**, but ensured that the statement is clearly framed as *future work* rather than a claim about demonstrated applicability:
>     *"Taken together, our contributions in DualXDA ultimately point towards a future of eXplainable AI applied at unprecedented scale, enabling transparent, efficient and novel analysis of even the largest neural architectures – such as Large Language Models – and fostering a new generation of interpretable and accountable AI systems."*
>     This phrasing is intended to convey the potential of the method while making clear that such applications lie beyond the scope of the present experiments. In light of the promising preliminary results reported in Appendix I, we felt it appropriate to retain this forward-looking sentence in the abstract, while ensuring it is explicitly presented as future prospects rather than as an empirical claim.
>   – In the first section, **Introduction**, we removed the reference to large-scale applicability from the second bullet point of the contribution list. We have also removed reference to large-scale models in the final paragraph of this section.
>   – In the **Discussion & Conclusion**, we toned down all phrasing and claims implying applicability to large-scale models. We also removed three sentences that previously overstated model-scale generality, including the first sentence of the section. The revised conclusion is lighter and focuses solely on demonstrated contributions, as suggested by the AE.
>
> We believe these changes address the remaining concerns. We thank the AE and Reviewers once again, and we are happy to address any remaining concerns, in case there are any.